# Two FAM134B isoforms differentially regulate ER dynamics during myogenesis

Viviana Buonomo[1,2], Kateryna Lohachova [3,4], Alessio Reggio [1,5], Sara Cano-Franco [3,4], Michele Cillo [1,2], Lucia Santorelli [1], Rossella Venditti[1,6], Elena Polishchuk[1], Ivana Peluso[1], Lorene Brunello [3,4], Carmine Cirillo[1], Sara Petrosino[1,7], Malan Silva[8], Rossella De Cegli [1], Sabrina Di Bartolomeo[9], Cesare Gargioli [10], Paolo Swuec[8], Mirko Cortese [1,11], Alexandra Stolz [3,4], Ramachandra M Bhaskara [3,4] & Paolo Grumati [1,2✉]

## Abstract

**Endoplasmic reticulum (ER) plasticity and ER-phagy are intertwined processes essential for maintaining ER dynamics. We investigated the interplay between two isoforms of the ER-phagy receptor FAM134B in regulating ER remodeling in differentiating myoblasts. During myogenesis, the canonical FAM134B1 is degraded, while its isoform FAM134B2 is transcriptionally upregulated. The switch, favoring FAM134B2, is an important regulator of ER morphology during myogenesis. FAM134B2 partial reticulon homology domain, with its rigid conformational characteristics, enables efficient ER reshaping. FAM134B2 action increases in the active phase of differentiation leading to ER restructuring via ER-phagy, which then reverts to physiological levels when myotubes are mature and the ER is reorganized. Knocking out both FAM134B isoforms in myotubes results in an aberrant proteome landscape and the formation of dilated ER structures, both of which are rescued by FAM134B2 re-expression. Our results underscore how the fine-tuning of FAM134B isoforms and ER-phagy orchestrate the ER dynamics during myogenesis providing insights into the molecular mechanisms governing ER homeostasis in muscle cells.**

**Keywords** Autophagy; Endoplasmic Reticulum; FAM134B; Myogenesis; Reticulophagy
**Subject Categories** Autophagy & Cell Death; Musculoskeletal System; Proteomics

## Introduction

The endoplasmic reticulum (ER), also known as the sarcoplasmic reticulum (SR) in muscle cells, undergoes significant remodeling to form a compact, organized, and specialized membrane network throughout the entire length of the myotubes (Knight and Kothary, 2011; Rossi et al, 2022). The formation of an extensive ER network with highly specialized structures is one of the hallmarks of muscle differentiation thus suggesting that myogenesis involves ER membrane remodeling. A few studies indicate that ER stress-related proteins are upregulated during myogenic differentiation, potentially influencing ER morphology and function (Turishcheva et al, 2022). However, molecular mechanisms of this process remain poorly characterized. The ER is a dynamic organelle with a constantly evolving morphology. It is continuously remodeled to meet cellular needs and respond to various stimuli within its microenvironment (Pendin et al, 2011; Westrate et al, 2015). Specific resident morphogenic proteins govern the intricate shaping of the ER network, generating tubules and sheets-like structures (Park and Blackstone, 2010; Zhang and Hu, 2016). Several biological processes, such as mitosis, cell differentiation, and alterations in cellular morphology, require significant rearrangement of the ER, including membrane fusion, fission, and network reorganization. ER-phagy, a catabolic process, profoundly impacts ER morphology through the selective elimination of discrete ER portions via lysosomal degradation (Dikic, 2018; Grumati et al, 2018). This selective form of autophagy is crucial for maintaining basal cellular homeostasis and is further upregulated by stressors such as protein misfolding or calcium imbalance (Reggiori and Molinari, 2022). ER-phagy relies on specific receptors, predominantly ER resident proteins harboring LC3-interacting regions (LIRs), directly interacting with autophagy modifiers LC3s and GABARAPs (Grumati et al, 2018; Reggiori and Molinari, 2022). ER-phagy receptors not only mediate ER degradation but also

[1]Telethon Institute of Genetics and Medicine (TIGEM), 80078 Pozzuoli, Italy. [2]Department of Clinical Medicine and Surgery, Federico II University, 80131 Naples, Italy. [3]Institute of Biochemistry II, School of Medicine, Goethe University, 60590 Frankfurt am Main, Germany. [4]Buchmann Institute for Molecular Life Sciences (BMLS), Goethe University, 60438 Frankfurt am Main, Germany. [5]Saint Camillus International University of Health Sciences, 00131 Rome, Italy. [6]Department of Molecular Medicine and Medical Biotechnologies, Federico II University, 80131 Naples, Italy. [7]Department of Basic Biotechnological Sciences, Intensivological and Perioperative Clinics, Catholic University of Sacred Heart, 00136 Rome, Italy. [8]Cryo-Electron Microscopy Unit, National Facility for Structural Biology, Human Technopole, 20157 Milan, Italy. [9]Department of Biosciences and Territory, University of Molise, 86090 Pesche, Italy. [10]Department of Biology, University of Rome "Tor Vergata", 00133 Rome, Italy. [11]DISTABiF, University Luigi Vanvitelli, 81100 Caserta, Italy. ✉E-mail: p.grumati@tigem.it

contribute to ER membrane fragmentation, facilitating their engulfment into autophagosomes, thus effectively serving as ER membrane morphogens (Gubas and Dikic, 2022).

So far, several ER-resident proteins have been identified and characterized as functional ER-phagy receptors: FAM134 protein family, RTN3, SEC62, TEX264, CCPG1, ATL3, UBAC2 and others have been predicted (Khaminets et al, 2015; Fumagalli et al, 2016; Grumati et al, 2017; Smith et al, 2018; An et al, 2019; Chino et al, 2019; Chen et al, 2019; Reggio et al, 2021; Cristiani et al, 2023; He et al, 2024). The quantity and diversity of ER-phagy receptors likely stem from the heterogeneity of the ER across different cell types and tissues, resulting in varied and unequal expression across cell types and in response to distinct stimuli (Reggiori and Molinari, 2022). Notably, proteins such as the FAM134 family, RTN3 and ATL3 play a role in shaping ER membranes and also have active roles in ER-phagy, underscoring the intimate relationship between ER dynamics and ER-phagy (Gubas and Dikic, 2022). While ATL3 promotes ER membranes fusion (Rismanchi et al, 2008), members of the FAM134 family and RTN3 contribute to ER fragmentation, facilitated by their reticulon homology domain (RHD) and LIR motif (Grumati et al, 2017; Reggio et al, 2021). The RHD stabilizes highly curved bilayer structures (Voeltz et al, 2006), while the LIR motif ensures a direct interaction with the autophagy machinery (Khaminets et al, 2015; Grumati et al, 2017; Reggio et al, 2021).

FAM134B, the first ER-phagy receptor to be identified, is the most extensively studied and is recognized for its dual function in ER-phagy and ER membrane shaping (Khaminets et al, 2015). Its RHD is flanked by amphipathic helices that senses highly curved membranes and forms clusters in the peripheral ER (Bhaskara et al, 2019). Oligomerization of FAM134B prompts drastic ER membrane remodeling, resulting in budding and recruitment of autophagy machinery (González et al, 2023; Foronda et al, 2023; Berkane et al, 2023; Wang et al, 2023).

Although autophagy impairment affects ER morphology in skeletal muscles, this was associated with a general deficit of macroautophagy (Masiero et al, 2009). This raises important questions about the role of ER-phagy receptors in remodeling the ER membranes of myoblasts into the structured network of the SR. Here, we employed C2C12 myoblasts which offer valuable insights into ER dynamics during myogenesis and serve as a platform to investigate further the role of ER shaping in skeletal muscle development. Our findings highlight the pivotal role of ER-phagy in reshaping ER membranes during myoblasts differentiation, with FAM134B emerging as a critical player in maintaining ER proteostasis and facilitating ER remodeling. Notably, we identify FAM134B2, a poorly characterized isoform of FAM134B (Keles et al, 2020; Kohno et al, 2019), as essential in ensuring proper ER homeostasis in muscle cells. During differentiation, myoblasts downregulate FAM134B1, via lysosomal degradation while transcriptionally upregulating FAM134B2, highlighting its role in maintaining ER dynamics and promoting myotube formation.

# Results

## Proteomic profiling of C2C12 myoblasts during myogenesis

The differentiation of myoblasts involves intricate signaling cascades that culminate in the activation of transcriptional networks and extensive rearrangement of the proteome (Chal and Pourquié, 2017). We conducted a comparative analysis of undifferentiated C2C12 cells and myotubes proteomes following spontaneous differentiation in a standard growth medium employing a mass spectrometry (MS) approach (Figs. 1A,B and EV1A). Our results revealed significant proteomic changes upon differentiation, as evidenced by the distinct samples' separation along the principal component 1, primarily driven by alterations in sarcomere proteins (Fig. 1C; Dataset EV1). Gene ontology analysis of biological processes (GOBP) further showed a clear separation between muscle and nuclear-related terms (Fig. 1D; Dataset EV1). Notably, proteins associated with the endoplasmic and sarcoplasmic reticulum, sarcomere, lysosome, and vesicles were significantly upregulated in differentiated myotubes compared to C2C12 myoblasts (Fig. 1E; Dataset EV1). Conversely, downregulated proteins were predominantly linked to nuclear elements involved in DNA replication, splicing and ribonucleoprotein functions (Fig. 1F; Dataset EV1).

Next, we focused on proteomic analysis of cellular organelles, with the ER exhibiting the most pronounced alterations. Protein levels of numerous ER and ER-associated proteins differed significantly between myotubes and myoblasts, indicating a substantial shift in the proteomic profile of the ER (Fig. 1G,H; Dataset EV1). The altered ER components encompass various categories of proteins, including membrane-spanning, cytosolic ER-associated, luminal ER, and ER morphogens (Fig. EV1B; Dataset EV1). As evidenced by GO enrichment analysis, lysosomal proteins were upregulated in myotubes (Figs. 1E and EV1C; Dataset EV1). Furthermore, we identified several autophagy and vesicle transport-related proteins that exhibited differential expression in myotubes compared to myoblasts (Fig. EV1D; Dataset EV1). Finally, we validated our MS findings by performing western blot analysis using specific antibodies targeting ER and lysosome-associated elements (Fig. EV1E,F).

To further substantiate our results, we extended our MS analysis to primary murine (Fig. EV1G–J; Dataset EV2) and human (Fig. EV1K–N; Dataset EV3) myoblasts. Similar to our findings in C2C12 cells, proteomic analysis of murine and human myotubes revealed significant alterations in the ER proteome and the involvement of the membrane trafficking machinery during myogenesis (Fig. EV1I,J and EV1M,N). We differentiated primary myoblasts into myotubes using two different procedures: spontaneous differentiation, in a standard grow media, and myoblasts fusion stimulation employing horse serum. We did not observe significant changes in the global proteomes (Fig. EV1G–N; Datasets EV2 and 3).

These data underscore the significance of ER proteome remodeling and lysosomal pathway activation in myoblast differentiation.

## ER-phagy is activated during myogenic differentiation

As myoblasts fuse to form a syncytium during differentiation (Chal and Pourquié, 2017), we observed heterotypic fusion between ER membranes from different cells, indicating dynamic changes in ER morphology resulting in the reorganization of ER networks from different myoblasts into a more regular and structured network in myotubes (Fig. 1I). Moreover, electron microscopy further confirmed substantial remodeling of ER membranes during active

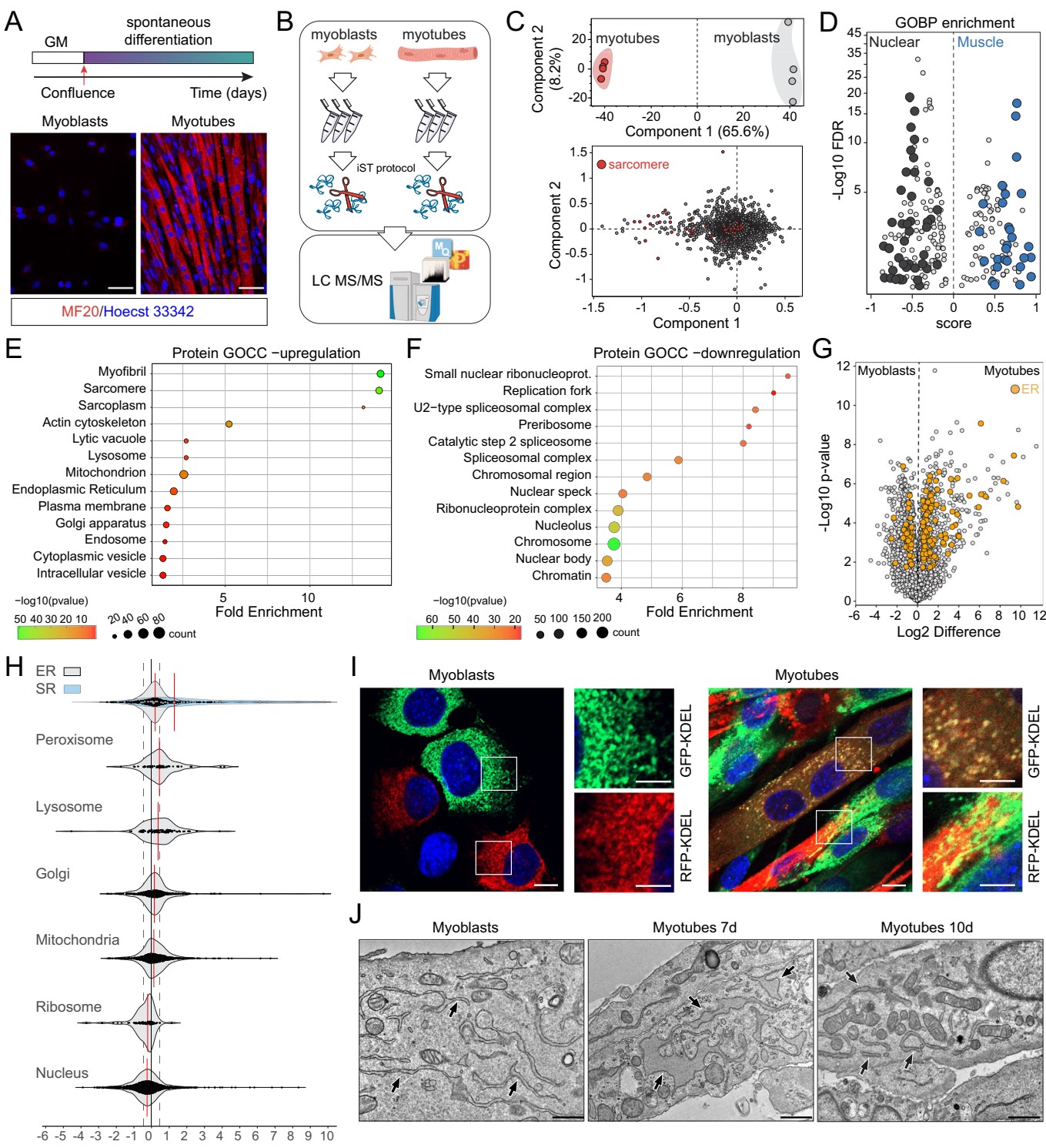

differentiation (day 7), characterized by significant membrane enlargement compared to the canonical shape observed in myoblasts. By day 10 of differentiation, ER membranes reverted to a more conventional structure and size (Fig. 1J).

The alterations in ER protein profiles and morphology coupled with the upregulation of lysosomal and autophagic proteins,

prompted us to investigate the role of ER-phagy during myogenesis. We uncovered a notable activation of the autophagy machinery during myogenesis (Figs. 2A,B and EV2A). Myoblasts were cultured in a standard growth medium that was regularly refreshed to avoid triggering autophagy via nutrient deprivation. Western blot analysis of Lc3b and p62 levels, at different time

**Figure 1.    Proteome remodeling during C2C12 myoblast differentiation in vitro.**

(A) Immunofluorescence of C2C12 myoblasts and myotubes stained with anti MF20 (Myosin heavy chain) antibody. MF20 is a marker of skeletal muscle differentiation. Scale bar: 20 μm. GM: Growth Medium. (B) Schematic of the proteomic analysis workflow for C2C12 myoblasts and myotubes. (C) Principal components analysis (PCA) of the full proteomes from C2C12 myoblasts and myotubes. The lower panel represents the loading of the PCA with hits that determine the separation, between myoblasts and myotubes, along the first component (upper panel). (D) GOBP enrichment analysis from MS data with nuclear processes marked in black and muscle-associated processes in blue. (E, F) GOCC showing classifications and fold enrichment of upregulated (E) and downregulated (F) proteins in myotubes compared to myoblasts. (G) Volcano plot displaying differences in protein expression between myoblasts and myotubes, with endoplasmic reticulum (ER)-related proteins highlighted in orange. (H) Violin plots illustrating changes in protein abundance associated with the indicated organelles between myoblasts and myotubes. (I) Confocal microscopy images of C2C12 myoblasts and differentiated myotubes overexpressing RFP-KDEL or GFP-KDEL, highlighting ER morphology and ER membranes fusion. Scale bar: 10 μm; inset scale bar: 5 μm. (J) Electron microscopy images of C2C12 cells at different stages of differentiation with black arrows indicating the ER. Scale bar: 1 μm. Mass Spectrometry was performed in quadruplicate. Imaging was replicated in three independent experiments. Source data are available online for this figure.

points during myogenesis, showed a progressive decrease in Lc3b-II and p62 protein levels, which likely indicate an activation of the autophagy flux parallel to the progression of myogenesis (Fig. EV2A). The addition of Bafilomycin A1 to undifferentiated C2C12 myoblasts resulted in the formation of a small number of autophagosome-like structures. Conversely, on day 7 of differentiation, a significant increase in the number of autophagosome-like structures was evident, which gradually decreased by day10 as myoblasts matured (Fig. 2A). To confirm that autophagy is activated during cell differentiation, we measured the autophagy flux employing stable C2C12 cells expressing GFP-LC3B-RFP reporter (Kaizuka et al, 2016). We monitored the GFP/RFP ratio that, been RFP cleaved upon LC3B lipidation, is a cell number independent readout. Of note, low GFP/RFP ratio correlates with high autophagy flux. Myoblasts had a certain basal level of autophagy flux, which increased while cells were growing and fusing and reached a new steady state when myotubes matured (Fig. 2B). Interestingly, in our EM imagines, we observed membrane structures reminiscent of ER, often decorated by ribosomes, within autolysosome structures in myotubes (Fig. 2A,C). To confirm the presence of discrete portions of ER within autophagosome-like structures, we investigated, via immunofluorescence, the co-localization of the ER marker Reep5 with Lamp1 in differentiating myotubes (Fig. EV2B,C). Moreover, we generated stable C2C12 cells expressing ssRFP-GFP-KDEL reporter (Chino et al, 2019) and monitored ER-phagy flux during cell differentiation. While undifferentiated C2C12 cells exhibited low basal levels of ER-phagy, as determined by the RFP/GFP ratio (high ratio correlates with high flux), the flux significantly increased starting from day 4 of differentiation (when myoblasts began to fuse), peaking between days 5 and 7 and gradually decreased thereafter (Figs. 2D,E and EV2D).

Thus, during muscle cells fusion, ER-phagy exhibits an oscillating pattern, starting to rise as cells begin to form contacts, peaking during cell fusion and declining as myotubes complete their maturation. This indicates that ER-phagy is only temporary needed and induced to rearrange the ER-structure.

## FAM134B regulates ER-phagy during myogenic differentiation

After establishing the activation of ER-phagy during myoblast differentiation, we focused our MS analysis on ER-phagy receptors. We identified six receptors: Fam134c, Tex264, Fam134b, Sec62, Rtn3 and Atl3; however, only Fam134b showed significant variations. Fam134b protein levels were downregulated upon differentiation

(Fig. 2F; Dataset EV1). We corroborated this finding by western blot analysis, where we observed a progressive decrease in Fam134b band intensity throughout myoblasts differentiation (Fig. 2G). Conversely, no significant differences were detected in the other receptors between C2C12 myoblasts and myotubes (Fig. EV2E).

Interestingly, only the canonical Fam134b1 isoform was degraded, whereas the shorter isoform Fam134b2 progressively increased during myoblasts differentiation (Figs. 2H and EV2F,G). Thus, the two Fam134b isoforms displayed opposite trends: Fam134b1 protein levels decreased during differentiation and were nearly absent in the mature skeletal muscle (Tibialis Anterior), whereas Fam134b2 was undetectable in myoblasts but progressively increased during their in vitro differentiation and became the dominant isoform in mature muscle (Fig. 2H,I). The reduction in Fam134b1 protein level was attributed to lysosomal degradation, as confirmed by Bafilomycin A1 treatment, which prevented its decrease (Fig. EV2H). By contrast, Fam134b2 was transcriptionally regulated, with its expression starting to increase on day 4 of differentiation (Fig. 2J). To identify any transcription factors potentially involved in Fam134b2 regulation, we employed the Enricher platform (Xie et al, 2021), using the ChEA3 database, where we uploaded all the significantly upregulated genes in myotubes identified by MS. Among the top ten enriched transcription factors, we identified four muscle-specific ones: Myog, Myod1, Myf6, and Myf5. To confirm their role, we then cloned the Fam134b2 promoter upstream of a firefly luciferase cassette and co-transfected it with plasmids coding for the above-mentioned transcription factors (all four individually). Myf6 was the transcription factor with the highest luciferase activation (Fig. 2K,L). Moreover, we performed the same assay with the Fam134b1 promoter. In this new setting, Myf5 was the transcription factor that predominantly activated the luciferase reaction (Figs. 2L and EV2I).

The myogenic process involves synchronized coordination between the remodeling of the ER membranes and the activation of ER-phagy. As myotubes mature, the ER network undergoes reshaping, leading to the re-establishment of a basal ER-phagy flux. This occurs following the lysosomal degradation of Fam134b1 and the transcriptional activation of Fam134b2. Consequently, by the end of the differentiation process, there is a complete switch between the Fam134b isoforms (Fig. 2M).

## FAM134B2 is an ER morphogen

The above results highlighted FAM134B2 as an ER protein with putative morphogenic properties. Although FAM134B2 has previously been described (Keles et al, 2020; Kohno et al, 2019), its structural characteristics and function as an ER-phagy receptor

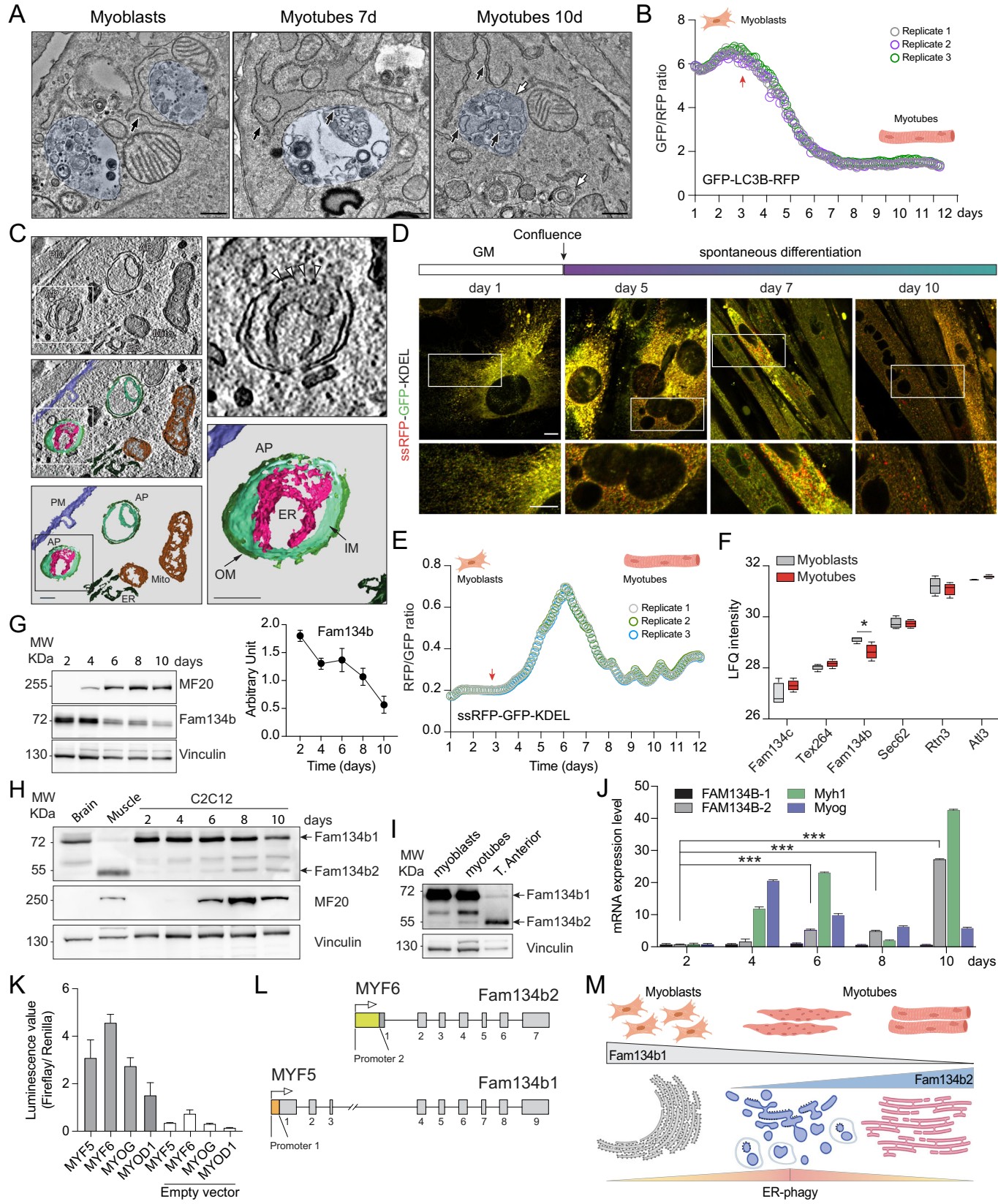

**Figure 2. ER-phagy is activated during C2C12 differentiation.**

(A) Electron microscopy images showing autophagosomes-like structures in C2C12 cells during differentiation. Black arrows indicate the ER; white arrows point to autophagosomes. Scale bar: 500 nm. (B) Analysis of macro-autophagy flux, represented by the GFP/RFP fluorescent intensities ratio, during C2C12 cells differentiation. Red arrow indicates when cells reached the full confluency. (C) Tomography (differentiating C2C12 cells) and imagine reconstruction of an autophagosome containing a portion of ER, with arrowheads indicating ribosomes. PM plasma membrane, AP autophagosome, ER endoplasmic reticulum, Mito mitochondria, OM outer autophagosomal membrane (dark green), IM inner autophagosomal membranes (light green). Magenta: ER membranes inside the autophagosome. Scale bar: 200 nm. (D) Confocal images of C2C12 cells, stably overexpressing ssRFP-GFP-KDEL during their differentiation. Scale bar: 10 μm. Inset Scale bar: 5 μm. GM: growth medium. (E) Analysis of ER-phagy flux, represented by the RFP/GFP ratio, during C2C12 cells differentiation. Red arrow indicates when cells reached the full confluency. (F) Box plot showing variations in LFQ intensities of the identified ER-phagy receptors (right panel). (G) Representative WB showing MF20 and Fam134b protein levels in C2C12 cells during their differentiation into myotubes. A graph displaying the densitometric analysis of Fam134b1 bands. (H) Representative WB displaying Fam134b1 and Fam134b2 protein levels during C2C12 differentiation, with MF20 serving as a marker of muscle differentiation. (I) Representative WB of Fam134b1 and Fam134b2 in myoblasts, myotubes and mature skeletal muscle (Tibialis Anterior). Skeletal muscle and brain samples are positive and negative controls. (J) Real-time PCR analysis of the indicated genes during C2C12 cells differentiation. (K) Luciferase assays assessing the activity of the indicated transcription factors when co-transfected, in HEK293T, with a plasmid containing the firefly luciferase downstream the Fam134b2 promoter. (L) Schematic illustration of the FAM134B isoforms and associated transcription factors. FAM134B1 and FAM134B2 promoters are highlighted. (M) Schematic representation of ER remodeling, ER-phagy flux, and the switch between FAM134B isoforms during myogenesis. All data are represented as mean ± s.d.; *$P < 0.05$. ***$P < 0.001$ (two-way ANOVA). All the experiments were replicated in triplicate. Source data are available online for this figure.

remain unclear. FAM134B2 is essentially a truncated isoform of FAM134B1, lacking the N-terminus domain and the first intramembrane hairpin structure (TM1/2). To elucidate the differences between FAM134B1 and FAM134B2, we compared their sequences and found that the helical segments of FAM134B2 were topologically equivalent to the TM3/4, $AH_L$, and $AH_c$ segments of the canonical FAM134B1-RHD (Appendix Fig. S1A). Moreover, previous sequence analyses identified this region of the RHD as highly conserved among all homologs within the FAM134 family, indicating significant functional constraints (Reggio et al, 2021). In addition, the entire C-terminus portion, including the LIR motif, is identical to that of the FAM134B1 isoform (Keles et al, 2020; Kohno et al, 2019), suggesting that FAM134B2 could function as an ER-phagy receptor too (Appendix Fig. S1A).

We first selectively expressed FAM134B2 in C2C12 cells to confirm its subcellular localization exclusively to the ER (Fig. 3A; Appendix Fig. S1B) and investigated the topology of the truncated RHD of FAM134B2. We conducted a fluorescent-based protein protection assay by utilizing stable and inducible C2C12 cells expressing HA-FAM134B2 (with the HA tag at the N-terminus). Following permeabilization with digitonin, which allows to permeabilize only the plasma membrane, we detected HA signals, indicating a cytosolic orientation of its N-terminus. For comparison, we confirmed the topology of FAM134B1 performing a similar experiment that was consistent with the previously findings (Khaminets et al, 2015) (Fig. 3B; Appendix Fig. S1C).

The role of FAM134B1-RHD in binding and shaping ER membranes is well-established (Bhaskara et al, 2019). To explore whether the preserved structural elements of FAM134B2 retain the RHD functions, we modeled its intramembrane region and embedded it in model lipid bilayers. Subsequently, we conducted coarse-grained molecular dynamics (MD) simulations and compared the data with those of the intact RHD structure of FAM134B1 (Fig. 3C; Dataset EV4). While FAM134B1-RHD adopted a wedge shape characterized by dynamic intramolecular interactions between intra-membrane hairpins TM1/2 and TM3/4, FAM134B2-RHD displayed a unique structure. It lacked the intramembrane hairpin TM1/2 and had only a single helical hairpin (TM3/4), flanked by the two amphipathic segments, $AH_L$ and $AH_C$. Although FAM134B2 partially resembled the wedge shape of FAM134B1, it displayed considerably reduced dynamics,

with reduced fluctuations and increased membrane footprint in our MD simulations (Fig. 3D,E; Appendix Fig. S1D,E). A detailed conformational analysis revealed that the intact FAM134B1-RHD was more dynamic and populated various conformations with 77 clusters, 15 of which were unique. The top 3 clusters accounted for ~30% of the total conformations and displayed different inter-hairpin arrangements, indicative of dynamic wedging. By contrast, the conformations of the partial-RHD structure of FAM134B2 populated a single dominant cluster, accounting for 88.12% of its conformations (in a total of 7 clusters) (Fig. 3D). This cluster represented more closely related conformations (RMSD $\leq$ 0.5 nm), indicating a more rigid partial-RHD structure (Appendix Fig. S1E). Next, we investigated whether these structural divergencies translated into differences in their ability to induce membrane curvature. We simulated discontinuous flat bicelle patches made of DMPC and DHPC lipids, embedded them with the FAM134B1 or FAM134B2 molecules, and monitored the shape changes in the bicelle. The embedded molecules perturbed the bicelle structure, causing them to bend and form cup-shaped structures before adopting a closed vesicle structure. We assessed the proteins' curvature induction capacity and the vesicle formation kinetics by running an ensemble of patch-closure simulations. The simulations revealed that both FAM134B2 and FAM134B1 could induce spontaneous bicelle-to-vesicle transitions within the simulation time. This confirms that despite structural disparities, both molecules exhibited an ability to induce membrane curvature and remodel flat bicelles into closed vesicles (Fig. 3F,G; Appendix Fig. S1F,G). Moreover, FAM134B1 and FAM134B2 also exhibited similar kinetics: $k_{FAM134B} = 0.0026$ ns$^{-1}$ and $k_{FAM134B2} = 0.0027$ ns$^{-1}$ (Fig. 3G). Finally, to test if FAM134B2 remodels membranes in vivo, we assessed FAM134B2's ability to induce ER vesiculation in C2C12 cells. Notably, overexpression of FAM134B2 in C2C12 cells was sufficient to fragment ER membranes, albeit at a slower rate compared to FAM134B1 (Figs. 3H,I and EV2J). The difference between the two isoforms was more pronounced in myotubes, where FAM134B2 overexpression regenerated fewer ER fragments as opposed to FAM134B1 (Fig. 3J,K). Of note, FAM134B1 constitutive expression, during cell differentiation, affected myotubes development (Fig. 3L,M). Differently, FAM134B2 over-expression did not affect myotubes formation neither showed additive effects compared to wild-type C2C12 (Fig. 3L,M).

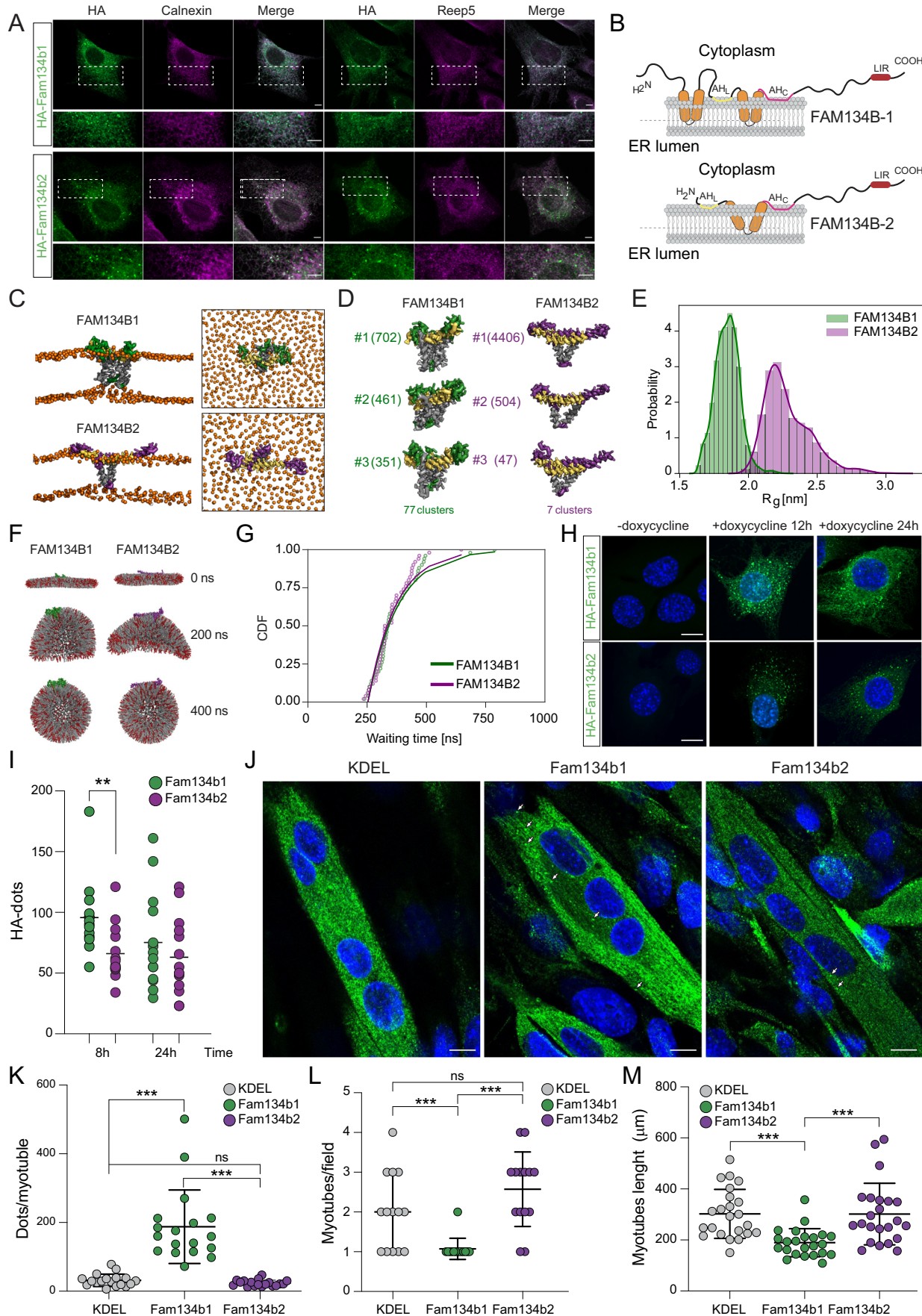

◀ **Figure 3. Structure and dynamics of FAM134B2.**

(A) Confocal images of C2C12 cells overexpressing HA-Fam134b1 and HA-Fam134b2 following 24 h of doxycycline induction and stained for HA and endogenous ER markers Calnexin and Reep5. Scale bar: 20 μm. Inset Scale bar: 5 μm. (B) Schematic representation of the structural domains of FAM134B1 and FAM134B2. (C) Representative snapshots from coarse-grained molecular dynamics (MD) simulations showing the conformations of the complete reticulon homology domain (RHD) structure of FAM134B (top) and the partial RHD structure of FAM134B2 (bottom) embedded within independent POPC bilayers (PO4 groups represented by orange spheres). Transmembrane (TM) hairpins, amphipathic helices (AH), and cytosolic loops are highlighted in gray, yellow, and green/purple, respectively. (D) The top three conformational states characteristic of the dynamic shapes of FAM134B1- and FAM134B2-RHD. (E) Distribution of the radius of gyration ($R_g$) showing the spatial footprints of FAM134B1 and FAM134B2 structures on the bilayer, based on their conformational sampling. (F) Snapshots from simulations depicting the bicelle-to-vesicle transition, with bicelles containing long-chain DMPC (gray) and short-chain DHPC (red) lipids embedded with FAM134B1 (green) and FAM134B2 (purple) molecules, which induce positive directional curvatures within these structures. (G) Cumulative distribution functions (CDF) capturing the kinetics of bicelle-to-vesicle transitions based on data from 50 simulation replicates. Individual data points are shown as circle, with solid lines representing fitted CDFs using a model with a single Poisson process model with constant lag time. (H) Immunofluorescence images of C2C12 cells overexpressing HA-Fam134b proteins, stained with anti HA antibody, after 12 and 24 h doxycycline induction. Cells were maintained under basal conditions with 200 nM Bafilomycin A1 added 2 h prior to fixation. Scale bar: 20 μm. (I) Quantitative analysis of HA-positive dot-like structures in C2C12 cells overexpressing either Fam134b1 or Fam134b2. (J) Immunofluorescence images of myotubes overexpressing Fam134b proteins (or KDEL as control), stained with anti Fam134b antibody, after 12 h doxycycline induction. Scale bar: 20 μm. K) Quantitative analysis of Fam134b positive dot-like structures in myotubes overexpressing either Fam134b1 or Fam134b2. (L, M) Quantitative analysis of myotubes number (L) and length (M) in differentiated C2C12 cells overexpressing either Fam134b1 or Fam134b2. All data are represented as mean ± s.d.; **$P < 0.01$. ***$P < 0.001$ (Student $t$ test). Imaging was replicated in three independent experiments. Source data are available online for this figure.

In summary, the partial RHD of FAM134B2 retains the ability to perturb and remodel ER membranes, albeit with a notable loss of dynamics attributed to the absence of the first intramembrane hairpin. Moreover, FAM134B2 likely confers stability to the ER network without affecting myotubes development.

## FAM134B2 is a pivotal ER-phagy receptor in muscle cells

The C-terminus intrinsically disordered region, housing the LIR motif [FELL], emerged as the most conserved feature between the two isoforms (Appendix Fig. S1A). To verify the functionality of FAM134B2's LIR motif, we performed pull-down experiments using purified GST-mATG8s and GST-Ub. Our findings revealed that FAM134B2 could bind to all six mATG8 proteins, while showing no affinity for Ub or tetra Ub (Fig. 4A). Consistently, mutations in either the LIR binding site on LC3B or within FAM134B2's LIR motif were sufficient to abolish the interaction between FAM134B2 and LC3B (Fig. 4B,C). Moreover, mutation of the LIR motif (FAM134B2ΔLIR [AELA]) resulted in the loss of ER membrane vesiculation capability and its subsequent delivery to lysosomes, thus supporting the functional role of the canonical LIR motif (Figs. 4D and EV3A,B).

We showed that FAM134B1 and FAM134B2 function as ER morphogens and ER-phagy receptors, albeit with distinct expression patterns during myogenesis. To unravel the underlying reason for the isoform switch, we generated stable and inducible C2C12 cells expressing either RFP-Fam134b1 or RFP-Fam134b2 and differentiated them into myotubes. Pull-down of Fam134b proteins, followed by IP-MS analysis, revealed that both isoforms interacted with several ER-associated proteins and mATG8s. However, Fam134b2 exhibited a lower number of interacting proteins, underscoring its close association with ER and autophagy-related factors. Of note, when we pulled down Fam134b1, we detected endogenous Fam134b2. Conversely, Fam134b1 was not among Fam134b2's significantly enriched interactors (Fig. EV3C; Dataset EV5). Moreover, the western blot uncovered high-molecular-weight bands possibly representing homo- or heterodimers of Fam134b2 protein in myotubes (Fig. EV3D). FAM134B1 forms clusters that promote ER fragmentation and ER-phagy (González et al, 2023); thus, we employed bimolecular complementation

affinity purification (BiCAP) to characterize the interactomes of FAM134B1 and FAM134B2 dimers in myotubes. We generated stable and inducible C2C12 cells co-expressing various combinations of proteins (V1-FAM134B1/V2-FAM134B1, V1-FAM134B1/V2-FAM134B2, V1-FAM134B2/V2-FAM134B2) and differentiated them into myotubes (Fig. EV3E). After 8 h of doxycycline treatment, we immuno-precipitated FAM134B complexes using anti-GFP antibodies. MS analysis revealed that FAM134B1 homo-clusters interacted with a plethora of proteins, predominantly from the ER and mitochondria (Figs. 4E and EV3F; Dataset EV6). Conversely, the FAM134B2 homo-clusters displayed a significantly lower number of interacting proteins with the ER-associated proteins, along with the mATG8s, as the primary interactors. Notably, these were not significantly enriched in the FAM134B1 homo-clusters indicating a different interaction profile for the two FAM134B isoforms (Figs. 4E and EV3F,G; Dataset EV6). The FAM134B1-FAM134B2 hetero-clusters exhibited an intermediate scenario, where the number of interactors was in between the FAM134B1 and FAM134B2 homo-clusters, with significant enrichment of mATG8s (Figs. 4E and EV3F; Dataset EV6). We further investigated the analogies and differences among the three interactomes. Several interactors were commonly shared by the three clusters, mainly ER and mitochondria-associated proteins. FAM134B1-FAM134B2 heterodimers shared most of their interactors with FAM134B1 homodimers, while the mATG8s were common with the FAM134B2 homodimers. Of note, every single combination of dimers had a percentage of unique interactors (Fig. 4F; Dataset EV6). FAM134B2 homodimers exclusive interactors were associated to developmental processes and extracellular matrix organization, while FAM134B1 homodimers showed a considerable number of unique interactors linked to ribosome and RNA processing (Figs. 4G and EV3H; Dataset EV6). Proteins associated to lysosome and endo- to lysosome transports were present among the unique interactors of FAM134B1-FAM134B2 heterodimers (Figs. 4H and EV3I; Dataset EV6).

Beside the interaction with the mATG8s, post-translational modifications, especially ubiquitination, enhances FAM134B1 clustering, forming large homomeric and heteromeric clusters and boosting ER-phagy function (González et al, 2023; Berkane et al, 2023). We investigated the ubiquitination status of

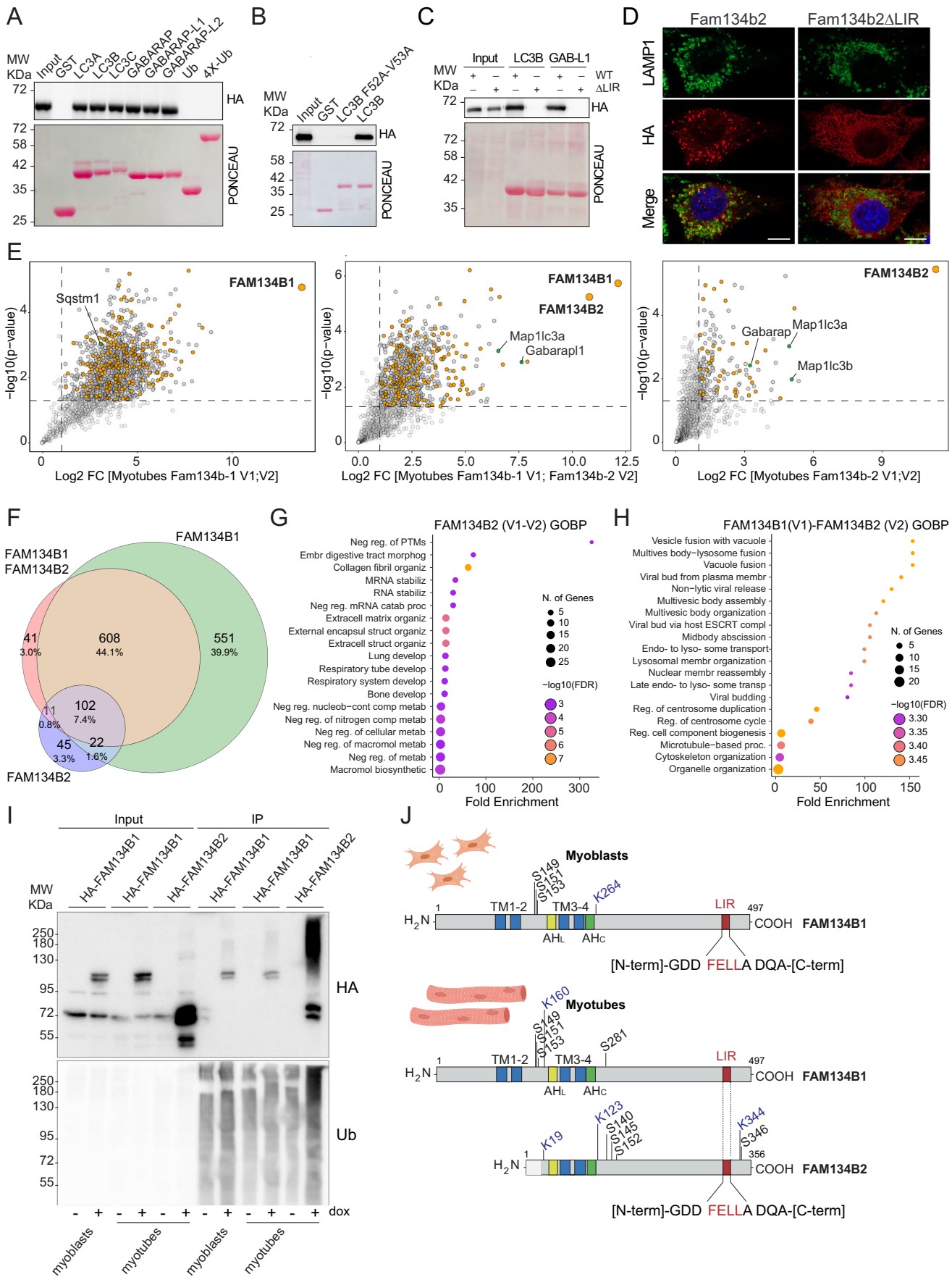

**Figure 4. FAM134B2 is an ER-phagy receptor.**

(A) GST pulldown experiment demonstrating the interaction between purified mATG8s and Ub with lysates from HEK293T cells transiently expressing HA-tagged FAM134B2. (B) GST pulldown experiment evaluating the binding affinity between purified LC3B and its mutant LC3B F52A-V53A with lysates from HEK293T cells transiently expressing HA-tagged FAM134B2. (C) GST pulldown experiment evaluating the interaction between purified LC3B and GABARAPL1, with lysates from HEK293T cells transiently expressing either HA-tagged FAM134B2 or its ΔLIR mutant. (D) Immunofluorescence images displaying LAMP1 and HA in C2C12 cells overexpressing either HA-FAM134B or the ΔLIR mutant FAM134B2 after 24 h of doxycycline induction. Scale bar: 20 µm. (E) Volcano plots showing the interactors of Fam134b1 V1-V2 complex (left), Fam134b1 V1 - Fam134b2 V2 complex (middle) and Fam134b2 V1-V2 complex (right) in differentiated myotubes, with ER proteins (orange dots) and the autophagy proteins (green dots). (F) Venn diagram of the interactors of FAM134B1 and FAM134B2 homodimers and FAM134B1-FAM134B2 heterodimers. Numbers represent the significantly enriched interactors that are unique or shared between FAM134B complexes. Percentages are calculated considering the number of all interactors for each complex. (G) GOBP enrichment analysis from FAM134B2 homodimers interactors. (H) GOBP enrichment analysis from FAM134B1-FAM134B2 heterodimers interactors. (I) TUBE2 pulldown experiments showing the ubiquitination status of FAM134B1 and FAM134B2. Pulldown experiments were performed using protein lysates from stable and inducible C2C12 and myotubes after FAM134B induction with doxycycline for 12 h. (J) Schematic representation of FAM134B1 and FAM134B2 PTMs revealed by MS in myoblasts and myotubes. Western blot and imaging were replicated in three independent experiments. Source data are available online for this figure.

FAM134B1 and FAM134B2 proteins in myotubes employing the tandem ubiquitin-binding entity 2 (TUBE 2) system. By using protein lysates from myoblasts, expressing HA-FAM134B1, and myotubes, expressing HA-FAM134B1 or HA-FAM134B2, we detected a significant amount of ubiquitinated FAM134B2 in myotubes. Conversely, FAM134B1 ubiquitination was consistently lower in both myotubes and myoblasts (Fig. 4I). These data were supported by MS analysis on FAM134B dimers. In myotubes, we identified three ubiquitinated sites (K19, K123, K344) in FAM134B2 and only one in FAM134B1 (K160 [K19 in FAM134B2]). Of note, FAM134B1 had an ubiquitinated Lysine also in myoblasts (K264 [K123 in FAM134B2]) (Fig. 4J; Dataset EV7). In addition to ubiquitinated Lysine, we also identified four phosphorylated Serine in FAM134B2 (Ser140, Ser145, Ser152, Ser346) and four Serine in FAM134B1 (Ser149, Ser151, Ser153, Ser281 [Ser140 in FAM134B2]) myotubes samples. In undifferentiated C2C12, we identified Ser149, Ser151 and Ser153 as phosphorylated amino acid in FAM134B1(Fig. 4J; Dataset EV7).

Our data indicate that in myotubes, the two isoforms can form homo- and heterodimers with FAM134B2 showing a propensity for homodimer formation. Of note, FAM134B2 showed fewer interactors, which are primarily ER-related proteins, autophagy adaptors and development elements. Moreover, FAM134B2 is post-translationally modified via ubiquitination and phosphorylation.

## FAM134B plays a critical role in maintaining proteostasis during myogenesis

To unravel the role of FAM134B during muscle cell differentiation, we employed the CRISPR-Cas9 technology to generate C2C12 cells lacking both isoforms of Fam134b (Fam134b$^{KO}$) and explored the proteomic and transcriptomic landscape of Fam134b$^{KO}$ C2C12 myoblasts and myotubes (Fig. 5A,B). We profiled and compared the transcriptome and proteome of undifferentiated wild-type (WT) and Fam134b$^{KO}$ C2C12 myoblasts and myotubes (Fig. 5C–F; Appendix Fig. S2A–C; Datasets EV8–10).

In WT samples, 1159 entries exhibited significant alterations upon cell differentiation in transcriptomic and proteomic analyses, with a notable correlation between the two datasets. Gene ontology analysis of cellular compartments (GOCC) highlighted alterations in ER, ribosome, actin cytoskeleton, sarcomere, and nucleus (Fig. 5C). Moreover, we scrutinized the GOCC of upregulated and downregulated transcripts in C2C12 cells pre- and post-

differentiation. Among the upregulated GOCC, we identified factors related to ER, sarcomere, and cytoplasmic vesicles. Differently, significantly downregulated mRNAs were predominantly associated with ribosomal and nuclear elements (Appendix Fig. S2A). The alterations in gene transcription were confirmed by protein levels obtained from MS analysis, validating the observed changes (Datasets EV9 and 10).

Fam134b$^{KO}$ myoblasts and myotubes exhibited numerous disparities compared to WT counterparts. We identified 16122 transcripts, of which 1968 were significantly different between undifferentiated Fam134b$^{KO}$ C2C12 myoblasts and differentiated myotubes (Dataset EV9). Among the upregulated genes, sarcomere components, extracellular matrix proteins, and ER factors were the most represented. The downregulated genes were linked to nuclear elements like the WT samples (Appendix Fig. S2B). From the proteomic analysis, we detected 2703 differentially expressed proteins between Fam134b$^{KO}$ myoblasts and myotubes (Dataset EV10). Notably, ER-associated components were markedly less represented in the upregulated proteins of Fam134b$^{KO}$ myotubes compared to WT samples. Conversely, sarcomere and mitochondrial components were highly enriched. Similarly, factors associated with nuclear elements and transcription events were downregulated in Fam134b$^{KO}$ myotubes compared to WT myotubes (Appendix Fig. S2E).

Principal component analysis (PCA) of the entire proteomes revealed that WT and Fam134b$^{KO}$ C2C12 myoblasts clustered closely together, with minimal separation along the second component. Conversely, Fam134b$^{KO}$ myotubes exhibited clear separation from WT myotubes along the second component (Fig. 5E; Dataset EV10). Moreover, we compared the significantly deregulated proteins between the two genotypes in myoblasts and myotubes. More than half of the deregulated proteins were shared between the genotypes, and they exhibited a high degree of correlation (Fig. 5F; Datasets EV8 and 9). GOCC analysis revealed common deregulation of nuclear and microtubule elements, while mitochondria, ER, and vesicle components displayed diverse associations with a wide range of proteins. Some variations were shared between WT and Fam134b$^{KO}$ samples, while others were genotype-specific (Fig. 5F; Datasets EV8 and 9). ANOVA test on significantly deregulated proteins highlighted nine different clusters (Fig. 5G; Appendix Fig. S2D–F; Dataset EV10). As a general indication, myogenesis induced an upregulation of mitochondria proteins and a downregulation of nuclear elements in both WT and Fam134b$^{KO}$ myotubes respect to their undifferentiated precursor

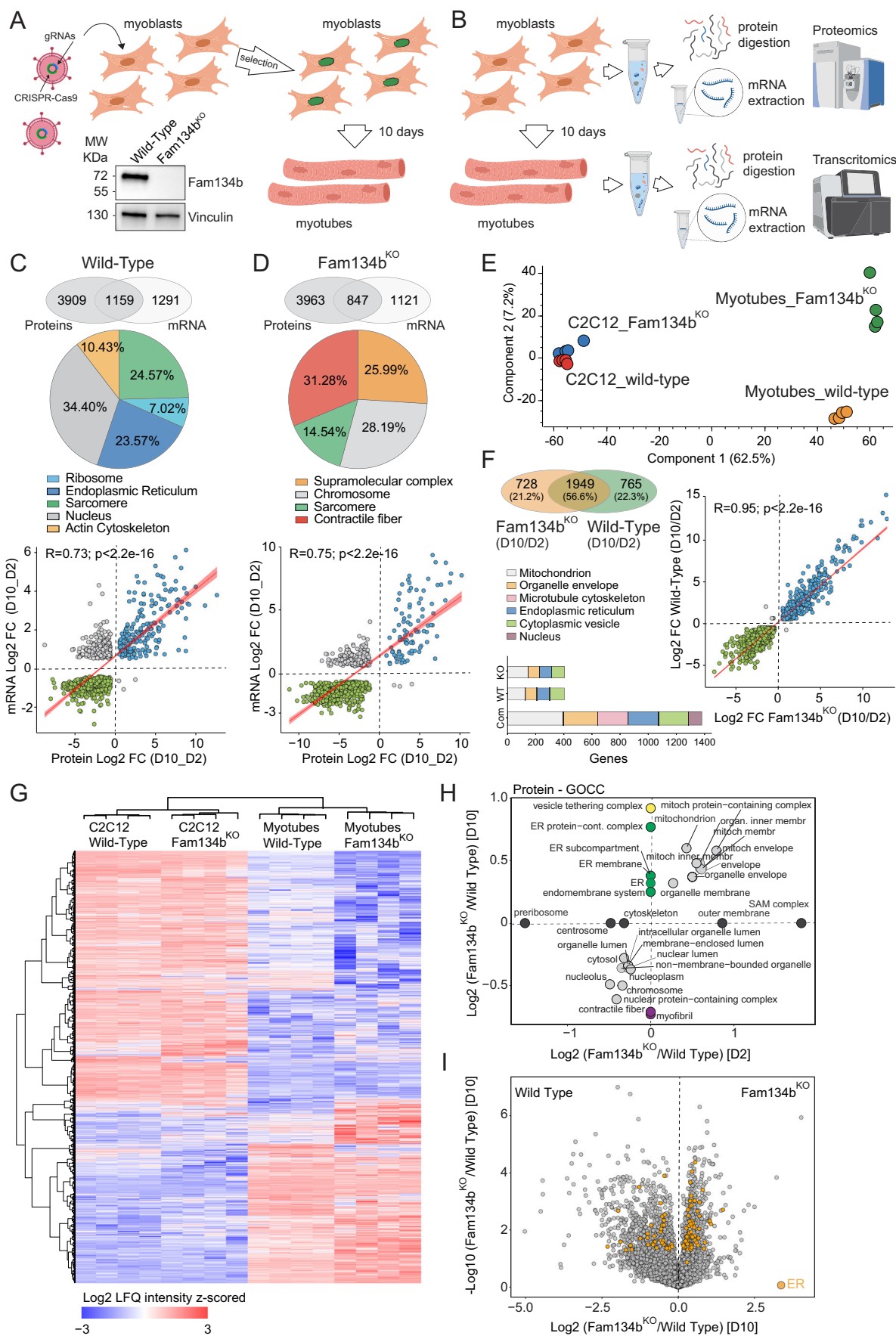

**Figure 5. Impact of Fam134b deletion on the proteome of myotubes.**

(A) Illustration of the workflow used to generate Fam134b knockout (Fam134b$^{KO}$) C2C12 myoblasts using CRISPR Cas9 technology, with a representative western blot demonstrating the absence of Fam134b in knockout myotubes. (B) Workflow diagram depicting the proteomic and transcriptomic analyses in wild-type and Fam134b$^{KO}$ C2C12 myoblasts and myotubes. (C, D) Venn diagrams indicating the overlap between the significantly deregulated proteins and mRNAs in WT (A) and Fam134b$^{KO}$ (B) C2C12 myoblasts and their corresponding myotubes. Pie charts display the most represented and commonly shared GOCC terms among proteins and mRNAs. Scatter plots illustrate the correlation between the commonly deregulated genes identified from the proteomic and transcriptomic analyses. (E) PCA of the MS full proteomes from WT and Fam134b$^{KO}$ C2C12 myoblasts and myotubes, highlighting distinct proteomic profiles. (F) Venn diagrams indicating significantly differing proteins between myoblasts and myotubes in WT and Fam134b$^{KO}$, and their overlap. The bar plot represents the GOCC of the proteins represented in the Venn diagrams. The scatter plot graph shows the correlation between the common and significantly regulated genes identified from the proteomic data of WT and Fam134b$^{KO}$ C2C12 myoblasts versus their respective myotubes. (G) Heatmap (Log$_2$ LFQ normalized intensity) of the significantly deregulated proteins (ANOVA) in WT and Fam134b$^{KO}$ myoblasts and myotubes. (H) The scatter plot graph shows the correlation between the common and significantly deregulated GOCC identified from the proteomic data of WT and Fam134b$^{KO}$ C2C12 myoblasts versus their respective myotubes. (I) Volcano plot displaying differences in protein expression between Fam134b$^{KO}$ and WT myotubes, with ER-related proteins highlighted in orange. Mass spectrometry was performed in quadruplicate. Source data are available online for this figure.

myoblasts. Of note, ER proteins were specifically accumulated in Fam134b$^{KO}$ myotubes when compared to control ones (Fig. 5H,I; Appendix Fig. S2D–F). More in details, clusters 2 and 3 showcased more pronounced differences between C2C12 myoblasts and differentiated myotubes than variability between genotypes. These clusters were predominantly associated with proteins related to muscle development, cell cycle regulation, mitochondria, and nucleus (Appendix Fig. S2D–F; Dataset EV10). Clusters 4–6 exhibited downregulation of proteins related to cell cycle, cytoskeleton organization (Cluster 4) and RNA processing (Cluster 5), mostly in myotubes compared to myoblasts (Appendix Fig. S2D–F; Dataset EV10). On the other hand, cluster 1 represented proteins associated with autophagy, mitochondria, and ER. The profile plot indicated that, in myotubes, the protein intensities were higher compared to C2C12 myoblasts in the two genotypes. However, in both undifferentiated myoblasts and myotubes, Fam134b$^{KO}$ cells showed higher protein levels compared to WT (Appendix Fig. S2D–F; Dataset EV10). Cluster 8 included mitochondria and ER proteins, showing no differences in protein levels between the two genotypes in C2C12 myoblasts. However, in myotubes, the absence of Fam134b resulted in enhanced protein levels compared to WT (Appendix Fig. S2D–F; Dataset EV10). Cluster 7 included proteins associated with vesicle-mediated transport and endosomes, showing higher protein levels in Fam134b$^{KO}$ C2C12 myoblasts compared to WT, which further increased after differentiation. Conversely, protein intensity decreased after differentiation in WT myotubes (Appendix Fig. S2D–F; Dataset EV10). Finally, cluster 9 included proteins associated with mitochondria and cytoskeletal organization, accumulating in WT myotubes compared to Fam134b$^{KO}$ (Appendix Fig. S2D–F; Dataset EV10).

These findings highlight the broad deregulation of the protein landscape during muscle differentiation in the absence of Fam134b. Notably, ER, autophagy and endosome-associated proteins demonstrated increased intensity in Fam134b$^{KO}$ myotubes compared to WT.

## C2C12 differentiation requires functional ER-phagy and autophagy

Building upon our findings regarding ER-phagy activation during myogenesis, we focused on the accumulation of ER proteins in Fam134b$^{KO}$ myotubes. Following the ANOVA analysis, we selected all proteins associated with the ER according to the GOCC and examined their intensity profiles in WT and Fam134b$^{KO}$ myoblasts and myotubes (Fig. 6A; Dataset EV10). This analysis identified six distinct clusters, each representing unique intensity profiles. Clusters 3 and 4 showcased ER proteins that exhibited similar

behavior in WT and Fam134b$^{KO}$ cells, with proteins in cluster 3 showing a lower intensity upon differentiation and proteins in cluster 4 displaying a higher intensity. Conversely, in clusters 1, 2 and 6, Fam134b$^{KO}$ myotubes exhibited higher intensity of ER proteins compared to the WT, while cluster 5 was the only cluster to present lower ER protein intensity in Fam134b$^{KO}$ compared to WT (Figs. 6A,B and EV4A; Dataset EV10). The heightened intensity of ER proteins in Fam134b$^{KO}$ myotubes suggested a generalized accumulation of ER due to defective ER-phagy flux. To test our hypothesis, we isolated lysosomal structures from WT and Fam134b$^{KO}$ myotubes and characterized their contents via MS. Fam134b$^{KO}$ lysosomes contained fewer ER-related proteins compared to WT lysosomes (Fig. 6C; Dataset EV11). We validated MS data co-staining the ER membranes and lysosomes in myotubes (Fig. 6D). In addition, we confirmed the deficit in ER-phagy in Fam134b$^{KO}$ myotubes by generating stable ssRFP-GFP-KDEL Fam134b$^{KO}$ C2C12 cells and monitoring the ER-phagy flux during their differentiation (Fig. 6E). Finally, we confirmed the accumulation of ER proteins in Fam134b$^{KO}$ myotubes via western blot (Fig. EV4A). Given that FAM134B is an autophagy receptor specific for the ER (Khaminets et al, 2015), its absence should primarily affect the selective degradation of the ER. Indeed, other forms of selective autophagy and macro-autophagy remained active, as evidenced by the analysis of macro-autophagy flux, indicating comparable activation and similar end-levels of autophagy in Fam134b$^{KO}$ myotubes (Figs. 6F and EV4B,C). Moreover, Fam134b$^{KO}$ lysosomes contained equal amounts of proteins belonging to other organelles, such as Golgi and peroxisomes and even larger amounts of mitochondrial proteins compared to WT lysosomes (Fig. 6C; Dataset EV11). These findings were further supported by electron microscopy images depicting mature autophagosomes in Fam134b$^{KO}$ myotubes (Fig. EV4D,E). Although Fam134b$^{KO}$ C2C12 cells could still form myotubes, the rate of differentiation diverged compared to that of WT myoblasts (Fig. EV4B,C,F–H). The differentiation marker MF20 (Myosin heavy chain) appeared earlier in differentiating WT cells (day4) compared to Fam134b$^{KO}$ cells (day6), with higher MF20 protein levels observed in Fam134b$^{KO}$ cells (Fig. EV4F–H). Interestingly, Fam134b$^{KO}$ cells also exhibited significantly higher Myh1 and Myog mRNA expression compared to controls (Fig. EV4G,H). Of note, we observed anomalous expression of MF20 in Atg7 knockdown C2C12, like what noted in Fam134b$^{KO}$ myotubes (Fig. EV4I).

These findings emphasize the pivotal role of Fam134b in preserving ER protein homeostasis during myoblast differentiation and highlight how disruptions in ER-phagy can hinder myogenesis.

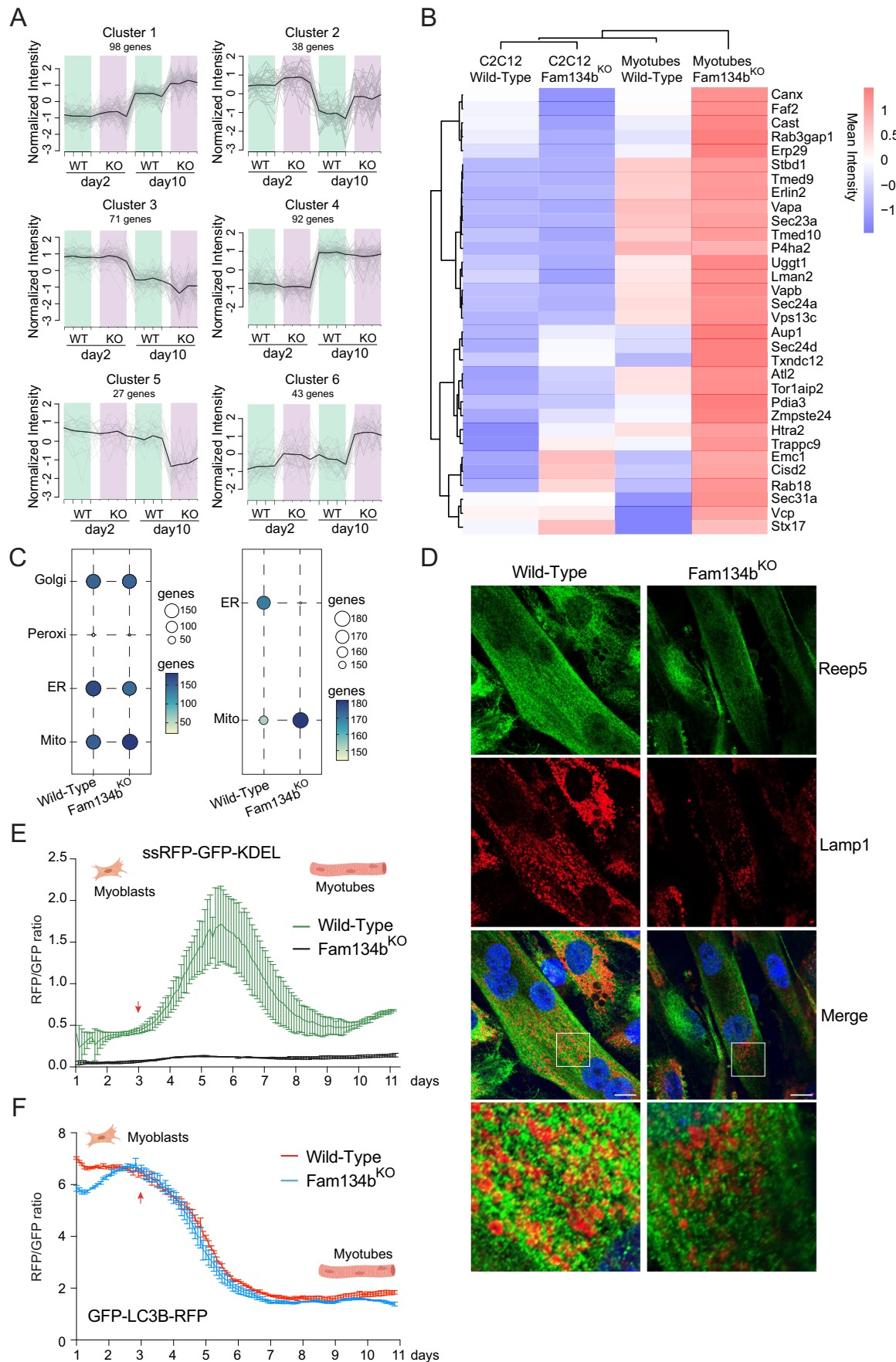

**Figure 6.  ER-phagy regulates ER dynamics during C2C12 differentiation.**

(A) Profile plots displaying the LFQ intensities of ER proteins grouped into clusters that are significantly different (ANOVA test) between WT and Fam134b[KO] C2C12 myoblasts and their respective myotubes. (B) Heatmap (Log$_2$ LFQ normalized intensity) of the ANOVA significantly altered ER proteins in WT, Fam134b[KO] myoblasts and myotubes. (C) Chart showing the frequency of proteins, with specified subcellular localization (GOCC), identified from the lysosomal IP-MS in WT and Fam134b[KO] myotubes. (D) Confocal microscopy images of WT and Fam134b[KO] myotubes stably overexpressing TMEM192-3xHA. HA labeling indicates lysosomal structure, Reep5 highlights the ER. Scale bar: 10 μm. (E) ER-phagy flux analysis, quantified by the RFP/GFP fluorescence intensity ratio, in WT and Fam134b[KO] C2C12 cells during differentiation. Red arrow indicates when cells reached the full confluency. (F) Macro-autophagy flux analysis, quantified by the GFP/RFP fluorescence intensity ratio, in WT and Fam134b[KO] C2C12 cells during differentiation. Red arrow indicates when cells reached the full confluency. Mass spectrometry was performed in quadruplicate. Imaging, ER-phagy- and autophagy flux were performed in three independent experiments. Source data are available online for this figure.

To counteract the deficiency in selective ER degradation, myoblasts upregulate macro-autophagy.

## FAM134B2 is sufficient and required to rescue ER dynamics in Fam134b[KO] myotubes

Having identified the role of ER-phagy and its receptor Fam134b in C2C12 cell differentiation, we investigated the reason behind the switch between Fam134b1 and Fam134b2 during myogenesis. To explore this, we reconstituted Fam134b[KO] C2C12 cells with constitutively expressed human FAM134B1 (hFAM134B1). In our in vitro system, FAM134B1 persisted in C2C12 cells and remained detectable in myotubes even after 10 days of differentiation, albeit with progressive degradation. In addition, we reconstituted Fam134b[KO] C2C12 cells with human FAM134B2 (hFAM134B2). Here, the expression of hFAM134B2 was inducible, under doxycycline control, allowing us to begin its expression from day 4 of differentiation to mimic its physiological transcription. Finally, we reconstituted Fam134b[KO] C2C12 cells with both isoforms, with hFAM134B1 constitutively present and hFAM134B2 induced at day 4 (Fig. EV5A). As previously evidenced by our MS data, the absence of Fam134b is mostly evident in myotubes (Fig. 5C). Consequently, we differentiated our C2C12 cell lines into myotubes and analyzed their proteomes. As anticipated, the heatmap (Log$_2$ LFQ Intensity) of the full proteomes initially revealed a distinct separation between Fam134b[KO] and WT myotubes. Moreover, all reconstituted Fam134b[KO] myotubes clustered together with the WT samples (Fig. EV5B; Dataset EV12). Notably, upon closer examination in the second level of division, WT and Fam134b[KO]-hFAM134B2 samples grouped together, while Fam134b[KO]-hFAM134B1 and Fam134b[KO]-hFAM134B1/2 proteomes formed distinct clusters (Fig. EV5B,C; Dataset EV12).

Considering the functional importance of ER-phagy, we generated a C2C12 cell line where we reintroduced the hFAM134B2ΔLIR mutant into Fam134b[KO] cells (Fig. EV5A). This cell line allowed us to investigate whether the restoration of the proteomic phenotype relied on the interaction between FAM134B2 and the autophagy machinery. Notably, Fam134b[KO]-hFAM134B2-ΔLIR myotubes clustered with Fam134b[KO] myotubes during the initial separation (Fig. EV5B; Dataset EV12). Subsequently, we conducted an in-depth analysis of our dataset by performing an ANOVA test on significantly deregulated ER proteins. Our results confirmed that only hFAM134B2 could rescue the Fam134b[KO] phenotype associated with ER proteins (Fig. 7A; Dataset EV12). Through this analysis, we categorized the ER proteins into four distinct clusters. Profile plots of the proteins showed that hFAM134B2 effectively restored ER protein intensities in cluster 4, albeit not entirely for proteins in cluster 1 (Figs. 7B and EV5D;

Dataset EV12). Cluster 4 encompassed various proteins involved in ER membrane and structural organization, muscle cell differentiation, protein synthesis, and trafficking, indicating their crucial role in ER function (Fig. 7B; Dataset EV12). Conversely, cluster 1 exhibited a significant abundance of proteins associated with vesicular trafficking, protein tethering, and transportation, underscoring their involvement in cellular transport processes (Fig. 7B; Dataset EV12). Intriguingly, hFAM134B1 exhibited an "over-correcting" effect in both cluster 4 and cluster 1, suggesting a potential dysregulation in ER protein homeostasis upon its reintroduction (Fig. 7B). In contrast, the reconstitution of hFAM134B2 and hFAM134B1 had similar effects on ER proteins in clusters 2 and 3, which notably deviated from the WT condition (Fig. EV5D; Dataset EV12). Of note, the simultaneous presence of the two isoforms had no additional or beneficial effects. FAM134B2 compensated FAM134B1's overcorrection in cluster 1 but, differently, it worsted the effect in cluster 4 (Fig. EV5E).

To further validate our findings regarding the impact of FAM134B2 and its LIR motif on ER membrane dynamics, we investigated how FAM134B2 influenced ER morphology. Electron microscopy analysis revealed significant swelling and dilation of the ER in Fam134b[KO] myoblasts, persisting throughout myogenic differentiation (Fig. 7C–E). Reconstitution of knockout cells with hFAM134B2, but not hFAM134B1, normalized ER dimensions in mature myotubes and restored ER delivery to lysosomes (Figs. 7C–E and EV5F). Moreover, ultrastructural analysis emphasized the crucial role of FAM134B2 LIR motif in rescuing ER morphology and ER delivery to lysosomes (Fig. 7D,E). High-resolution microscopy provided further insights, highlighting abnormal and enlarged ER structures in Fam134b[KO] myoblasts and myotubes. Remarkably, re-expression of hFAM134B2, at the proper time-point during myogenesis, was sufficient to reshape ER membranes and rescue ER morphology in myotubes. Moreover, the presence of the LIR motif was pivotal for promoting ER dynamics, as the reconstitution of Fam134b[KO] myotubes with hFAM134B-ΔLIR failed to rescue the ER structure (Figs. 7F and EV5G,H).

Taken together, these findings underscore the importance of FAM134B2 in myotubes stability. FAM134B2 plays a critical role in maintaining ER protein homeostasis and shaping ER membranes, both of which are important for a proper myoblasts differentiation. Importantly, the function of FAM134B2 is closely linked to its LIR motif and the autophagy machinery.

## Discussion

ER-phagy is crucial for maintaining ER protein homeostasis (Forrester et al, 2019; Cunningham et al, 2019). However, despite

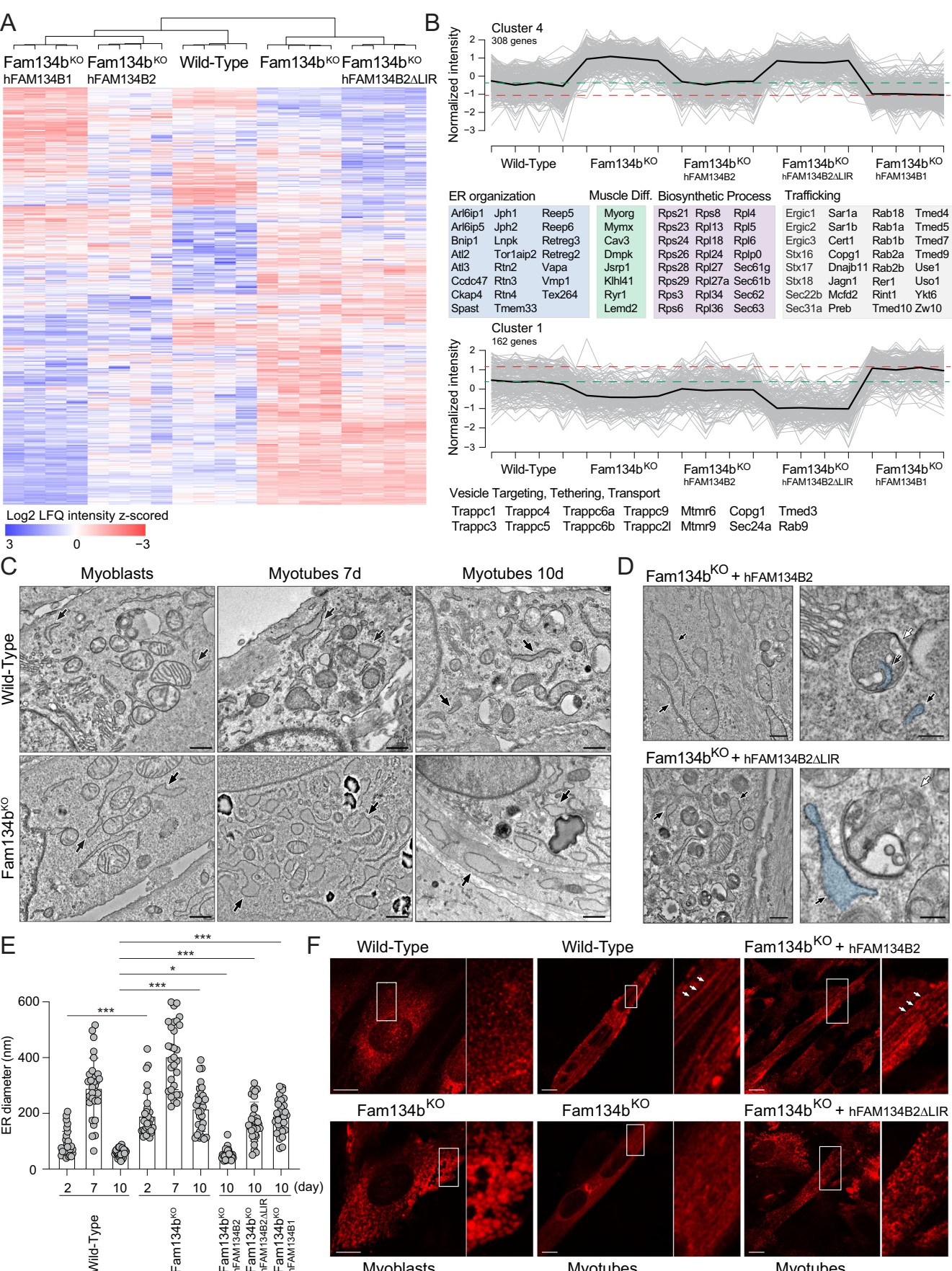

**Figure 7. Fam134b2 coordinates ER dynamics during C2C12 differentiation.**

(A) Heatmap ($Log_2$ LFQ normalized intensity) of the ANOVA significantly altered ER proteins in WT, Fam134b$^{KO}$ and reconstituted myotubes. (B) Profile plots of significantly altered ER proteins belonging to clusters 4 and 1, as identified by ANOVA. (C) Electron microscopy images displaying the ER structure in WT and Fam134b$^{KO}$ C2C12 cells at different stages of differentiation. Black arrows indicate the ER. Scale bar: 500 nm (WT); 700 nm (Fam134b$^{KO}$). (D) Electron microscopy images of the ER in Fam134b$^{KO}$ myotubes reconstituted with either wild-type hFAM134B2 or hFAM134B2ΔLIR. Scale bar: 500 nm. Insets highlight autophagosome structures. Scale bar: 200 nm. Black arrows indicate the ER; white arrows indicate autophagosomes. (E) Quantitative analysis of ER diameter in WT, Fam134b$^{KO}$, and reconstituted C2C12 cells at different stages of differentiation. (F) High-resolution microscopy images of WT, Fam134b$^{KO}$, and reconstituted Fam134b$^{KO}$ myotubes expressing RFP-KDEL, illustrating ER morphology. Scale bar: 10 µm. All data are represented as mean ± s.d.; *$P < 0.05$; ***$P < 0.001$ (Student $t$ test). Mass spectrometry was performed in quadruplicate. Imaging was performed in three independent experiments. Source data are available online for this figure.

the extensive research on ER-phagy receptors and regulatory mechanisms, its role in physiological processes remains poorly understood. Of note, ER-phagy biology is important not only for its catabolic functions but also for its impact on ER membrane remodeling (Gubas and Dikic, 2022). During neurogenesis, ER-phagy substantially contributes to ER network and proteome remodeling in neurons (Hoyer et al, 2024).

We investigated the role of ER-phagy during myogenic differentiation. This process involves the reorganization of individual ERs of myoblasts into a single highly structured membrane network in a shared cytosol (Rossi et al, 2022). We observed an increased ER-phagy flux during myoblasts differentiation, which helps to facilitate the heterotypic fusion of ER membranes, network reorganization, and remodeling of the ER proteome. Notably, FAM134B emerged as the predominant receptor in this process. Although its absence is not significantly affecting the whole differentiation process, FAM134B ablation impairs ER-phagy flux and alters ER morphology and proteome profile in differentiated myotubes. Moreover, the presence of the other ER-phagy receptors, which cannot compensate for FAM134B lack, underling the importance of this specific protein.

The classic isoform, FAM134B1, is expressed in myoblasts and degraded, via ER-phagy, during differentiation. Conversely, FAM134B2 expression increases during differentiation and becomes the dominant isoform in mature skeletal muscle. Once the differentiation is complete and the switch between FAM134 isoforms occurs, ER-phagy returns to a physiological level.

The FAM134B gene encodes at least two distinct isoforms: the full-length FAM134B1 and the N-terminus truncated FAM134B2. These isoforms exhibit differential expression and regulation patterns, varying in cell and tissue specificity and their response to stimuli (Keles et al, 2020). FAM134B2 emerged as the predominant isoform in skeletal muscles (Keles et al, 2020), which aligns with its increasing expression during myogenesis. The FAM134B2 promoter is positioned inside the intron 3 of the *FAM134B* gene, suggesting that specific transcription factors govern its expression (Keles et al, 2020). We identified Myf6 as a key transcription factor that binds the *FAM134B2* promoter, indicating its role in regulating *FAM134B2* expression. Myf6, specifically expressed in mature skeletal muscle and adult tissues (Moretti et al, 2016), is a plausible candidate for driving *FAM134B2* transcription during myogenesis. Conversely, Myf5, expressed in early somite stages and involved in directing myogenic precursors towards myoblasts differentiation (Ott et al, 1991), exhibits greater affinity for the *FAM134B1* regulatory region. The divergent transcriptional regulation of these two isoforms during muscle differentiation can be attributed to the distinct binding preferences of Myf6 and Myf5 transcription factors.

The two isoforms are also distinctly regulated at the protein level. FAM134B1 is mostly controlled by lysosomal degradation and PTMs modulate its function (Khaminets et al, 2015; González et al, 2023; Foronda et al, 2023). FAM134B2 regulation is less well understood. It appears to occur primarily at the transcriptional level and is largely cell-type and stimulus-specific (Cinque et al, 2020; Keles et al, 2020; Kohno et al, 2019; Shiozaki et al, 2022). Our observations indicate that FAM134B2 can homo- and hetero-oligomerize with FAM134B1. Moreover, FAM134B2 interacts with a limited set of proteins, mostly associated with the ER and the autophagy machinery, some of which are involved in development processes. Differently, FAM134B1 has a broader pool of interactors. The differences in the two interactomes indicates that the switch between the two isoforms could be necessary to support the turnover of unique subset of ER-phagy substrates as well as that the molecular machinery, regulating the ER-phagy process, may change from the beginning to the end of the cell differentiation. Further studies, more targeted to the specific interactors of the two FAM134B isoforms, will be informative to elucidate these points.

The propensity of FAM134B2 to oligomerize and the selectivity of its interactors could be a consequence of its unique RHD and PTMs profile. In myotubes, FAM134B2 is highly ubiquitinated and phosphorylated. These two types of PTMs play an essential role in FAM134B1 clustering formation, membranes curvature and ER-phagy regulation (Jiang et al, 2020; González et al, 2023; Berkane et al, 2023). Therefore, they could be important for FAM134B2 biochemical and biological properties too. FAM134B2 has a partial RHD; thus, many of the topologically equivalent ubiquitination and phosphorylation sites of FAM134B1-RHD are lost. In FAM134B1 myoblasts and myotubes, we identified the known phospho- and ubiquitin-sites in the RHD (González et al, 2023; Berkane et al, 2023; Jiang et al, 2020). Differently, we detected most PTMs sites after the FAM134B2 partial RHD and in the C-terminus part of the protein. The lack of ubiquitination and phosphorylation of FAM134B2 RHD likely contributes to its rigidity. Moreover, the different pattern of PTMs may mechanistically influence FAM134B2 cytosolic structure affecting protein-protein interaction, thus explaining the significant differences in the interacting partners between the two isoforms. All these observations underscore the unique nature of FAM134B2 and emphasizes that it cannot be simply regarded as a compensatory isoform.

The divergence of the two isoforms is further highlighted in Fam134b$^{KO}$ myotubes, where the expression of FAM134B2 is necessary and sufficient to rescue alterations in the proteomic landscape. Focusing on the ER proteome, FAM134B2 demonstrated the ability to restore protein intensities of several ER membrane shaping factors and proteins involved in muscle cell differentiation, vesicular trafficking, and other biological processes. However, while

FAM134B2 normalized the ER protein levels to those observed in WT samples, FAM134B1 expression led to an "overcorrection", resulting in an aberrant protein profile in Fam134b$^{KO}$ -hFAM134B1 myotubes. Of note, the switch between the isoforms is necessary for a correct ER homeostasis. In the long term, during myogenesis, FAM134B1's persistence had adverse effects that are only partially rescued by FAM134B2. Moreover, constitutive expression of FAM134B1 in WT cells significantly affects myotube maturation. Differently, FAM134B2 expression does not have significant effects on myotube formation; therefore, it rather stabilizes the differentiation process without having negative nor additive effects on myogenesis. Of note, a premature expression of FAM134B2 affects myotubes formation too. Therefore, for a correct myogenesis is necessary a fine-tuning, in time and protein levels, of the FAM134B isoforms.

Importantly, mutation of the FAM134B2 LIR motif abolished its beneficial effects, underscoring the critical role of autophagy in ER dynamics during biological processes in which ER-phagy receptors are directly involved. Notably, the biological functions of FAM134B have been closely linked to a proper cellular autophagy flux (Khaminets et al, 2015; Reggio et al, 2021; Forrester et al, 2019). Moreover, macro- and selective autophagy have been established to play a vital role during cell differentiation (Boya et al, 2018; Ordureau et al, 2021; Hoyer et al, 2024). Indeed, proper autophagy flux is required for skeletal muscle homeostasis, emphasizing the significance of these mechanisms in cellular function and differentiation (Grumati et al, 2010; Carmignac et al, 2011). We described the first, in vitro, evidence of the role of ER-phagy in skeletal muscle development. However, further studies, employing animal models, will be necessary to fully elucidate the role of ER-phagy during skeletal muscle development and regeneration.

Both FAM134B isoforms share the same active LIR motif and are recognized as ER-phagy receptors. However, the distinct properties of FAM134B2 likely stem from its partial RHD, with its rigid V-shaped structure, the different interactomes and PTMs. The reduced dynamicity of FAM134B2 seems to be advantageous, given the more organized and rigid structure of the ER network in myotubes (Rossi et al, 2008). Consequently, the switch from FAM134B1 to FAM134B2 likely reflects the adaptation of the ER proteome to the new ER morphology. As muscle matures, ER dynamicity becomes less relevant, leading to the progressive degradation of FAM134B1. Thus, the transition between the two isoforms appears to be an adaptive process, allowing for the preservation of FAM134B role in ER-phagy while modifying its morphogenic activity, protein interactions, and biological functions to suit the requirements of mature muscle fibers. The ER-phagy activity is strongly maintained reflecting the importance of autophagy in skeletal muscle (Masiero et al, 2009).

Further research into the biophysics and biochemical properties of FAM134B2 is needed to better clarify its regulation and biological functions. Exploring the balance between the two FAM134B isoforms in skeletal muscle, not only during myogenesis but also in muscle pathologies, would provide valuable insights. In conditions where muscle tissue undergoes constant cycles of degeneration and regeneration, such as in muscle dystrophies, understanding the regulation of FAM134B isoforms and their role in autophagy flux becomes particularly relevant (Bonaldo and Sandri, 2013). Moreover, considering that FAM134B2 expression has been observed in various cell types, including hepatocytes under starvation conditions and chondrocytes after stimulation

(Cinque et al, 2020; Kohno et al, 2019), it is plausible that FAM134B2 influences the ER structure and protein landscape in other tissues too. Finally, given the critical involvement of ER dynamics in major biological processes and the association of the ER morphology with various human diseases, such as axonopathies (Hübner and Kurth, 2014; Kurth et al, 2009), the role of ER-phagy and its receptors, in human pathologies, including muscle diseases, warrants thorough investigation.

# Methods

**Reagents and tools table**

| Reagent/resource | Reference or source | Identifier or catalog number |
|---|---|---|
| **Experimental models** | | |
| HEK293T | ATCC | CRL-3216 |
| C2C12 | ATCC | CRL-1772 |
| Murine primary myoblasts | Prof. Cesare Gargioli | |
| Human primary myoblasts | Prof. Cesare Gargioli | |
| **Recombinant DNA** | | |
| pLJC5_TMEM192_3XHA | Prof. Carmine Settembre | |
| pCMV-Hyg_RFP_EGFP_KDEL | Dr. Alexandra Stolz | |
| pCW57-CMV-ssRFP-GFP-KDEL | AddGene | #128257 |
| pMRX-IP-GFP-LC3B-RFP | AddGene | #84573 |
| pLTD_V1_hFAM134B1_PURO | This paper | |
| pLTD_V2_hFAM134B1_BSD | This paper | |
| pLTD_V1_hFAM134B2_PURO | This paper | |
| pLTD_V2_hFAM134B2_BSD | This paper | |
| pCMV6-Myod1 | Origene | NM_010866 |
| pCMV6-Myog | Origene | NM_031189 |
| pCMV6-Myf6 | Origene | NM_008657 |
| pCMV6-Myf5 | Origene | NM_008656 |
| pGL4.10 | Promega | AY738222 |
| pRL-TK | Promega | AF025846 |
| pGL4.10 Prom_mFam134B_iso1 | Vector Builder | On demand |
| pGL4.10 Prom_mFam134B_iso2 | This paper | |
| pKLV2.2-h7SKgRNA5(SapI)-hU6gRNA5(BbsI)-EF1a-Cas9-FLAG-P2A-BleoR | Dr. Koraljka Husnjak | |
| pLTD-HA_hFAM134B1-puro DEST | This paper | |
| pLTD-HA_hFAM134B2-puro DEST | This paper | |
| pLTD-HA_hFAM134B2_DLIR-puro DEST | This paper | |
| pLTD-RFP_hFAM134B1-puro DEST | This paper | |
| pLTD-RFP_hFAM134B2-puro DEST | This paper | |
| pLTD-RFP_hFAM134B2_DLIR-puro DEST | This paper | |

| Reagent/resource | Reference or source | Identifier or catalog number |
|---|---|---|
| pLTD-RFP_mFam134b1-puro DEST | This paper | |
| pLTD-RFP_mFam134b2-puro DEST | This paper | |
| pLV_hFAM134B_iso1-hygro | This paper | |
| pLTD-GFP_KDEL-puro | This paper | |
| pLTD-RFP_KDEL-puro | This paper | |
| **Antibodies** | | |
| HA Tag | Roche | 11867423001 |
| HA Tag | Covance | MMS-101P |
| LC3B | Nanotools (Clone5F10) | 0231-100/LC3-F10 |
| p62 | ENZO Lifescience | BML-PW9860 |
| LAMP1 | AbCam | AB2971 |
| VINCULIN | Sigma | V9264 |
| CATEPSIN D | Thermo Fisher Scient. | #PA5-17353 |
| ATL3 | Proteintech | 169-21-1-AP |
| RTN3 | Proteintech | 12055-2-AP |
| RTN4 | Cell Signaling Tech. | #13401 |
| REEP5 | Proteintech | 14643-1-AP |
| FAM134A | Sigma-Aldrich | HPA011844 |
| FAM134B | Sigma-Aldrich | HPA010277 |
| FAM134C | Sigma-Aldrich | HPA016492 |
| a TUBULIN | Cell Signaling Tech. | #2125 |
| MYOG | Santa Cruz Biotech. | sc-345 |
| MF20 | DSHB | 2147781 |
| TGN38 | Bio-Rad | AHP499 |
| CANX | AbCam | ab22595 |
| MEF2C | AbCam | ab79436 |
| VAPA | Prof De Matteis | |
| VAPB | Prof De Matteis | |
| Anti-Mouse HRP conjugate | Millipore | #12349 |
| Anti-Rabbit HRP conjugate | Millipore | #12348 |
| Anti-Rat HRP conjugate | Millipore | PI-9400-1 |
| Anti-rabbit Alexa Fluorâ 647 | Thermo Fisher Scient. | A21244 |
| Anti-rat Alexa Fluorâ 647 | Thermo Fisher Scient. | A21472 |
| Anti-rat Alexa Fluorâ 488 | Thermo Fisher Scient. | A21208 |
| **Oligonucleotides and other sequence-based reagents** | | |
| caccGAGAAACGTGAGAGATCTGGT | IDT | mFAM134B-g1FW |
| aaacACCAGATCTCTCACGTTTCTC | IDT | mFAM134B-g1RW |
| ctcGGGACCTTCAACCTTTCAGAG | IDT | mFAM134B-g2 FW |
| aacCTCTGAAAGGTTGAAGGTCCC | IDT | mFAM134B-g2RW |

| Reagent/resource | Reference or source | Identifier or catalog number |
|---|---|---|
| caccGATATCATTACATTTAAACAG | IDT | mFAM134B-g3FW |
| aaacCTGTTTAAATGTAATGATATC | IDT | mFAM134B-g3RW |
| ctcGGATGAAGATGAATTAAGCCT | IDT | mFAM134B-g4 FW |
| aacAGGCTTAATTCATCTTCATCC | IDT | mFAM134B-g4RW |
| AGTGTTTCCTCGTCC | Eurofins | GAPDH FW |
| TTCCCATTCTCGGCC | Eurofins | GAPDH RV |
| TGGCTCAGTCTGGCTCTTTC | Eurofins | FAM134B-iso1FW |
| CATAATAGTCCACTCCTCGGC | Eurofins | FAM134B-iso2FW |
| TTCTGGTTCCTTGCCTTGAC | Eurofins | FAM134B RW |
| GGGCCAACTTGTCAT | Eurofins | MYH1 FW |
| GAGGGACAGTTCATCGATAGCAA | Eurofins | MYH1 RW |
| ACCTTCCTGTCCACCTTCAG | Eurofins | MYOG FW |
| CACCGACACAGACTTCCTCT | Eurofins | MYOG RW |
| **Chemicals, enzymes, and other reagents** | | |
| Bafilomycin | Cell Signaling Tech. | 54645 |
| ProLong™ Gold Antifade Mountant | Invitrogen | P36935 |
| PhosSTOP | Roche | 4906837001 |
| protease inhibitor cocktail | Roche | 4693132001 |
| NEM | Sigma | 128287 |
| TFA | Merck | T6508-500ML-M |
| TCEP | Thermo Fisher Scient. | 77720 |
| Trypsin | Thermo Fisher Scient. | 90305 |
| Lys-C | Wako | 498748142764800 |
| Bio-Rad protein Assay Reagent | Bio-Rad | 5000006 |
| Geltrex | Thermo Fisher Scient. | A1413302 |
| HEPES | Gibco | 15630080 |
| Hygromycin | Sigma | H3274 |
| Blasticidin | Sigma | 15205 |
| Puromycin | Sigma | P8833 |
| Neomycin | Merck | N1142 |
| Jene juice | Merk | 70967 |
| Turbofect | Thermo Fisher Scient. | R0531 |
| Polybrene | Sigma | TR-1003 |
| Dulbecco's Modified Eagle Medium (DMEM) | Gibco | 41965-039 |
| Fetal Bovine Serum tetracycline free | Euroclone | ECS0182L |
| Opti-mem | Thermo Fisher Scient. | 31985062 |
| IPTG | Merck | I5502 |
| Paraformaldehyde | Sigma | D5207 |
| RFP-Trap Agarose | ChromoTek | rta-20 |
| GFP-Trap Agarose | ChromoTek | gta-20 |

| Reagent/resource | Reference or source | Identifier or catalog number |
|---|---|---|
| Glutathione Sepharose 4B beads | (GE Healthcare) | GE17-0756-01 |
| Anti-HA Magnetic Beads | Thermo Fisher Scient. | 88837 |
| TUBE2 agarose beads | Life Sensors | UM-0402-1000 |
| Negative control beads | Life Sensors | UM-0400-0400 |
| nitrocellulose transfer membrane | GVS North America | 1213314 |
| PVDF membrane | Millipore | IPVH00010 |
| Potter | Sigma | P7734-1EA |
| Reprosil-PUR, C18-AQ | Dr. Maisch | r119.aq.0001 |
| C18 (Octadecyl) | Empore | 2215 |
| SDB-RPS | Empore | 2241 |
| C18 | Bioanalytical Technol. | 2215-C18 |
| Pre-Separation Filter | Miltenyi Biotec | 130-041-407 |
| Anti-CD31 microbeads | Miltenyi Biotec | 130-097-418 |
| Anti-CD45 microbeads | Miltenyi Biotec | 130-052-301 |
| MS column | Miltenyi Biotec | 130-042-201 |
| α7-integrin microbeads | Miltenyi Biotec | 130-104-261 |
| Doxycycline | Sigma | D9891 |
| Horse serum | Euroclone | ECS0090D |
| Chicken Embryo extract | Seralab | CE-650-J |
| T4 polynucleotides kinase | NEB | M02021S |
| T4 ligase | NEB | E7664 |
| Hoechst 33342 | Merck | B2261 |
| LookOut Mycoplasma PCR Detection Kit | Sigma | MP0035-1KT |
| Gateway LR clonase mix | Thermo Fisher Scient. | 11791020 |
| Gateway BP clonase mix | Thermo Fisher Scient. | 11789020 |
| Rneasy kit | Qiagen | 74104 |
| Quantitect RT | Qiagen | 205313 |
| LightCycler 480 SYBR Green I Master | Roche | 4887352001 |
| Dual-luciferase reporter assay | Promega | E1910 |
| DH5 alpha *E. coli* competent cells | Homemade | |
| STBL3 *E. coli* competent cells | Homemade | |
| BL21 *E. coli* competent cells | Homemade | |
| **Software** | | |
| MaxQuant | Max Planck Institute of Biochemistry | |
| Perseus | Max Planck Institute of Biochemistry | |
| Spectronaut | Biognosys | |
| MARTINI force field v2.2 | | PMID: 26589065 |
| martinize.py script | | PMID: 26589065 |

| Reagent/resource | Reference or source | Identifier or catalog number |
|---|---|---|
| DSSP assignments | | PMID: 6667333 |
| OPM database and PPM web server | | PMID: 21890895 |
| AlignME | | PMID: 24753425 |
| Jalview | | PMID: 19151095 |
| AlphaFold2 | | PMID: 34265844 |
| CHARMM-GUI | | PMID: 18351591 |
| GROMACS v2020.1 | | Gromacs |
| MemCurv | Bhaskara et al, 2019 | PMID: 31147549 |
| SerialEM | | PMID: 16182563 |
| Etomo pipeline | | PMID: 8742726 |
| Enrichr (http://amp.pharm.mssm.edu/Enrichr) | Xie et al, 2021 | PMID: 23586463 PMID: 27141961 PMID: 31114921 PMID: 33780170 |
| Adobe Illustrator | | |
| BioRender | | |
| Microsoft Word and Excel | | |
| Prism 10 | | |
| Fiji | | |

## Cloning procedures and DNA mutagenesis

cDNAs were cloned into lentiviral vectors using GATEWAY (Thermo Fisher Scientific) or In-Fusion (Takara Bio) systems. For the GATEWAY cloning the cDNAs were cloned into the pDONR223 vector using the BP Clonase Reaction Kit (Thermo Fisher Scientific) and further recombined, though the LR Clonase Reaction Kit (Thermo Fisher Scientific), into GATEWAY lentiviral destination vectors: pLTD-N-HA-PUROMYCIN lentiviral vector. We designed pLTD-N-HA-BLASTICIDIN and pLTD-N-HA-HYGROMYCIN lentiviral vectors that were generated by Vector Builder. Using the above lentiviral plasmid backbones, we enzymatically changed the tag from HA to RFP, V1 (non-fluorescent N-terminus of Venus, Met$^1$-Gln$^{157}$) and V2 (non-fluorescent C-terminus of Venus, Lys$^{158}$-Lys$^{238}$). V1 and V2 tags were amplified by PCR from V1- and V2- pcDNA3.1 (from Prof. Ivan Dikic). All cDNA mutations were generated via PCR site-directed mutagenesis according to standard protocols. All cDNAs and plasmids used in the manuscript are reported in Reagents and Tools.

## Immunoblotting

Cells were harvested by removing the culture medium and washed with PBS 1X. Cells were lysed in ice-cold lysis buffer (150 mM NaCl, 50 mM Tris-HCl pH 7.5, 1% Nonidet P-40 (NP-40)) supplemented with protease and phosphatase inhibitors and N-Ethylmaleimide (NEM) 1 mM. Insoluble cell components were separated at 15,500× *g* for 30 min at 4 °C. Protein denaturation was performed in Laemmli buffer at 95 °C for 10 min. Protein lysates were resolved in SDS-PAGE gels and transferred to nitrocellulose

or PVDF membrane. Membranes were saturated in TBS 0.1% Tween containing 5% low-fat milk and incubated overnight at 4 °C with the specific primary antibody reported in Reagents and Tools.

## Co-immunoprecipitation

Stable C2C12 cell lines were induced with doxycycline (1 μg/ml). Undifferentiated and differentiated C2C12 were harvested in 50 mM Tris/HCl (pH 8.0), 120 mM NaCl, 1% NP40, complete protease and phosphatase inhibitors and N-Ethylmaleimide (NEM). Lysates were cleared by centrifugation at 10,000×*g* for 10 min and incubated for 4 h at 4 °C with monoclonal anti-RFP-agarose, for 1 h with monoclonal anti-GFP-agarose and overnight with TUBE-2 agarose beads. Beads were then washed three times in wash buffer: 10 mM Tris/Cl pH 7.5, 150 mM NaCl, 0.05% NP-40, 0.5 mM EDTA, resuspended in Laemmli buffer and boiled at 95 °C. Supernatants were loaded on SDS-PAGE and western blots were performed using the indicated antibodies.

## Lysosomal-IP

C2C12 wild-type and C2C12 *Fam134b*^KO cells, both expressing TMEM192-HA, were seeded in 15-cm dishes. After 7 days, the plates were washed twice with PBS, and pellets were collected and centrifugated at 1000×*g* for 5 min at 4 °C. Supernatants were discarded, and pellets were washed once with 1 ml of cold Sub-fraction buffer: 140 mM KCL, 50 mM sucrose, 1 mM DTT, 2 mM EGTA, 2.5 mM MgCl2, 25 mM HEPES (pH 7.25) supplemented with NEM, protease and phosphatase inhibitor and centrifuged at 1000×*g* for 2 min at 4 °C. Supernatants were discarded while pellets were resuspended in 500 μl of complete Sub-fraction buffer. Samples were homogenized using a manual potter and centrifugated at 1000×*g* for 3 min at 4 °C and the supernatants were collected. Lysates were incubated with HA magnetic beads for 2 h at 4 °C with gentle rotation. Using a magnetic stand, beads were then washed twice with 1 ml of Sub-fraction buffer for 10 min at 4 °C, five times with 1 mL of Sub-fraction buffer for 3 min at room temperature and twice with 1 mL of 300 mM NaCl in Sub-fraction buffer for 3 min at room temperature. Samples were eluted adding Laemmli buffer, boiled at 95 °C for 10 min and processed for mass spectrometry.

## GST pull-down

LC3s, GABARAPs and Ub proteins were cloned into a pGEX-4T-1 vector, as GST fusion proteins, and expressed in *Escherichia coli* BL21 (DE3) cells grown in LB medium. Expression was induced by the addition of 0.5 mM IPTG. Cells were incubated at 37 °C for 5 h and subsequently harvested and lysed using sonication in a lysis buffer: 20 mM Tris-HCl pH 7.5, 10 mM EDTA, 5 mM EGTA, 150 mM NaCl. GST-fused proteins were immuno-precipitated using Glutathione Sepharose 4B beads. Fusion protein-bound beads were used directly in GST pull-down assays. HEK293T cells were transfected with an expression plasmid containing FAM134B2 cDNA using GeneJuice. After 24 h, cells were lysed in lysis buffer: 50 mM HEPES, pH 7.5, 150 mM NaCl, 1 mM EDTA, 1 mM EGTA, 1% Triton X-100, 10% glycerol supplemented with protease inhibitors and NEM. Lysates were cleared by centrifugation at 12,000×*g* for 10 min and incubated with GST fusion protein-loaded

beads overnight at 4 °C. Beads were then washed three times in lysis buffer, resuspended in Laemmli buffer and warmed at 95 °C for 5 min. Supernatants were loaded on SDS-PAGE.

Stable and inducible C2C12 were differentiated and HA-FAM134B expressed adding doxycycline. Samples were lysed in lysis buffer: 50 mM HEPES, pH 7.5, 150 mM NaCl, 1 mM EDTA, 1 mM EGTA, 1% Triton X-100, 10% glycerol supplemented with protease inhibitors and NEM. Lysates were cleared by centrifugation at 12,000×*g* for 10 min and incubated with TUBE2 and respective control beads overnight at 4 °C. Beads were then washed three times in lysis buffer, resuspended in Laemmli buffer and warmed at 95 °C for 5 min. Supernatants were loaded on SDS-PAGE.

## Cell culture

HEK293T and murine C2C12 cells were purchased from ATCC. Cells identities were authenticated by STR analysis. Human immortalized myoblasts and murine primary myoblasts were provided by Prof. Gargioli (University Tor Vergata, Rome, Italy). All cell lines were regularly tested for the presence of mycoplasma using LookOut Mycoplasma PCR Detection Kit. Cells were maintained at 37 °C with 5% $CO_2$ in 4.5 g/L glucose DMEM medium supplemented with 10% fetal bovine serum, 100 U/ml penicillin and streptomycin, and 10 mM of HEPES. C2C12 stable and inducible cells were treated with 1 μg/ml of doxycycline to induce FAM134B expression and with 200 ng/ml bafilomycin A1 to block autophagy. Inducible C2C12 cell lines were cultivated in DMEM supplemented with 10% tetracycline-screened fetal bovine serum. C2C12 myoblasts differentiation into myotubes was spontaneously induced by allowing cells to reach confluency. We seeded cells to reach their confluency after 2/3 days. At day 4, when cells started touching each other, myoblasts began to fuse together and form myotubes. Geltrex™ Matrix solution was used to perform the coating of the plates before myoblasts seeding. To avoid any possible perturbance in the autophagy flux, we kept C2C12 in standard DMEM, and media was changed every day. Murine and human-derived myoblasts were cultured in a standard growth medium (DMEM GlutaMAX, 10% FBS and 100 U/ml P/S). Spontaneous differentiation in mature myotubes was achieved by culturing confluent cells for 10 days with daily replacement of the growth medium. Alternatively, cells were differentiated in low serum medium (DMEM GlutaMAX, 2% horse serum).

For transient expression, plasmids were transfected with GeneJuice or Turbofect according to the manufacturer's instructions. Stable C2C12 cell lines were generated using lentiviral virus infections. Lentiviruses were generated by co-transfecting in HEK293T cells with the pLTD vector, pPAX2 packaging plasmid, and pMD2.G envelope plasmid. Viral supernatant was harvested after 48 h and filtered through a 0.45-μm filter. C2C12 ($5 \times 10^4$) cells were incubated in six-well plates with 1 ml of viral supernatant and after 48 h, the medium was replaced with antibiotic-containing medium.

## Purification and culture of muscle satellite cells (MuSCs)

Muscle Satellite Cells (MuSCs) were purified from 1.5-month-old C57BL/6J mice using the magnetic beads cell sorting MACS purification system. Briefly, freshly isolated mononuclear muscle

cells were resuspended in Magnetic Beads Buffer (MBB, 0.2% BSA, 2 mM EDTA in PBS 1×) and filtered through a 30-μm pre-separation filter to remove large particles from the single-cell suspension. The single-cell suspension was incubated in MBB with mouse anti-CD31 microbeads and mouse anti-CD45 microbeads antibodies. Cells were positively selected through the MS column depleting the CD31/CD45-positive cells from the cell suspension. Negative lineage (Lin-) cells were resuspended in MBB and incubated with mouse anti α7-integrin microbeads and positively selected as α7-integrin-positive Muscle Satellite Cells (MuSCs). Antibody labeling procedures using the microbeads conjugated antibodies were performed according to the manufacturer's instructions.

Freshly sorted MuSCs were pre-plated in pre-warmed Satellite Cell-Growth Medium (SC-GM) composed of 20% FBS, 10% donor horse serum, 2% chicken embryo extract, 10 mM HEPES, 1 mM sodium pyruvate, 100 U/ml P/S in high glucose DMEM GlutaMAX for 2 h to reduce fibroblasts contamination. MuSCs were seeded at a density of $1.5 \times 10^4$ cells/cm$^2$ in geltrex-coated wells or dishes. Myotube differentiation was induced either by culturing confluent cells (spontaneous differentiation) or by exposing MuSCs to differentiation medium (DMEM GlutaMAX, 10 mM HEPES, 1 mM sodium pyruvate, 2% horse serum).

## Generation of C2C12 Fam134b knockout cells

The Fam134b knockout C2C12 cell line was generated using a lentiviral CRISPR–Cas9 system. sgRNAs, reported in Reagents and Tools, were designed using the Broad Institute genetic perturbation platform (https://portals.broadinstitute.org/gpp/public/analysis-tools/sgrna-design). Oligonucleotides were cloned into the lentiviral vector pKLV2.2-h7SKgRNA5(SapI)-hU6gRNA5(BbsI)-EF1a-Cas9-FLAG-P2A-BleoR, kindly provided by Dr. Koraljka Husnjak and Prof. Ivan Dikic. Oligonucleotides for the gRNAs were annealed and phosphorylated using T4 polynucleotides kinase. To clone the first gRNA, the vector was digested with BsmBI enzyme, while to clone the second gRNA, the vector was digested with SapI. The ligase reaction was achieved using the T4 ligase. Following the lentiviral transduction, cells were selected with 20 μg/ml Blasticidin and 1 mg/ml of Neomycin for 2 weeks.

## Antibodies used for western blot, immunoprecipitation, and immunofluorescence staining

A complete list of primary and secondary antibodies used for the experiments is reported in Reagents and Tools.

## Fluorescence protein protection (FPP) assay

Stable and inducible C2C12 cells expressing HA-FAM134B1 or HA-FAM134B2 were grown on coverslips, induced as described previously, and fixed with 4% PFA for 10 min, washed three times with buffer A (20 mM PIPES pH 6.8, 137 mM NaCl and 2.7 mM KCl) and permeabilized with 20 μM digitonin (Calbiochem) diluted in buffer A for 5 min at room temperature. Coverslips were blocked for 30 min with blocking solution (5% FBS (v/v) and 50 mM NH4Cl in buffer A) without any permeabilizing agent and incubated with primary anti-HA or anti-TGN38 antibodies diluted in blocking

solution. Digitonin treatment permeabilizes only the plasma membrane. This condition allows the recognition only of cytosolic exposed epitopes, but not luminal ones. The TGN38 antibody recognized specifically the luminal portion of the protein, that is not accessible after digitonin permeabilization only. This represents a control that the plasma membrane has been permeabilized exclusively. Coverslips were washed with buffer A and incubated with fluorochrome-conjugated secondary antibodies (Alexa Fluor 568 for TGN38 in buffer A) for 1 h at room temperature. After incubation, cells were fixed with 2% PFA for 5 min and washed once with 50 mM NH4Cl in PBS. Coverslips were subsequently permeabilized with 0.1% Triton-X-100 in PBS for 5 min. Cells were then blocked with blocking solution (0.05% saponin, 0.5% BSA and 50 mM NH4Cl in PBS) and incubated with the same primary antibodies used in the first step. Coverslips were then washed with PBS and incubated with fluorochrome-conjugated secondary antibodies (Alexa Fluor 488 for TGN46 in PBS) for 1 h at room temperature. The TGN38 epitope becomes accessible to the primary antibody under these conditions, confirming selective permeability and identifying luminal epitopes. Indeed, N-terminally tagged FAM134B1 and FAM134B2 turned to be readily detectable by antibodies after plasma membrane permeabilization only.

## Fluorescence microscopy

C2C12 cells were plated and differentiated on glass coverslips and fixed with 4% paraformaldehyde for 10 min. Cells were permeabilized with Triton or Digitonin solution in PBS at room temperature for 5 min and then blocked in PBS containing 10% fetal bovine serum (FBS) for an additional hour at room temperature. Cells were incubated overnight at 4 °C with primary antibody diluted in PBS with the same detergent used during permeabilization. Washes were performed in PBS with the same detergent used during permeabilization. Cells were incubated with secondary antibodies for 40 min at room temperature and then washed 3 times. Nuclei were stained with Hoechst 33342. The coverslips were mounted on slides with an aqueous mounting medium (Mowiol) and placed on a glass holder. Images were acquired with a Zeiss LSM800 laser-scanning microscope.

For high-resolution microscopy, cells were imaged using a Plan-Apochromat 63×/1.4 oil objective on a Zeiss LSM880 confocal system equipped with an AiryScan module and controlled by the Zen blue software. The images used for phenotype quantification were processed with Fiji (ImageJ; National Institute of Health (NIH)) software. Brightness and contrast were adjusted with Adobe Photoshop.

## Electron microscopy

For electron microscopy (EM), undifferentiated and differentiated C2C12 cells were fixed with 1% GA prepared in 0.2 M HEPES for 30 min, then scraped, pelleted, post-fixed in $OsO_4$ and uranyl acetate, dehydrated in ethanol and embedded in Epon. From each sample, thin 60-nm sections were cut using a Leica EM UC7 (Leica Microsystems, Wetzlar, Germany). EM images were acquired from thin sections using a FEI Tecnai-12 electron microscope (FEI, Eindhoven, Netherlands) equipped with a VELETTA CCD digital camera (Soft Imaging Systems GmbH, Munster, Germany).

## Sample preparation by ultramicrotomy and electron tomography

Serial ultramicrotomy was performed on the samples using a Leica UC7 ultramicrotome. 150 nm sections were decompressed with chloroform and collected onto formvar carbon slot grids (Prod No. 01805-F, Ted Pella). Imaging was performed using Thermo Scientific Spectra 300 kV transmission electron microscope equipped with a Thermo Scientific Ceta-S camera. Large montages of serial sections were collected using Thermo Scientific MAPS software to identify regions of interest for tomography. The target areas were then transferred to SerialEM software for automated tilt series acquisition. At each position, a 2-by-2 montage tilt series was acquired at nominal pixel size of 0.5 nm per pixel. We collected 121 tilt images from −60 to 60 degrees per tilt series. These tilt-series were processed using the etomo pipeline. When applicable tomograms from serial sections were stacked using etomo to generate serial tomograms.

## Mass spectrometry

All the experiments were performed in a labeling-free setting. Proteins were precipitated in acetone and then reduced and alkylated in a solution of 6 M Guanidine-HCl, 5 mM TCEP, and 20 mM chloroacetamide. Peptides were obtained by digesting proteins with LysC (WAKO) for 3 h at 37 °C and with sequencing-grade endopeptidase Trypsin (Promega) overnight at 37 °C. Collected peptide mixtures were concentrated and desalted using the Stop and Go Extraction (STAGE) technique. For Ubiquitin and Phosphorylation sites detection, after the IP, GFP-trap beads were resuspended in SDC-buffer (2% SDC, 1 mM TCEP, 4 mM CAA, 50 mM Tris pH 8.5) and heated at 60 °C for 30 min. Peptides were obtained by Trypsin/LysC digestion and cleaned up using SDB-RPS stage tips.

Instruments for LC MS/MS analysis consisted of a NanoLC 1200 coupled via a nano-electrospray ionization source to the quadrupole-based Q Exactive HF benchtop mass spectrometer. Peptide separation was carried out according to their hydrophobicity on a homemade chromatographic column, 75 μm ID, 8 μm tip, 350 mm bed packed with Reprosil-PUR, C18-AQ, 1.9 μm particle size, 120 Angstrom pore size, using a binary buffer system consisting of solution A: 0.1% formic acid and B: 80% acetonitrile, 0.1% formic acid.

For DDA acquisition, runs of 120 min runs were used for proteome samples, with a constant flow rate of 300 nl/min. After sample loading, the run started at 5% buffer B for 5 min, followed by a series of linear gradients, from 5% to 30% buffer B in 90 min, then a 10 min step to reach 50% and a 5 min step to reach 95%. This final step was maintained for 10 min.

Q Exactive HF settings: MS spectra were acquired using 3E6 as an AGC target, a maximal injection time of 20 ms and a 120,000 resolution at 200 *m/z*. The mass spectrometer operated in a data-dependent Top20 mode with subsequent acquisition of higher-energy collisional dissociation (HCD) fragmentation MS/MS spectra of the top 20 most intense peaks. Resolution for MS/MS spectra was set to 15,000 at 200 *m/z*, AGC target to 1E5, max injection time to 20 ms and the isolation window to 1.6Th. The intensity threshold was set at 2.0 E4 and Dynamic exclusion at 30 seconds.

For DIA acquisition, runs of 60 min were used for proteome samples, with a constant flow rate of 300 nl/min. After sample loading, the run followed a series of linear gradients, from 7% to 32% buffer B in 45 min, then 5 min and 30 s to reach 45% and 3 min and 30 s to reach 95%. This final step was maintained for 3 min and 30 s.

Q Exactive HF settings (DIA): MS spectra were acquired using 32 variable windows covering a mass range of 300–1650 *m/z*. The resolution was set to 60,000 for $MS^1$ and 30,000 for $MS^2$. The AGC was 3E6 in both $MS^1$ and $MS^2$, with a maximum injection time of 60 ms in $MS^1$ and 54 ms in $MS^2$. NCE were set to 25%, 27.5%, 30%.

## BiCAP interactome analysis

We generated stable and inducible (via doxycycline) C2C12 cells expressing the following combinations of lentiviral plasmids carrying a Split GFP tag: V1-FAM134B1/V2-FAM134B1; V1-FAM134B2/V2-FAM134B2; V1-FAM134B1/V2-FAM134B2. We differentiated C2C12 into myotubes, induced FAM134B expression for 8 h and then lysed samples (150 mM NaCl, 50 mM Tris-HCl pH 7.5, 1% Nonidet P-40 supplemented with protease and phosphatase inhibitors and N-Ethylmaleimide (NEM) 1 mM). If the two V1 and V2 tags are in proximity they will associate to re-establish the full GFP molecule. We incubated 1 mg of sample lysate with GFP-trap beads (Chromoteck) for 1 h on rotating wheel at 4 °C. Beads were then washed three times with cold lysis buffer and one time in PBS before on-beads trypsin digestion. Beads were than incubates with 25 μl SDC buffer (2% sodium deoxycholate, 1 mM TCEP, 4 mM chloroacetamide and 50 mM Tris-HCl pH 8.5) and heated at 60 °C for 30 min. We then digested samples with trypsin overnight at 37 °C (500 ng trypsin and 500 ng LysC in 25 μl 50 mM Tris-HCl pH 8.5). Digestion was blocked with 150 ml of 1%TFA in isopropanol. Peptides were purified using SDB-RPS stage tips (Empore). We washed peptides with 1% TFA in isopropanol and with 0.2% TFA in water. Peptides were eluted with 80% acetonitrile plus 1.25% ammonia. Peptides were dried and processed for the LC-MS/MS.

## mRNA sequencing

For gene expression analyses, cells and myotubes were grown in DMEM with 10%FBS. Total RNA was extracted from C2C12 myoblasts and differentiated myotubes using RNeasy Mini Kit according to the manufacturer's instructions. Total RNA was quantified using the Qubit 4.0 fluorimetric Assay (Thermo Fisher Scientific). Total RNA was extracted from $n = 4$ for each experimental group. RNA was quantified and diluted to 10 ng/μL. Libraries were prepared from 125 ng of total RNA using the NEGEDIA Digital mRNA-seq research grade sequencing service (Next Generation Diagnostic srl) which included library preparation, quality assessment and sequencing on a NovaSeq 6000 sequencing system using a single-end, 100 cycle strategy (Illumina Inc.). Raw data were analyzed by Next Generation Diagnostics srl proprietary NEGEDIA Digital mRNA-seq pipeline (v2.0) which involves a cleaning step by quality filtering and trimming, alignment to the reference genome and counting by gene (BBMap–Bushnell B.—sourceforge.net/projects/bbmap/). The reads were trimmed to remove adapter sequences and low-quality ends, and reads mapping to contaminating sequences (e.g., ribosomal RNA, phiX control) were filtered out. Illumina NovaSeq

6000 base call (BCL) files were converted into fastq files through bcl2fastq. Trimming and cleaning was performed with bbduk. Alignment was performed with STAR 2.6.0a. The expression levels of genes were determined with HTseq-counts 0.9.1.

## Quantitative real-time PCR (qRT–PCR)

Total RNA extraction was performed using the RNeasy Mini Kit and the concentration and purity of RNA were measured using a spectrophotometer at 260/280 nm. First-strand cDNA for qRT–PCR was generated using QuantiTect Reverse Transcription Kit. The primers were designed using online primer design software and are shown in Reagents and Tools. GAPDH was used as a housekeeping gene, and the data normalization was performed according to the 2-ΔΔCt method. The qRT–PCR experiments were performed using SYBR Green I Mastermix (Roche) and a Light-cycler96 instrument following the manufacturer's instructions.

## Dual-luciferase reporter assay

HEK293T cells ($10^4$) were plated in 96-well plates for 24 h and were then co-transfected with firefly luciferase reporter plasmid containing 1.5-kb mouse FAM134B-2 promoter (pGL4), Renilla plasmid (pRL-TK) and different transcription factors cloned into pCMV6-entry all listed in Reagents and Tools. The co-transfection was prepared in Opti-mem, with Turbofect as the transfection reagent. After 48 h of transfection, luciferase activity was assessed with the Dual-luciferase reporter assay following the manufacturer's instructions. Firefly luciferase activity was then normalized against Renilla luciferase activity.

## ER-phagy and macro-autophagy assay

C2C12 cells were infected with a retrovirus vector carrying GFP-LC3B-RFP or a lentivirus vector carrying ssRFP-GFP-KDEL obtained from the pCW57-CMV-ssRFP-GFP-KDEL. Fluorescence of GFP and RFP fluorescence as well as cell confluence and differentiation status were monitored over time via the IncuCyte S3 (Sartorius, Germany) in 12 and 96-well format. 22800 cells/cm² (C2C12) were seeded in DMEM media, supplemented with 10% Fetal Bovine Serum (FBS) and 1% penicillin/streptomycin 150 μg/ml Hygromycin, and cultivated for 10 days. The medium was changed daily. Screens in the two channels were taken at indicated time points following the treatment. Macro-autophagy and ER-phagy flux were monitored through changes in the ratio of the total fluorescence intensity of RFP/GFP. Each point represents the averaged ratio of data obtained from two or three individual wells (technical replicates) and experiments have been performed in four biological replicates with comparable results. For GFP-LC3B-RFP construct, where the RFP is cleaved upon LC3B lipidation and not degraded into lysosome, we used the RFP as normalizer to have a cell number independent readout.

## FAM134B2 sequence and structure

The sequence of FAM134B2, a short isoform of FAM134B1, was obtained and annotated from UniProt (Accession code: Q9H6L5-2), aligned with FAM134B1 using AlignME and visualized with Jalview. A 3D structure of this region of the RHD region was modeled using AlphaFold2. For comparison, a previously modeled intact FAM134B1-RHD structure was used as a template (Bhaskara et al, 2019).

## Coarse-grained MD simulations of FAM134B2

A coarse-grained (CG) model of FAM134B2 was prepared using the MARTINI force-field v2.2 using the *martinize.py* script. Secondary structure restraints were imposed with DSSP assignments. The protein insertion depth and orientation with respect to the model bilayer were obtained using the OPM database and the PPM web server. For MD simulations in flat bilayers, the CG protein model of the intramembrane region of FAM134B2 was embedded in a POPC (16:0-18:1 PC) bilayer spanning a periodic box of $20 \times 20$ nm². This system was then solvated it with CG water containing 150 mM NaCl using CHARMM-GUI. This system was energy minimized using the steepest descent algorithm for 3000 steps and then equilibrated the system with position restraints on the backbone (BB) beads, first under NVT conditions (T = 310 K) and then under NPT (P = 1 bar) conditions. Five short equilibrations, with increasing time steps ($dt = [1, 2, 5, 10, 20]$ fs), were employed following the CHARMM-GUI protocol. Subsequently, we performed production runs for 10 μs for each system using a 20 fs timestep under NPT conditions (1 bar; 310 K). The Berendsen thermostat and the Berendsen semi-isotropic barostat were used for the equilibration phases and the velocity-rescale thermostat, while the Parinello-Rahman semi-isotropic barostat was employed for the production phase, respectively. All CGMD simulations were performed using GROMACS v2020.1. For comparison, initial CG models of FAM134B-RHD in POPC bilayers were obtained from previous work and simulated using the same above-mentioned protocol.

## Analysis of simulation data

Conformational dynamics and flexibility of FAM134B or FAM134B2 proteins were quantified by measuring the root mean square fluctuations (RMSF) of the main-chain residue positions from the mean structure using the *gmx rmsf* program implemented in GROMACS. The protein structures ($n = 5000$) sampled evenly at 2 ns intervals from the 10 μs MD simulations, were clustered using the gromos methods with an RMSD cut-off of 0.5 nm. Most populated clusters obtained for FAM134B1 and FAM134B2 were then analyzed for key structural features (e.g., inter-hairpin orientation).

## Bicelle-to-vesicle transition simulations

Discontinuous membrane simulations were utilized to study active membrane curvature induction. A well-equilibrated CG structure of FAM134B2 was embedded in pre-equilibrated flat circular bicelles made from DMPC (14:0-14:0 PC) and DHPC (7:0–7:0 PC) lipids. This assembly was then solvated with CG water using *insane.py*. This system was then equilibrated under NPT conditions ($P = 1$ bar; $T = 310$ K) with additional position restraints on protein backbone (BB) beads. Fifty replicates of production runs (1μs each) were initiated with different starting velocities after releasing position restraints. For comparison, we resampled 50 bicelle simulations from a total of 100 previous runs using the same protocol.

Bicelle-to-vesicle transitions were analyzed by computing signed mean curvatures of the bicelles along the individual trajectories ($H(t)$) using MemCurv. The waiting times ($t$) for bicelle closure (bilayer curvature $|H| \geq 0.15$ nm$^{-1}$) were estimated from the curvature time series of 50 independent replicates. The kinetics for bicelle-to-vesicle transition was modeled as a Poisson process with time ($t' = 1/k'$) and a rate ($k'$), and a lag time ($\tau$), where the total waiting time for vesicle formation is $t = t' + \tau$. The constant lag time $\tau$ captures the time required for vesicle closure from the curved bilayer disc. The distributions of waiting times are thus $p(t) = k'e^{-k'(t-\tau)}$ for $t > \tau$. The rate of vesicle formation was determined ($k = 1/(t' + \tau)$) by fitting the cumulative distribution function for the probability density, $P(t - \tau) = ke^{-k(t-\tau)}$ corresponding to $p(t)$, to the observed waiting time distributions estimated from 50 replicates.

## Data analysis

All experiments were performed in at least three independent biological replicates. Data are presented as mean ± s.e.m or s.d. Statistical analysis between two experimental groups was performed using a parametric two-tailed Student's *t* test or ANOVA. For immunostaining quantifications, cells were analyzed for each experimental condition in each replicate. To analyze the ER morphology shown in Fig. 7F, we analyzed images from Z-projection. KDEL channel was used for the quantification. Images were thresholded similarly to generate binary images that were subsequently measured using the Analyze particle function of the Fiji software. The Feret's diameter and perimeter data for each individual ER profiles were considered. The analyses were performed on 15–20 cells (T0) or myotubes (T10) for the indicated conditions. For autophagy flux experiments, data obtained from one scan containing >1000 cells of three different wells with indicated treatment were analyzed. For western blot quantifications, densitometric analysis were performed considering three independent blot images.

Software and Platforms employed for data analysis are all listed, together with their references, in Reagents and Tools.

For the DDA acquisition modality, all acquired RAW files were processed using MaxQuant (1.6.2.10) and the implemented Andromeda search engine, determining both iBAQ and LFQ values. For protein assignment, spectra were searched against the UniProt Homo Sapiens (UP000005640) and Mus Musculus (UP000000589) entries. Searches were performed with tryptic specifications and default settings for mass tolerances for MS and MS/MS spectra. Carbamidomethyl at cysteine residues was set as a fixed modification, while oxidations at methionine and acetylation at the N-terminus were defined as variable modifications. The minimal peptide length was set to seven amino acids, and the false discovery rate for proteins and peptide-spectrum matches to 1%. The match-between-run feature with a time window of 0.7 min was used.

For DIA acquisition mode, RAW files were analyzed using DIA-NN 1.8 and Spectronaut 18.3.23, ran in library-free search (direct-DIA) To assess protein identification and quantification, default parameters were used: precursor ion generation: deep learning-based spectra, with RTs, and IMs prediction enable; precursor range: 300–1800 *m/z*; fragment ion range to 200–1800 *m/z* (DIA-NN search); precursor ion mass tolerance: dynamic; Orbitrap fragment ion mass tolerance: dynamic (Spectronaut); fixed

modifications: cysteine carbamidomethylation; variable modifications: methionine oxidation, protein N-terminus acetylation, enzyme: trypsin, Lys-C; the number of allowed missed cleavage sites: 2; minimum peptide length: seven amino acid residues (both DIA-NN and Spectronaut). The DIA spectra were probed against UniProt *Homo Sapiens* (UP000005640) and *Mus Musculus* (UP000000589) FASTA database.

Both acquired DDA and DIA MS data were filtered at 1% FDR on the protein level and exported, followed by analysis with different software packages (Excel, Rstudio, SRplot).

Specifically, statistical analyses for quantitative evaluation were performed using Perseus software (1.6.2.3), the subtraction of contaminants, reverse entries as well as the identification by a modified peptide only, were applied as the first filter. The LFQ and iBAQ ratios were logarithmized, grouped, and filtered for min. valid number (min. 3 or 2 in at least one group). Missing values were replaced by random numbers that are drawn from a normal distribution. For the comparison among the different conditions in terms of protein abundance level, Anova and Student's *t* test were used. For each test, the level of significance was set equal to 0.05. Proteins with at least 0.5-fold or onefold changes were considered differentially regulated. A Gene Ontology (GO) enrichment analysis was carried out on the dysregulated proteins, with cellular components and biological processes set as the main terms of interest. The hierarchical clustering was performed using Fisher's exact test to calculate significant GO-enriched terms (FDR < 0.1).

For mRNA sequencing, the threshold for statistical significance chosen was FDR < 0.05. Gene Ontology (GOEA) and Functional Annotation Clustering analyses were performed using DAVID Bioinformatic Resources restricting the output to Cellular Component terms (CC_FAT). The threshold for statistical significance of GOEA was FDR < 0.1 and Enrichment Score >1.5.

## Data availability

The mass spectrometry proteomic data have been deposited to the ProteomeXchange Consortium via the PRIDE partner repository with the dataset identifiers: PXD050867, PXD050893, PXD050898, PXD050899, PXD050902, PXD053358, PXD053361, PXD050874, PXD050903, PXD050907. Transcriptome data that support the findings of this study have been deposited in the Gene Expression Omnibus (GEO) under accession code GSE254885. The Series GSE254885 has been named: Transcriptome profile of C2C12 wild-type and Fam134b$^{KO}$ myoblasts and differentiated myotubes. Source data and quantifications given in the main text have associated raw data. Electron tomography data have been deposited in EMPIAR (ID 47486388).

The source data of this paper are collected in the following database record: biostudies:S-SCDT-10_1038-S44318-024-00356-2.

## Peer review information

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

## Acknowledgements

The authors thank Prof. Ivan Dikic (IBC2, Frankfurt am Main, Germany) for critical comments. Dr. Koraljka Husnjak (IBC2, Frankfurt am Main, Germany) for CRISPR-Cas9 lentiviral plasmids. The authors acknowledge Anna Vainshtein (Craft Science Inc.) for manuscript editing. The authors thank Concetta Bianco, Maria Miriam Amoroso and Sergio Alejandro Poveda Cuevas for their help. The authors thank the microscopy, mass spectrometry, and bioinformatic cores at TIGEM Institute. The authors thank the Center for Supercomputing and Goethe University of Frankfurt am Main for computing time on the Goethe-HLR cluster. The authors thank the Cryo-Electron Microscopy Unit of the National Facility for Structural Biology. This work was supported by grants to PG: Fondazione Telethon (TMPGMFU22TT), MDA (Idea Grant Program 22), AIRC (MFAG 2020), PNRR (PNRR-MR1-2022-12376821), PRIN-2022-PNRR (P2022JLNZ7), PRIN (20224FL9T5). RMB: CRC project on Selective Autophagy, DFG Project-ID 259130777-SFB1177. AS: Deutsche Forschungsgemeinschaft (DFG, German Research Foundation)—grant numbers 259130777 (SFB1177), the Dr. Rolf M Schwiete Stiftung (13/2017), the EU/EFPIA/OICR/McGill/KTH/Diamond Innovative Medicines Initiative 2 Joint Undertaking (EUbOPEN grant no. 875510). AR: AFM-Téléthon, grant number 23551, Fondazione Umberto Veronesi. KL was supported by Prof. Ivan Dikic's ERC (ER-REMODEL) and CRC project on Selective Autophagy, DFG Project-ID 259130777-SFB1177.

## Author contributions

**Viviana Buonomo**: Formal analysis; Investigation; Methodology. **Kateryna Lohachova**: Formal analysis; Investigation; Methodology. **Alessio Reggio**: Data curation; Investigation; Methodology. **Sara Cano-Franco**: Formal analysis; Investigation; Methodology. **Michele Cillo**: Formal analysis; Investigation; Methodology. **Lucia Santorelli**: Data curation; Software; Visualization. **Rossella Venditti**: Formal analysis; Investigation; Methodology. **Elena Polishchuk**: Formal analysis; Investigation; Methodology. **Ivana Peluso**: Methodology. **Lorene Brunello**: Formal analysis; Investigation; Methodology. **Carmine Cirillo**: Data curation. **Sara Petrosino**: Methodology. **Malan Silva**: Formal analysis; Methodology. **Rossella De Cegli**: Data curation. **Sabrina Di Bartolomeo**: Writing—original draft; Writing—review and editing. **Cesare Gargioli**: Resources. **Paolo Swuec**: Methodology. **Mirko Cortese**: Data curation; Visualization; Methodology. **Alexandra Stolz**: Data curation; Writing—original draft; Writing—review and editing. **Ramachandra M Bhaskara**: Methodology; Writing—original draft; Writing—review and editing. **Paolo Grumati**: Conceptualization; Data curation; Supervision; Funding acquisition; Investigation; Writing—original draft; Project administration; Writing—review and editing.

Source data underlying figure panels in this paper may have individual authorship assigned. Where available, figure panel/source data authorship is listed in the following database record: biostudies:S-SCDT-10_1038-S44318-024-00356-2.

## Disclosure and competing interests statement

The authors declare no competing interests.

# Expanded View Figures

**Figure EV1.  Proteome profile of muscle cells during myogenesis.**

(A) Representative WB of total cell lysate from C2C12 cells and myotubes. MF20: Myosin heavy chain; MyoG: Myogenin. Bar plots represent the densitometric quantification of WB bands. (B–D) Heatmap (Log$_2$ LFQ normalized intensity) of proteins related to the ER (B), lysosome (C) and autophagy-ubiquitin pathway (D) that are significantly deregulated in differentiated myotubes versus C2C12 myoblasts. (E, F) Representative WB of ER and lysosomal proteins from total cell lysate from myoblasts and myotubes. (G) Heatmap (Log$_2$ LFQ normalized intensity) of significantly regulated proteins in murine primary myoblasts and myotubes differentiated in growth medium (GM) or in differentiation medium (DM). The profile plot highlights significantly upregulated proteins. H) Principal components analysis (PCA) of the full proteomes from primary murine myoblasts and myotubes differentiated in growth medium (GM) or in differentiation medium (DM). (I) GOCC enrichment analysis from MS data with ER elements in purple and lysosomal components in green. (J) GOCC terms and frequencies of the upregulated proteins identified in murine myotubes. (K) Heatmap (LFQ normalized intensity) of significantly deregulated proteins in human myoblasts and myotubes differentiated in growth medium (GM) or in differentiation medium (DM). The profile plot highlights significantly upregulated proteins. (L) Principal components analysis (PCA) of the full proteomes from primary human myoblasts and myotubes differentiated in growth medium (GM) or in differentiation medium (DM). (M) GOCC enrichment analysis from MS data with ER elements in purple and lysosomal components in green. (N) GOCC terms and frequencies of the upregulated proteins identified in human myotubes. (O) The bar graph represents the quantification of the ER area in myoblasts and myotubes (Fig. 1A). All data are represented as mean ± s.d.; ***$P < 0.001$ (Student $t$ test). Mass spectrometry was performed in triplicate. Western blots were performed in three independent experiments. Source data are available online for this figure.

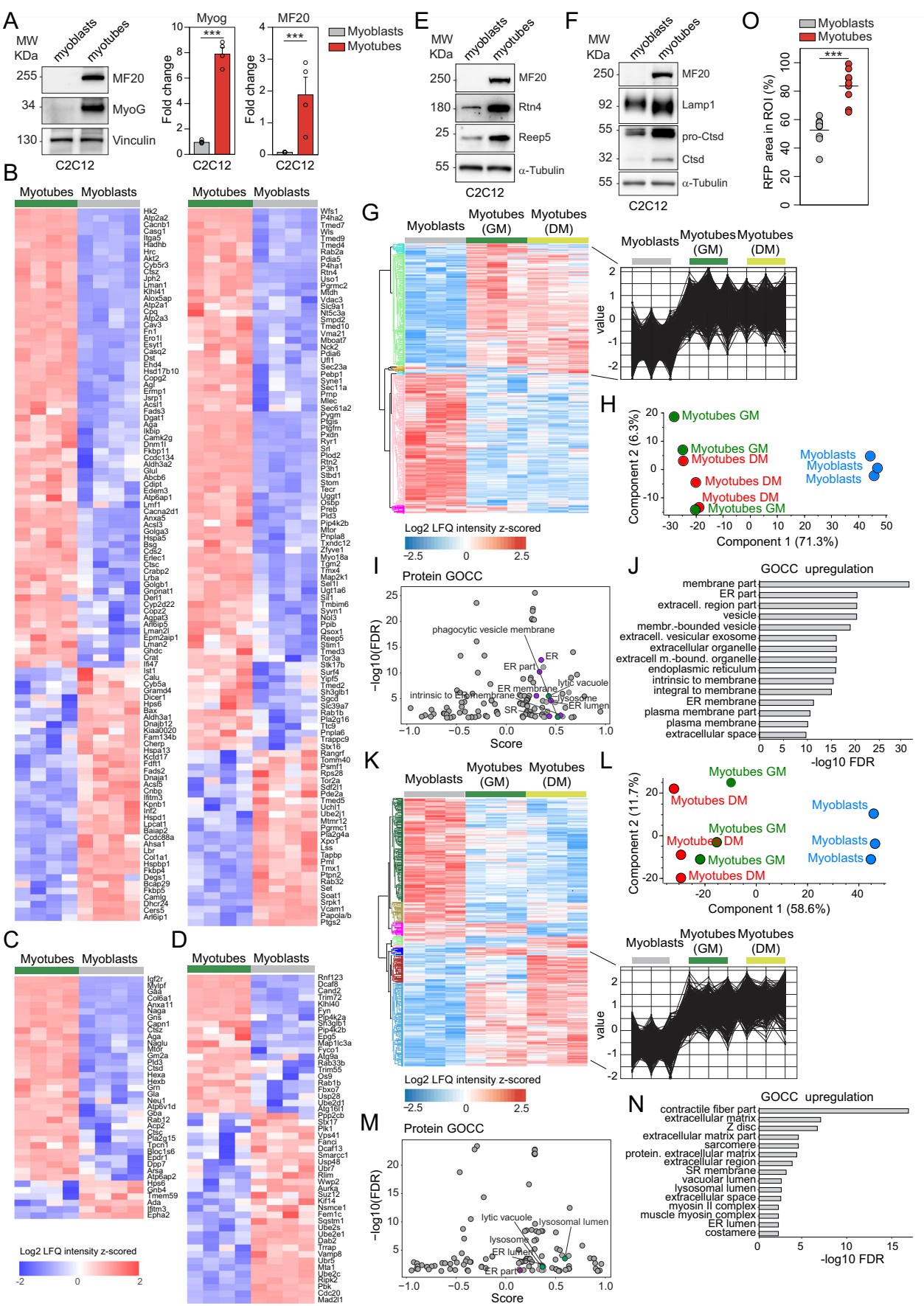

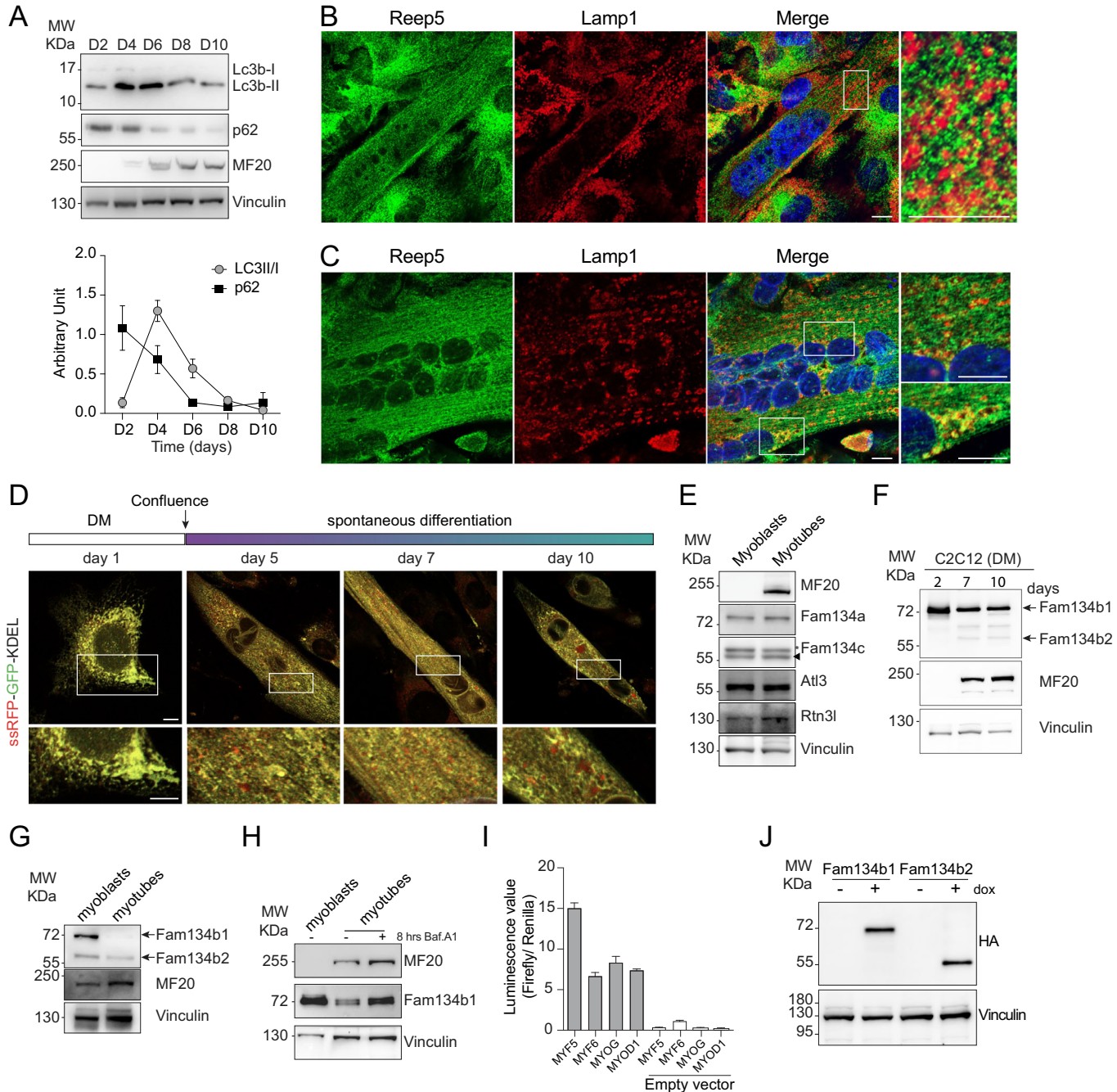

**Figure EV2. Fam134b isoforms are differently modulated during myogenesis.**

(A) WB analysis showing Lc3b, p62, and MF20 protein levels at various differentiation time points during C2C12 differentiation. Graphs represent the densitometric analysis of WB bands. (B) Confocal images of differentiated C2C12 cells stably expressing TMEM192-3xHA stained for HA and the ER protein REEP5. Scale bar: 10 μm. (C) Confocal images of differentiated human myoblasts, stained for the ER protein REEP5 and the lysosomal protein LAMP1. Scale bar: 10 μm. (D) Confocal images of C2C12 cells, stably overexpressing ssRFP-GFP-KDEL during their differentiation. Scale bar: 10 μm. Inset Scale bar: 5 μm. DM: differentiation medium (2% horse serum) (E). (E) WB analysis of the endogenous ER-phagy receptors in C2C12 myoblasts and myotubes. (F, G) WB analysis of endogenous FAM134B1 and FAM134B2 in C2C12 cells (F) and human myoblasts (G) differentiated into myotubes using differentiation medium (DM). (H) WB analysis of endogenous Fam134b1 in C2C12 myoblasts and myotubes, with and without Bafilomycin A1 (Baf.A1) treatment. MF20 was used as a muscle differentiation marker. (I) Graphical quantification of luciferase assays demonstrating the activity of the indicated transcription factors co-transfected, in HEK 293T), with a plasmid containing the firefly luciferase gene downstream of the Fam134b1 promoter. (J) WB representing the protein levels of HA-Fam134b1 and HA-Fam134b2 in C2C12 stable cell lines. Imaging, western blots and luciferase assay were performed in three independent experiments. Source data are available online for this figure.

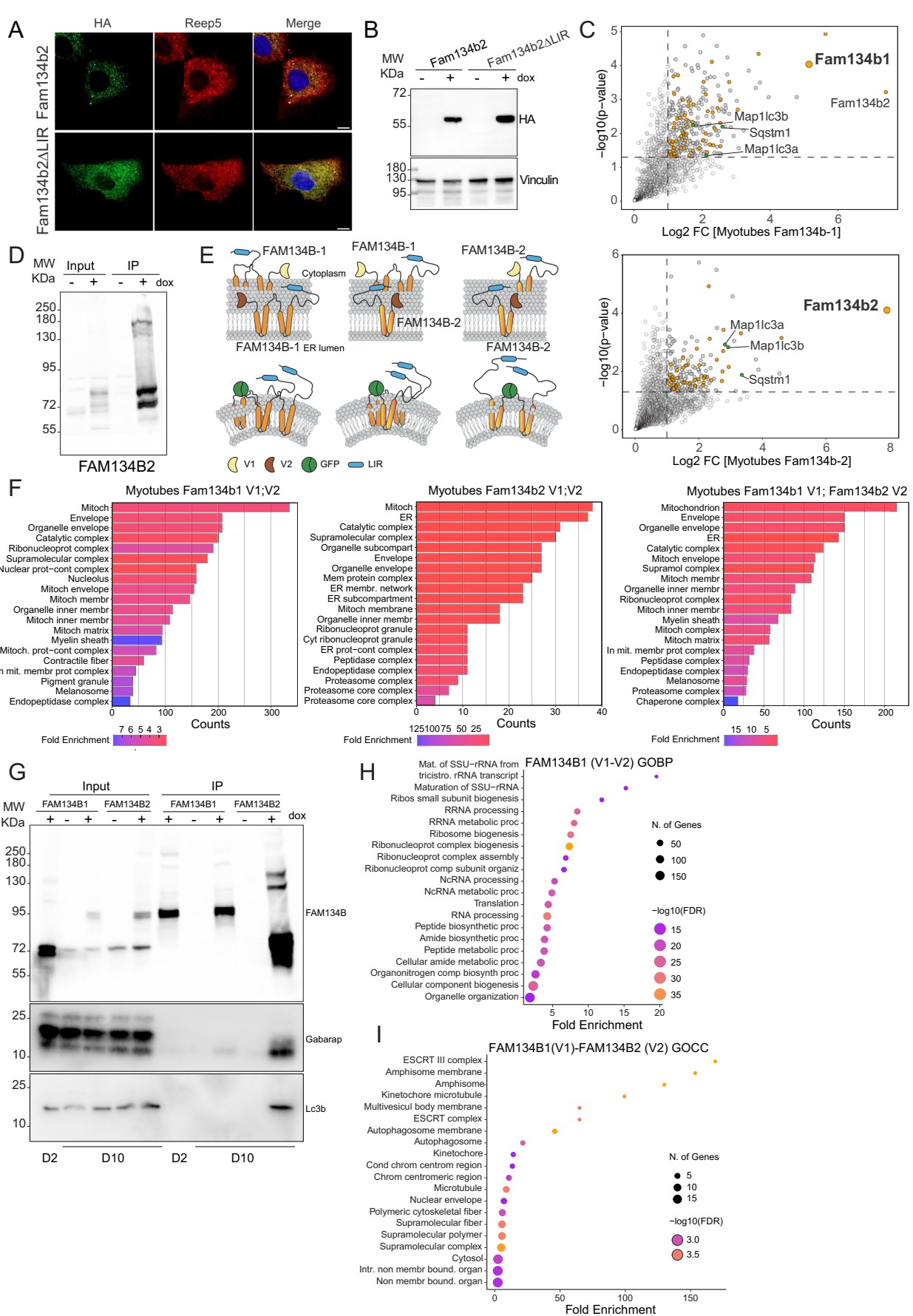

**Figure EV3.  FAM134B2 mostly interacts with ER and autophagy proteins.**

(**A**) Confocal images of C2C12 cells expressing HA-Fam134b2 and HA-Fam134b2ΔLIR, stained for the ER protein REEP5 and HA. Scale bar: 10 μm. (**B**) WB analysis of HA tag in C2C12 cells expressing HA-Fam134b2 and HA-Fam134b2ΔLIR. (**C**) Volcano plot illustrating the interactome of Fam134b1 (left) and Fam134b2 (right) in differentiated myotubes, highlighting ER proteins (orange dots) and the autophagy proteins (green dots). (**D**) WB analysis following RFP-tagged Fam134b2 pull-down in differentiated myotubes. (**E**) Schematic representation of the bimolecular complementation affinity purification system employed to pulldown FAM134B complexes. (**F**) GOCC terms and frequencies of the different FAM134B complexes interactors in myotubes. (**G**) IP of GFP tag in C2C12 cells expressing V1/V2-FAM134B1 or V1/V2-FAM134B2 and WB for Lc3b and Gabarap. (**H**) GOBP enrichment analysis from FAM134B1 homodimers interactors. (**I**) GOCC enrichment analysis from FAM134B1-FAM134B2 heterodimers interactors. Imagine, western blots and mass spectrometry were performed in three independent experiments. Source data are available online for this figure.

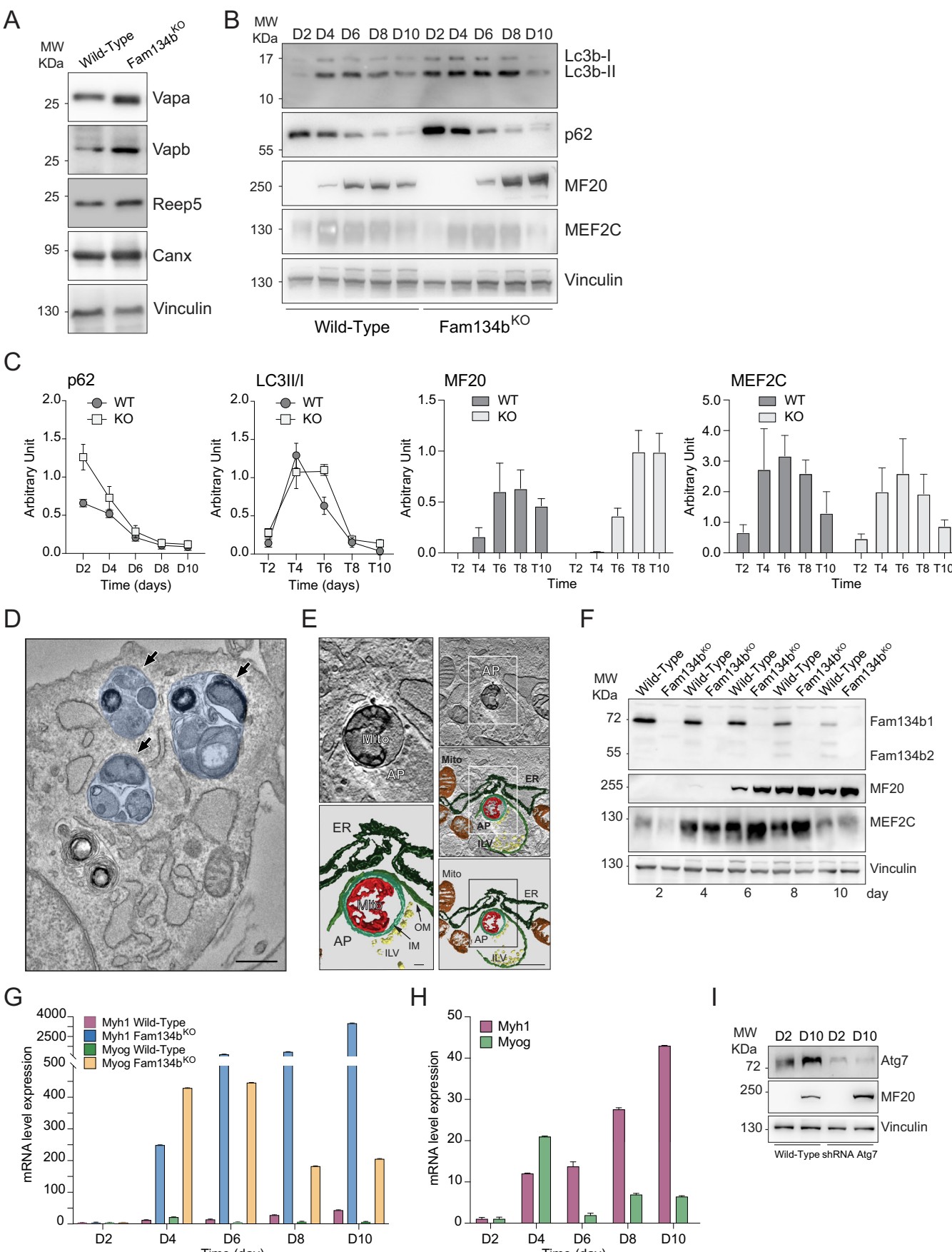

◀ **Figure EV4. A proper ER-phagy flux is necessary for myogenesis.**

(A) Representative WB analysis of endogenous ER proteins in WT and Fam134b[KO] myotubes. (B) Representative WB analysis of Lc3b, p62, MF20 and MEF2C in WT and Fam134b[KO] C2C12 cells throughout differentiation. (C) Densitometric analysis of WB bands in panel (B). (D) Electron microscopy images of Fam134b[KO] myotubes, with black arrows indicating autophagosome structures. Scale bar: 400 nm. (E) Tomography and imagine reconstruction depicting a mitochondrion inside an autophagosome in a Fam134b[KO] myotube. AP autophagosome, ER endoplasmic reticulum, Mito mitochondria, OM outer autophagosomal membrane (dark green), IM inner autophagosomal membranes (light green), ILV intra luminal vesicles. Red: mitochondria inside the autophagosome. Scale bar: 200 nm. (F) Representative WB analysis of endogenous Fam134b isoforms and the muscle differentiation markers MF20 and MEF2C in WT and Fam134b[KO] C2C12 cells throughout differentiation. (G) Real time PCR analysis illustrating the expression profiles of the indicated genes during the differentiation of WT and Fam134b[KO] C2C12 cells. (H) Magnified view of the bar graphs presented in panel G, focusing on WT samples. (I) Representative WB imagines of endogenous Atg7 and MF20 protein levels in WT and Atg7[KD] C2C12 cells during differentiation. Data are represented as mean ± s.d. Western blots, imaging and real time pcr were performed in three independent experiments. Source data are available online for this figure.

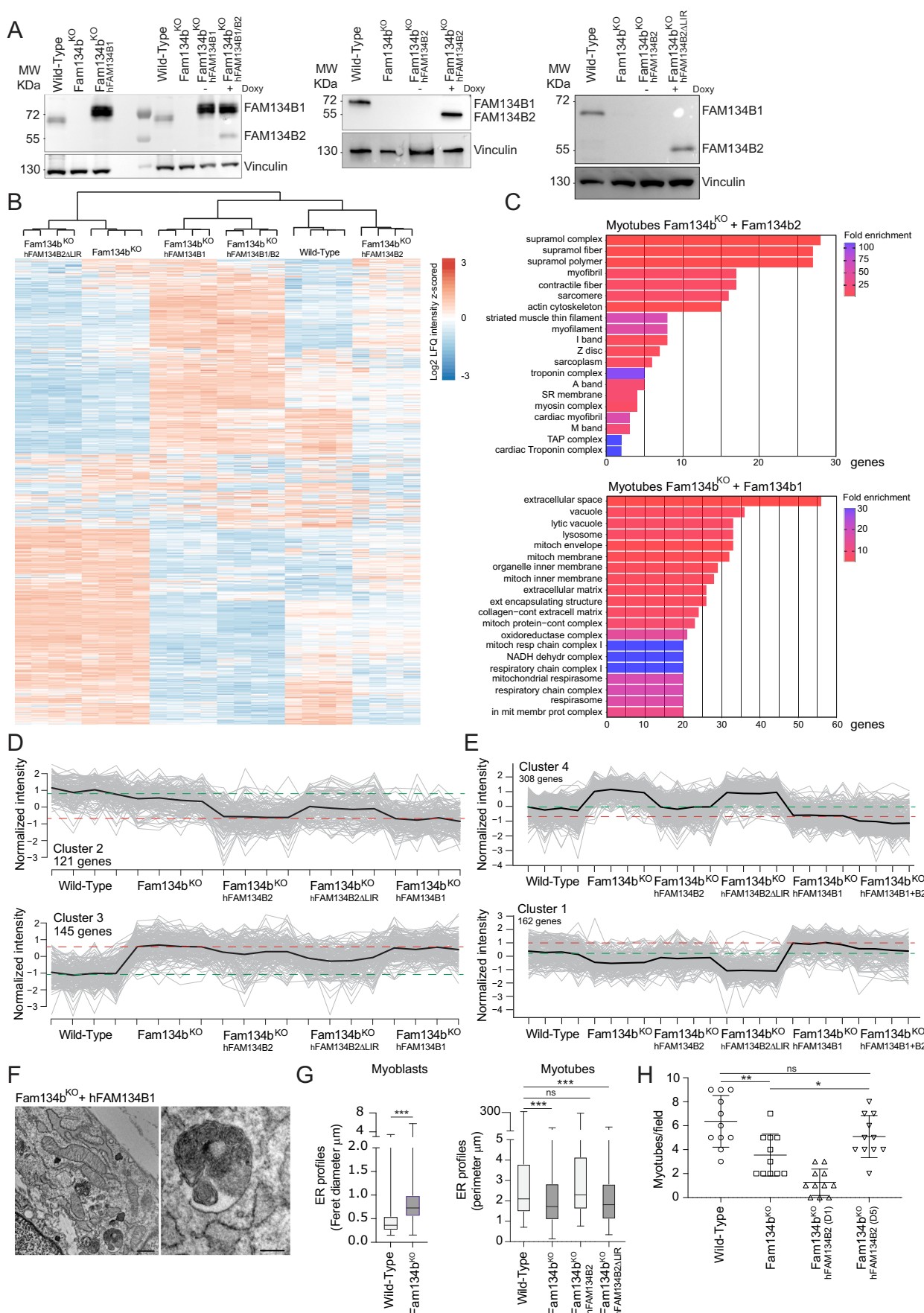

◀ **Figure EV5. Fam134b2 reconstitution is sufficient to rescue the proteomic abnormalities in Fam134b^KO myotubes.**

(A) WB analysis of FAM134B in Fam134b^KO C2C12 cells reconstituted with the indicated FAM134B isoforms. (B) Heatmap (Log$_2$ LFQ normalized intensity) of the full proteomes from WT, Fam134b^KO and reconstituted myotubes. (C) GOCC terms and frequencies of the significantly deregulated proteins in the proteomic analysis of Fam134b^KO myotubes reconstituted with hFAM134B2 or hFAM134B1. (D) Profile plot showing significantly regulated proteins according to ANOVA, specifically within cluster 2 and 3 of ER associated proteins. (E) Profile plots of significantly altered ER proteins belonging to clusters 4 and 1, as identified by ANOVA. (F) Electron microscopy images of the ER in Fam134b^KO myotubes reconstituted with hFAM134B1. Scale bar: 500 nm. Insets highlight autophagosome structures. Scale bar: 200 nm. (G) Left histograms represent the Feret's diameter measures from the ER of Wild-Type and Fam134b^KO myoblasts. Right histograms represent the perimeter of the ER from Wild-Type, Fam134b^KO and reconstituted myotubes. (H) Quantitative analysis of myotubes number in Wild-Type and Fam134b^KO C2C12 after 10 days of differentiation as well as in Fam134b^KO C2C12 overexpressing hFAM134B1 since D1 or hFAM134B2 from D5 of differentiation. Data are represented as mean ± s.d. *$P < 0.05$; **$P < 0.01$; ***$P < 0.001$ (Mann–Witney EV5G; one-way ANOVA EV5H). Mass spectrometry was performed in quadruplicate. Western blots and imaging were performed in three independent experiments. Source data are available online for this figure.

