## [Peer Review File · The EMBO Journal]

Two FAM134B isoforms differentially regulate ER form and dynamics during myogenesis

Paolo Grumati, Viviana Buonomo, Kateryna Lohachova, Alessio Reggio, Sara Cano, Michele Cillo, Lucia Santorelli, Rossella Venditti, Elena Polishchuk, Ivana Peluso, Lorene Brunello, Carmine Cirillo, Sara Petrosino, Malan Silva, Rossella De Cegli, Sabrina Di Bartolomeo, Cesare Gargioli, Paolo Swuec, Mirko Cortese, Alexandra Stolz, and Ramachandra Bhaskara

Corresponding author(s): Paolo Grumati (p.grumati@tigem.it)

Review Timeline:

Submission Date:	9th May 24
Editorial Decision:	17th May 24
Appeal Received:	25th Jun 24
Editorial Decision:	23rd Jul 24
Revision Received:	15th Oct 24
Editorial Decision:	26th Nov 24
Revision Received:	6th Dec 24
Accepted:	13th Dec 24

Editor: William Teale

Transaction Report:

Dear Paolo,

Thank you for submitting your manuscript entitled 'ER-phagy regulates ER dynamics during myogenesis' (EMBOJ-2023-117825) to our editorial office. We have now considered the study within our editorial team, and unfortunately come to the conclusion that we cannot offer publication in The EMBO Journal. The manuscript investigates the role of FAM134B proteins in shaping the sarcoplasmic reticulum. Focussing on ER-phagy-independent mechanisms, you use a sturdy proteomic platform to explore the functional divergence of FAM134B1 and FAM134B2 in developing muscle cells. From the experiments you present, you conclude that amounts of FAM134B1 and B2 protein are inversely regulated during myoblast differentiation, and that their functions are distinct. This work is likely to be of interest to the field. However, after careful reading we are not convinced that, as presented, the work is able sufficiently to discern between the mechanisms of FAM134B1 and B2 action to make the study a strong candidate for publication in a broad journal like The EMBO Journal at this stage.

However, given the potential value of your results to researchers directly in the field, we feel that the study should be well-suited for Life Science Alliance (<http://www.life-science-alliance.org/>), our open access journal launched in partnership with Rockefeller University Press and Cold Spring Harbor Laboratory Press. LSA aims to publish solid findings of value to particular communities across all areas of biology. I have therefore briefly discussed the work with the editors of Life Science Alliance, who would indeed be pleased to send your work for in-depth external review, without need for prior reformatting. Should you be interested in this option, please simply follow the link below for transfer. Eric Sawey, Executive Editor of Life Science Alliance (e.sawey@life-science-alliance.org), will be happy to answer any questions you may have.

I am sorry that I cannot be more positive for The EMBO Journal on this occasion, but very much hope that you will find this transfer option worthwhile.

Best wishes,

William

William Teale, PhD
Editor
The EMBO Journal
w.teale@embojournal.org

** As a service to authors, EMBO Press provides authors with the possibility to transfer a manuscript that one journal cannot offer to publish to another EMBO publication or the open access journal Life Science Alliance launched in partnership between EMBO Press, Rockefeller University Press and Cold Spring Harbor Laboratory Press. The full manuscript and if applicable, reviewers' reports, are automatically sent to the receiving journal to allow for fast handling and a prompt decision on your manuscript. For more details of this service, and to transfer your manuscript please click on Link Not Available. **

Dear Paolo,

Thank you again for the submission of your manuscript entitled "ER-phagy regulates ER dynamics during myogenesis" (EMBOJ-2024- 117825R-Q). We have now received the reports from the referees, which I copy below.

As you can see from their comments, the referees all suggest a handful of follow-up experiments that would certainly increase the robustness of your study. These issues will require your attention before your manuscript can be published in The EMBO Journal.

However, based on the overall interest expressed in the reports, I would like to invite you to address the comments of all referees in a revised version of the manuscript. I should add that it is The EMBO Journal policy to allow only a single major round of revision and that it is therefore important to resolve the main concerns at this stage. I believe the concerns of the referees are reasonable and addressable, but please contact me if you have any questions, need further input on the referee comments or if you anticipate any problems in addressing any of their points. A Zoom call to discuss the concerns of the referees might be useful; please suggest a time (after you have had a chance to digest the reports) that would be convenient. Please, follow the instructions below when preparing your manuscript for resubmission.

I would also like to point out that as a matter of policy, competing manuscripts published during this period will not be taken into consideration in our assessment of the novelty presented by your study ("scooping" protection). We have extended this 'scooping protection policy' beyond the usual 3 month revision timeline to cover the period required for a full revision to address the essential experimental issues. Please contact me if you see a paper with related content published elsewhere to discuss the appropriate course of action.

Again, please contact me at any time during revision if you need any help or have further questions.

Thank you very much again for the opportunity to consider your work for publication. I look forward to your revision.

Best regards,

William

William Teale, Ph.D.
Editor
The EMBO Journal

When submitting your revised manuscript, please carefully review the instructions below and include the following items:

- 1) a .docx formatted version of the manuscript text (including legends for main figures, EV figures and tables). Please make sure that the changes are highlighted to be clearly visible.
- 2) individual production quality figure files as .eps, .tif, .jpg (one file per figure).
- 3) a .docx formatted letter INCLUDING the reviewers' reports and your detailed point-by-point response to their comments. As part of the EMBO Press transparent editorial process, the point-by-point response is part of the Review Process File (RPF), which will be published alongside your paper.
- 4) a complete author checklist, which you can download from our author guidelines ([https://wol-prod-cdn.literatumonline.com/pb-assets/embo-site/Author Checklist%20-%20EMBO%20J-1561436015657.xlsx](https://wol-prod-cdn.literatumonline.com/pb-assets/embo-site/Author%20Checklist%20-%20EMBO%20J-1561436015657.xlsx)). Please insert information in the checklist that is also reflected in the manuscript. The completed author checklist will also be part of the RPF.
- 5) Please note that all corresponding authors are required to supply an ORCID ID for their name upon submission of a revised manuscript.
- 6) We require a 'Data Availability' section after the Materials and Methods. Before submitting your revision, primary datasets produced in this study need to be deposited in an appropriate public database, and the accession numbers and database listed under 'Data Availability'. Please remember to provide a reviewer password if the datasets are not yet public (see

<https://www.embopress.org/page/journal/14602075/authorguide#datadeposition>). If no data deposition in external databases is needed for this paper, please then state in this section: This study includes no data deposited in external repositories. Note that the Data Availability Section is restricted to new primary data that are part of this study.

Note - All links should resolve to a page where the data can be accessed.

8) For data quantification: please specify the name of the statistical test used to generate error bars and P values, the number (n) of independent experiments (specify technical or biological replicates) underlying each data point and the test used to calculate p-values in each figure legend. The figure legends should contain a basic description of n, P and the test applied. Graphs must include a description of the bars and the error bars (s.d., s.e.m.).

9) We would also encourage you to include the source data for figure panels that show essential data. Numerical data can be provided as individual .xls or .csv files (including a tab describing the data). For 'blots' or microscopy, uncropped images should be submitted (using a zip archive or a single pdf per main figure if multiple images need to be supplied for one panel). Additional information on source data and instruction on how to label the files are available at .

10) We replaced Supplementary Information with Expanded View (EV) Figures and Tables that are collapsible/expandable online (see examples in <https://www.embopress.org/doi/10.15252/embj.201695874>). A maximum of 5 EV Figures can be typeset. EV Figures should be cited as 'Figure EV1, Figure EV2" etc. in the text and their respective legends should be included in the main text after the legends of regular figures.

12) Our journal encourages inclusion of *data citations in the reference list* to directly cite datasets that were re-used and obtained from public databases. Data citations in the article text are distinct from normal bibliographical citations and should directly link to the database records from which the data can be accessed. In the main text, data citations are formatted as follows: "Data ref: Smith et al, 2001" or "Data ref: NCBI Sequence Read Archive PRJNA342805, 2017". In the Reference list, data citations must be labeled with "[DATASET]". A data reference must provide the database name, accession number/identifiers and a resolvable link to the landing page from which the data can be accessed at the end of the reference. Further instructions are available at .

We realize that it is difficult to revise to a specific deadline. In the interest of protecting the conceptual advance provided by the work, we recommend a revision within 3 months (21st Oct 2024). Please discuss the revision progress ahead of this time with the editor if you require more time to complete the revisions. Use the link below to submit your revision:

Referee #1:

Summary:

This study examines how two isoforms of the ER-phagy receptor FAM134B in regulating ER remodeling in differentiating myoblasts. During myogenesis, FAM134B1 is degraded, while FAM134B2 is upregulated, indicating its role in ER morphology. FAM134B2 reshapes the ER efficiently due to its rigid structure and is active during differentiation, reverting to normal levels in mature myotubes. Knocking out both isoforms results in abnormal proteomes and dilated ER structures, which are corrected by re-expressing FAM134B2. The results underscore how the fine-tuning of FAM134B isoforms and ER-phagy orchestrate ER dynamics during myogenesis, providing insights into the molecular mechanisms governing ER homeostasis in muscle cells, filling a gap in scientific knowledge. Overall, this is a very elegant and comprehensive study with important mechanistic findings that may also offer potential molecular targets for further research into muscle cell-related diseases and ER homeostasis mechanisms. Nevertheless, several critical points remain to be addressed.

1. This study mainly used C2C12 cells, while C2C12 myoblasts are a commonly used model for studying myogenesis, they are an immortalized cell line and may not fully recapitulate the physiological conditions and complexities of primary myoblasts or in vivo muscle tissue. This could limit the generalizability of the findings. So the authors should complement their study with experiments using primary myoblasts and satellite cells isolated from mouse or human tissues to validate the findings.
2. This study mentions that differentiation was induced by culturing confluent cells for 10 days with daily replacement of the growth medium. This approach might not ensure uniform differentiation conditions, potentially leading to variability in the differentiation state of the cells, which can affect the consistency and reliability of the proteomic data. If possible, the authors should validate key findings by the use of defined differentiation protocols, such as switching to low-serum media or adding specific differentiation factors (e.g., horse serum) to ensure consistent and uniform differentiation of myoblasts.
3. Page 6, this study relies on general markers of autophagy (LC3B, p62) and EM analysis to infer ER-phagy activation. However, specific markers for ER-phagy receptors or markers indicating ER components (such as ER luminal proteins) in autophagosomes or autolysosomes are not monitored at endogenous levels. This makes it challenging to conclusively attribute the observed autophagosome-like structures to ER-phagy. The authors should corroborate their findings by probing autophagosomes or autolysosomes for ER proteins.
4. Page 7, Fig 2G, the expression of control protein vinculin is distinct lower in T.Anterior sample than the other two, which makes this western blot result unreliable.
5. Page 13, "Indeed, other forms of selective autophagy and macroautophagy

remained active, as evidenced by WB for Lc3b and p62, indicating robust activation of autophagy in Fam134bKO (Figs 6D, E)." There is no statistic difference of Lc3b and p62 between WT and KO group via western blot results, hence this conclusion is not really supported by the data.

6. Page 14, "To explore this, we reconstituted Fam134bKO C2C12 cells with constitutively....." This experiment used constitutive expression for hFAM134B1 and inducible expression for hFAM134B2. These different expression strategies might introduce bias since constitutive expression may not mimic the physiological conditions accurately. Comparing results from cells with constitutively expressed proteins to those with inducibly expressed proteins may not yield comparable results. The authors should unify their expression strategies for the comparison of the two FAM134B isoforms.

7. Page 36, line 7 "p<0.5", a typo?

Referee #2:

Buonomo et al probe the alterations in ER dynamics during an in vitro differentiation model of myoblasts to myotubes, modelling myogenesis. They show that remodelling of the ER and formation of myotubes depends on FAM134B, in particular a switch from FAM134B-1 to the lesser characterised FAM134B-2, which may be more suited to the ER-phagy demands of the differentiated state.

Strengths

- Deep, data-driven datasets describing role of FAM134B during myogenesis
- Insight into regulation of FAM134B ER-phagy axis, including further elucidation of the mysterious FAM134B-2 isoform
- New insight into molecular mechanisms of muscle cell differentiation
- Potentially new physiological relevant roles for ER-phagy

Weakness

- FAM134B-2 involvement in myotubes versus FAM134B not yet clear from rescue experiments performed in latter figures.

Additional controls needed (see below).

Overall, this is a substantial, informative study that will be of broad interest to the readership of EMBO journal, but more convincing evidence for the switch to FAM134B-2 function is required before publication.

MAJOR POINTS

- 1) Fig 11 - confusing. Is this ER-phagy reporter ss-RFP-GFP-KDEL or separate RFP-KDEL constructs and GFP-KDEL in the same myoblast population? If so, why? Is this an ER-phagy assay or simply a visualisation of ER morphology? If the latter, why the two fluorophores that may be behaving differently (but not easy to see in this panel)?
- 2) Fig 2A - cannot make the claim that bulk autophagy flux is activated during myogenesis without employing one of the well-established flux assays that capture the dynamics of the process. Steady-state LC3 and p62 blots are insufficient.
- 3) Fig. 2D - Mean LFQ intensity? In which cell type? Not sure if this plot is informative in main data if simply to show that ER-phagy receptors were detected in the ms.
- 4) Fig. 3 - It is interesting that FAM134B-2 can form foci, albeit with less efficiency than FAM134B-1. This suggests that some ER vesiculating activity is retained. However, expression levels should be properly controlled for in these experiments. Furthermore, independent investigations of ER morphology that do not rely on FAM134B foci counts are required to truly interpret these data as ER remodelling, rather than simply FAM134B clustering. Same applies to LIR mutant FAM134B-2 overexpression in Fig. 4D.
- 5) Fig. 3K - the effect of FAM134B-2 expression on driving myotube formation is weak, but not nothing. However, what is this compared to? Empty vector? Is it not equally likely that FAM134B-1 inhibits myotube formation and FAM134B-2 is neutral.
- 6) Really, knockout of endogenous FAM134B and rescue with either FAM134B-1 or -2 in myoblasts is required to establish both that ER-phagy flux alterations/ER remodelling and myotube formation are influenced in a specific manner by these different isoforms. I acknowledge that this is done extensively for the proteome data in latter figures, but direct comparison of FAM134B-1 and -2 in this manner is not done for myotube formation or ER remodelling (Fig. 7E-F).
- 7) Furthermore, expression levels versus endogenous need more carefully controlled for in the comprehensive FAM134B knockout rescue experiments, especially given the phenomenon of "over-rescue" by FAM134B-1 seen in some of the proteomic dataset. This highlights possibility of overexpression artefact (e.g. see Figure EV7 panel A where reconstituted FAM134B is expressed at high levels compared to physiologic levels).

Referee #3:

In their manuscript, Buonomo and colleagues study the role of ER-selective autophagy in myogenesis. They begin by showing that myotube formation of C2C12 model myoblasts involves substantial proteome remodeling. ER proteins show particularly strong changes, which is logical given the development of muscle-specific sarcoplasmic reticulum. Next, they provide evidence that myogenesis involves an induction of autophagy, including ER-phagy. Interestingly, this activation is accompanied by downregulation of the B1 isoform of the ER-phagy receptor FAM134 and an upregulation of the B2 isoform, suggesting that the two isoforms have different roles at different stages of myogenesis. Using molecular dynamics simulations, the authors go on to show that the two FAM134B isoforms are likely to have different structures but that both have the ability to remodel the ER membrane. The authors then apply proteomic analyses to define the interactome of FAM134B1 and B2 homo- and heterodimers, the protein abundance changes that occur in myoblasts and myotubes upon FAM134B knockout and the protein abundance changes upon re-expression of wild-type and autophagy-defective variants of the B1 and B2 isoforms. Based on

these data, the authors conclude that ER remodeling by FAM134B-dependent ER-phagy plays an important role in myogenesis.

This study is interesting because it addresses a potential role of ER-phagy in cell differentiation. It thus analyzes a more physiological setting than the extreme starvation conditions that are typically applied to trigger autophagy experimentally. The initial analysis of ER-phagy induction and the characterization of the two FAM134B isoforms (Figures 1-3) are, for the most part, carefully done, informative and technically impressive (as far as I can tell because I cannot judge the MD simulations). The second part consists mainly of transcriptomics and proteomics data (Figures 4-7), which unfortunately remain largely descriptive. The proteomics data show broad trends but there is little biochemical and morphological follow-up work (essentially only Figure 6D,E and Figure 7C-F). As a result, the data do not elucidate the specific roles of FAM134B in myogenesis. It therefore remains unclear which ER proteins are, and perhaps need to be, degraded via ER-phagy to allow efficient myogenesis. In fact, it is not clear if and to what extent the formation of myotubes is disturbed in the absence of ER-phagy. To bring this study to a satisfying level of understanding regarding both the molecular role of FAM134B and its functional significance in myogenesis it would be necessary to determine the turnover of specific ER proteins during myogenesis and assess the efficiency of myotube formation itself.

Overall, this is an interesting study but lacks in-depth analysis of the proteomics data and functional follow-up experiments, and thus remains too descriptive (for more detail, see major comments below). Other than that the data are overall presented well but there are inaccuracies and places in which they could be made more accessible to the reader (see minor comments below). Therefore, I do not recommend publication in the EMBO Journal, for which I think the authors would need to revise and extend their study substantially.

Major comments:

1. Figure 4E-H: The IP-MS shows hundreds of proteins that are precipitated with dimeric FAM134, including many mitochondrial proteins. The authors point out that LC3B family proteins are more strongly enriched in the precipitate of the B2 isoform but otherwise little effort is made to understand which of the many co-precipitating proteins are the real and main interactors of FAM134. As a result, the value of the data is unclear. I understand that this is what mass spec data look like but I cannot envision that FAM134 dimers directly bind to several hundred proteins.
2. Figures 5E-F, 6A-B and 7A-B: From the presentation of the data, I am unable to picture the impact of FAM134 manipulation on the proteome. Proteins are grouped in large clusters, no individual protein is subsequently analyzed (although this was done for REEP5 and RTN4 in the first part of the paper, see Figure EV1) and extent of abundance changes is unclear. To make the data intelligible for the reader, a more in-depth analysis is needed. A point of reference could be the conceptually similar study Hoyer et al, NCB 2024, PMID 38429475, in which the authors do extensive follow-up work on their proteomic data to highlight quantitatively changes in the levels as well as the autophagic turnover of individual proteins.
3. p14, first line and Figure 6D, E: The authors state that 'the rate of differentiation diverged compared to that of WT myoblasts' in FAM134B knockout cells. However, they look at MF20 as the only marker of differentiation (without justifying that choice) and see that its levels rise later in the knockout but than increase more strongly than in the wildtype. This may indicate a problem in myogenesis but it remains unclear what the altered expression pattern of MF20 means. I think more experiments are needed here to assess myotube formation. The authors used a count of myotubes/field of view in Figure 3I, K, which is a start. Additionally, I wonder if perhaps it could be informative to look at ER stress during differentiation. Seeing the massive but transient ER dilation during myotube formation, there may be an activation of the UPR in these cells and it would be interesting to know if this potential UPR activation is altered in the absence of FAM134B.

Minor comments:

1. p3, first paragraph: "Few studies indicate ...". I suspect the authors want to point out that there indeed are studies on ER proteins in myogenesis, not that there is only a small number of them. In this case, the sentence should read "A few studies indicate ...".
2. p4, second paragraph: "helixes" should be helices.
3. p4, second paragraph: when citing papers on the oligomerization of FAM134 (references 24 - 26), it would be good to include also Wang et al, JCB 2023, PMID: 37043189.
4. p4, results, first paragraph: "... spontaneous differentiation in standard growth media ...". Was there only one growth medium or several media? If the former, then it should be "in standard growth medium ...".
5. Figure 1A: Spell out 'GM'. Indicate that the schematic represents progression of time. Please explain in the text or the figure legend what MF20 is.

6. p4, results, first paragraph. Please make explicit in the text that cells spontaneously differentiate once they reach confluency and state how long this takes. This becomes apparent when looking at the figures but explaining these things in the beginning would be helpful for readers unfamiliar with C2C12 cells.
7. Figure EV1A: Please explain what MyoG is.
8. Figure 1C: Please explain in the text or the figure legend what is plotted, especially in the lower part of the panel. I assume that the lower part in some way shows a comparison of the abundance of various proteins in myoblasts versus myotubes. Not being a mass spectrometry expert, it was not clear to me why a PCA plot is shown rather than a simple volcano plot.
9. p6, first paragraph and Figure 2A: In the text it says that "Western blot analysis of Lc3b ... confirmed a progressive activation of autophagy flux ...". The western and the accompanying quantification appears to show the levels of LC3B-I and -II. This does not allow to say anything about autophagic flux. For this, it is not sufficient to show the combined abundance of non-lipidated LC3B-I and lipidated LC3B-II but it would be necessary to show that their levels increase upon inhibition of lysosomal degradation (see, for example, PMID 17611390 or PMID 22966490).
10. Figure 2C: Please label in the figure that ER-phagy is assayed with RFP-GFP-KDEL so that this is immediately obvious to the reader.
11. p8, first paragraph: the N-terminal end of a protein is called N-terminus, not "N-terminal".
12. Figure EV3D: Please explain why TGN38 was included and also what the comparison of Triton and Digitonin signifies.
13. p8, second paragraph: "The embedded molecules perturbed of the bicelle". I think the "of" should be deleted.
14. p8, last line: "...both molecules exhibited an almost similar ability". 'Almost similar' sounds odd. Two things can be similar or almost identical but not almost similar.
15. p9, first paragraph and Figure 3G, H: "Overexpression of FAM134B2 ... was sufficient to fragment ER membranes, albeit at a slower rate compared to FAM134B1." The micrographs show puncta of FAM134B, which in itself does not mean that the ER membrane fragments but could simply mean that FAM134B clusters within the ER. To show ER fragmentation, it would be necessary to show that a luminal ER marker forms physically separate structures or that FAM134B is transported to lysosomes (as in Figure 4D). Second, the snap shots in Figure 3G do not allow any statement about "rates".
16. p9, first paragraph and Figure 3I, K: To show that FAM134B2 "contributed to myotube development" it would be necessary to show that FAM134B2 overexpression changes the efficiency of myogenesis, i.e. it would be necessary to compare myotube number in FAM134B2-overexpressing and control cells. As is, it is impossible to say whether FAM134B1 or FAM134B2 overexpression has any impact.
17. p9, first paragraph: "regenerated less ER fragments". It should be 'fewer', not 'less'.
18. p9, second paragraph: "FAM134B2 confers more stability of the ER network promoting myotubes development." I do not understand what is meant by "stability of the ER network" here. As for myotube development, see comment above.
19. p10, first paragraph: Please explain the assay and how only FAM134B dimers are precipitated by reconstituted GFP. This is a very elegant idea but many readers won't be familiar with it.
20. p11, second paragraph: "FAM134B1 exhibits ... but a higher affinity for ER-related proteins, autophagy adaptors and development elements." I do not think that the mass spec results allow any statement about affinities.
21. Figure 7F: The authors show fluorescence micrographs to illustrate the morphological impact of loss and re-expression of FAM134B. These images look promising - can they be quantified?

Rebuttal Letter

We thank the Referees for their insightful comments and suggestions. We performed several new experiments to address the open questions and included the new data in reshaped figures and text accordingly.

Referee #1:

Summary:

This study examines how two isoforms of the ER-phagy receptor FAM134B in regulating ER remodeling in differentiating myoblasts. During myogenesis, FAM134B1 is degraded, while FAM134B2 is upregulated, indicating its role in ER morphology. FAM134B2 reshapes the ER efficiently due to its rigid structure and is active during differentiation, reverting to normal levels in mature myotubes. Knocking out both isoforms results in abnormal proteomes and dilated ER structures, which are corrected by re-expressing FAM134B2. The results underscore how the fine-tuning of FAM134B isoforms and ER-phagy orchestrate ER dynamics during myogenesis, providing insights into the molecular mechanisms governing ER homeostasis in muscle cells, filling a gap in scientific knowledge. Overall, this is a very elegant and comprehensive study with important mechanistic findings that may also offer potential molecular targets for further research into muscle cell-related diseases and ER homeostasis mechanisms. Nevertheless, several critical points remain to be addressed.

1. This study mainly used C2C12 cells, while C2C12 myoblasts are a commonly used model for studying myogenesis, they are an immortalized cell line and may not fully recapitulate the physiological conditions and complexities of primary myoblasts or in vivo muscle tissue. This could limit the generalizability of the findings. So the authors should complement their study with experiments using primary myoblasts and satellite cells isolated from mouse or human tissues to validate the findings.

We thank the Referee for this observation which is an important point.

In the old Figs EV1G-J, we presented mass spectrometry analysis performed in human and murine primary myoblasts and differentiated myotubes. From biochemical analysis, we could

appreciate that the protein variations were like the ones observed for C2C12 myoblasts and differentiated myotubes (at least concerning ER and lysosome/autophagy related proteins). We further elaborate our data to better highlight the analogies with C2C12 myoblasts and myotubes (New Figs EV1G-N; Dataset EV2,3).

We also performed new analysis in human myoblasts and myotubes. We monitored ER-phagy activation during human myoblasts differentiation following ER proteins degradation into lysosome (New Fig. EV2C). Moreover, we confirmed the behavior of FAM134B isoforms, as observed in C2C12 myoblasts, during their differentiation (Fig. EV2I). Human myoblasts are more complicated to handle and differentiate; therefore, we had to seed them at a high density. At day2 they already started to get in contact. This explains why Fam134b2 and MF20 were detected at this timepoint. During differentiation both protein levels increased.

2. This study mentions that differentiation was induced by culturing confluent cells for 10 days with daily replacement of the growth medium. This approach might not ensure uniform differentiation conditions, potentially leading to variability in the differentiation state of the cells, which can affect the consistency and reliability of the proteomic data. If possible, the authors should validate key findings by the use of defined differentiation protocols, such as switching to low-serum media or adding specific differentiation factors (e.g., horse serum) to ensure consistent and uniform differentiation of myoblasts.

We previously tested if differentiating primary murine and human cells, employing a differentiation media (using 2% horse serum), could affect the proteome landscape and/or the variations in the ER proteins. We did not reported significant changes therefore we decided to continue with the spontaneous differentiation protocols which is easier to handle (New Figs EV1H,L).

We also performed additional experiments differentiating C2C12 cells with 2% horse serum. We did not report significant differences between the two differentiation setups (New Figs EV2E and EV2H).

3. Page 6, this study relies on general markers of autophagy (LC3B, p62) and EM analysis to infer ER-phagy activation. However, specific markers for ER-phagy receptors or markers indicating ER components (such as ER luminal proteins) in autophagosomes or autolysosomes are not monitored at endogenous levels. This makes it challenging to conclusively attribute the

observed autophagosome-like structures to ER-phagy. The authors should corroborate their findings by probing autophagosomes or autolysosomes for ER proteins.

To address this point, we performed IF experiments in TMEM192-3xHA (lysosomal marker) stable C2C12 differentiated into myotubes using HA specific Ab together with a Reep5 specific Ab. We could appreciate the colocalization of TMEM192-3xHA positive structures with the ER marker Reep5. We added these data in New Fig EV2B and New Fig 6D.

In the old Fig. 6B (now Fig. 6C), we reported the results obtained from Lyso-IP MS/MS analysis. For this experiment, we generated stable C2C12 expressing TMEM192-3xHA (lysosomal marker) and differentiated them into myotubes. We pulled down lysosomes, employing HA beads, and performed MS/MS to identify the endogenous proteins in the lysosomal compartments. We could detect several ER related proteins, Dataset EV11.

We monitored ER degradation employing the ER-phagy reporter ssRFP-GFP-KDEL that is commonly used to monitor ER degradation via lysosomes (Chino et al., Mol Cell 2019; Reggio et al., EMBO Rep 2021).

4. Page 7, Fig 2G, the expression of control protein vinculin is distinct lower in T. Anterior sample than the other two, which makes this western blot result unreliable.

In this WB, we loaded the same amount of total protein extract (20 μ g). The difference in the protein level of Vinculin is because mature muscle tissue has a general protein profile different respect to myoblasts and myotubes. The idea was to highlight the fact that Fam134b2 is the dominant isoform in mature muscle.

5. Page 13, "Indeed, other forms of selective autophagy and macroautophagy remained active, as evidenced by WB for Lc3b and p62, indicating robust activation of autophagy in Fam134bKO (Figs 6D, E)." There is no statistic difference of Lc3b and p62 between WT and KO group via western blot results, hence this conclusion is not really supported by the data.

It is correct, there is no difference in macro-autophagy activation between WT and KO groups. We wanted to point out that Fam134b knockout, as expected, influences only the selective autophagy of the ER (ER-phagy) while the general macro-autophagy remains unaltered. We performed new experiments employing GFP-LC3B-RFP stable C2C12 cells (New Fig 6F).

6. Page 14, "To explore this, we reconstituted Fam134bKO C2C12 cells with constitutively....." This experiment used constitutive expression for hFAM134B1 and inducible expression for hFAM134B2. These different expression strategies might introduce bias since constitutive expression may not mimic the physiological conditions accurately. Comparing results from cells with constitutively expressed proteins to those with inducibly expressed proteins may not yield comparable results. The authors should unify their expression strategies for the comparison of the two FAM134B isoforms.

It is difficult to reproduce the physiological conditions. FAM134B1 is expressed in myoblasts and its protein level decreases during myogenesis, while FAM134B2 is not expressed in myoblasts while its transcription starts only when cells fuse together.

We think that the strategy we adopted is not perfect but appropriate to investigate the function of the two isoforms. We ensured to have FAM134B1 since the beginning, as it is for the wild-type C2C12. Differently, FAM134B2 is modulated by doxycycline to be expressed when its natural transcription is activated, and it is absent in myoblasts where it is not detected even at the endogenous level.

The two isoforms have a different expression and regulation in physiological conditions. Moreover, the switch of the two isoforms is a key element in the process. We expressed FAM134B2 in C2C12 cells, since the D1 of differentiation, so that they were not differentiating yet. The premature expression of Fam134B2 had the effect of reducing the myogenesis (New Fig EV7H).

7. Page 36, line 7 " $p < 0.5$ ", a typo?

Yes, it is, Sorry

Referee #2:

Buonomo et al probe the alterations in ER dynamics during an in vitro differentiation model of myoblasts to myotubes, modelling myogenesis. They show that remodelling of the ER and formation of myotubes depends on FAM134B, in particular a switch from FAM134B-1 to the lesser characterised FAM134B-2, which may be more suited to the ER-phagy demands of the differentiated state.

Strengths

- Deep, data-driven datasets describing role of FAM134B during myogenesis
- Insight into regulation of FAM134B ER-phagy axis, including further elucidation of the mysterious FAM134B-2 isoform
- New insight into molecular mechanisms of muscle cell differentiation
- Potentially new physiological relevant roles for ER-phagy

Weakness

- FAM134B-2 involvement in myotubes versus FAM134B not yet clear from rescue experiments performed in latter figures. Additional controls needed (see below).

Overall, this is a substantial, informative study that will be of broad interest to the readership of EMBO journal, but more convincing evidence for the switch to FAM134B-2 function is required before publication.

MAJOR POINTS

1) Fig 1I - confusing. Is this ER-phagy reporter ss-RFP-GFP-KDEL or separate RFP-KDEL constructs and GFP-KDEL in the same myoblast population? If so, why? Is this an ER-phagy assay or simply a visualisation of ER morphology? If the latter, why the two fluorophores that may be behaving differently (but not easy to see in this panel)?

We are sorry if we were not clear regarding Fig. 1I.

We generated two different C2C12 stable cell lines: GFP-KDEL – C2C12 and RFP-KDEL – C2C12. We selected the cells and mixed GFP-KDEL with RFP-KDEL myoblasts in equal amount and let them fuse and differentiate. We wanted to highlight the heterotypic fusion of the ER membranes during C2C12 differentiation.

The behavior of the fluorophores is not critical because it does not influence the myoblasts differentiation. Differences in the fluorophores' intensities are also due to different stage of myotubes differentiation. We fixed the experiment at an early time-point when the C2C12 starts to fuse. This set up better show how the ER membranes from single myoblasts fuse together in the more organized ER of the myotubes.

2) Fig 2A - cannot make the claim that bulk autophagy flux is activated during myogenesis without employing one of the well-established flux assays that capture the dynamics of the process. Steady-state LC3 and p62 blots are insufficient.

To address this point, we generated GFP-LC3B-RFP C2C12 stable cell line and monitored autophagy flux during myoblasts differentiation.

New results are reported in New Fig 2C and New Fig 6F.

3) Fig. 2D - Mean LFQ intensity? In which cell type? Not sure if this plot is informative in main data if simply to show that ER-phagy receptors were detected in the ms.

We agree with the Referee, this plot is not so informative, so we moved in New Fig EV2F. The plot was obtained starting from the Dataset EV1 performed in C2C12 and differentiated myotubes.

4) Fig. 3 - It is interesting that FAM134B-2 can form foci, albeit with less efficiency that FAM134B-1. This suggests that some ER vesiculating activity is retained. However, expression levels should be properly controlled for in these experiments. Furthermore, independent investigations of ER morphology that do not rely on FAM134B foci counts are required to truly interpret these data as ER remodelling, rather than simply FAM134B clustering. Same applies to LIR mutant FAM134B-2 overexpression in Fig. 4D.

As suggested by the Referee we monitored the FAM134B1 and FAM134B2 expression (New EV Fig 3H; New Fig EV4B).

FAM134B dots are not simply aggregates of the protein itself. They are portion of ER that are remodeled into vesicles. We performed IF using Ab against HA together with specific Abs for the ER proteins Canx and Reep5. The colocalization of HA signal with the ER proteins staining indicates that the ER membranes are subjected to remodeling that is driven by FAM134B. We

now better highlighted these data moving them in New Fig 3A. Additional IF are reported in New Figs 4D and EV4A.

The fact that FAM134B dots are associated to a remodel of the ER membranes is also supported by previous literature on FAM134B: Khaminets et al. 2015 Nature; Bhaskara et al. 2019 Nat Comm; Reggio et al. 2021 EMBO Rep; Gonzalez et al. 2023 Nature).

5) Fig. 3K - the effect of FAM134B-2 expression on driving myotube formation is weak, but not nothing. However, what is this compared to? Empty vector? Is it not equally likely that FAM134B-1 inhibits myotube formation and FAM134B-2 is neutral.

To clarify this point, we added the control condition. We used ssRFP-GFP-KDEL that is cloned in the same backbone vector we used for the Fam134b. Moreover, we acquired new imagines to increase the number of counts and improve new statistical analysis. Data are reported in the New Figs 3J-M.

6) Really, knockout of endogenous FAM134B and rescue with either FAM134B-1 or -2 in myoblasts is required to establish both that ER-phagy flux alterations/ER remodelling and myotube formation are influenced in a specific manner by these different isoforms. I acknowledge that this is done extensively for the proteome data in latter figures, but direct comparison of FAM134B-1 and -2 in this manner is not done for myotube formation or ER remodelling (Fig. 7E-F).

We agree with the Referee that a direct comparison between FAM134B1 and FAM134B2 in terms of ER remodeling and myotubes formation is missing. To address this point, we investigated the rescue of ER diameter of Fam134b knockout myotubes after the reconstitution with hFAM134B1. We did not appreciate a rescue as we observed for FAM134B2 (New Figs 7E and EV7F). From the MS data (see also Fig EV7E) and the new EM analysis it is quite clear that FAM134B1 does not have the same effect as FAM134B2 at least in this biological context. Regarding myotubes formation, we reported the comparison between the two FAM134B isoforms in the New Figs 3J-M. Our data indicate that FAM134B2 expression is a propeller for myotubes formation when correctly expressed.

7) Furthermore, expression levels versus endogenous need more carefully controlled for in the comprehensive FAM134B knockout rescue experiments, especially given the phenomenon of

"over-rescue" by FAM134B-1 seen in some of the proteomic dataset. This highlights possibility of overexpression artefact (e.g. see Figure EV7 panel A where reconstituted FAM134B is expressed at high levels compared to physiologic levels).

It is difficult to reproduce the physiological conditions. FAM134B1 is expressed in myoblasts and decreases during myogenesis, while FAM134B2 is not expressed in myoblasts and its transcription starts only when myoblasts fuse together.

We think that the strategy we adopted is not perfect but appropriate to investigate the function of the two isoforms. We ensured to have FAM134B1 since the beginning, as it is for the Wild-Type C2C12. Differently, FAM134B2 is modulated by doxycycline to be expressed when its natural transcription is activated, and it is absent in myoblasts where it is not detected even at the endogenous level.

The two isoforms have a different expression and regulation in physiological conditions. However, the switch of the two isoforms is a key element in the process too.

Moreover, it is difficult to modulate the expression of FAM134B isoform, like the endogenous level, considering that the experiments required ten days of differentiation with a constant refresh of the media.

Overall, both isoforms are over-expressed respect to their endogenous level and only the FAM134B2 showed a clear rescue. We performed several types of analysis including full proteome mass spectrometry, immuno-fluorescence, EM and western blot. All the experiments showed the same results therefore, even if some artefacts are possible, we think that our experimental setting is valuable, and our data are solid.

Referee #3:

In their manuscript, Buonomo and colleagues study the role of ER-selective autophagy in myogenesis. They begin by showing that myotube formation of C2C12 model myoblasts involves substantial proteome remodeling. ER proteins show particularly strong changes, which is logical given the development of muscle-specific sarcoplasmic reticulum. Next, they provide evidence that myogenesis involves an induction of autophagy, including ER-phagy. Interestingly, this activation is accompanied by downregulation of the B1 isoform of the ER-phagy receptor FAM134 and an upregulation of the B2 isoform, suggesting that the two isoforms have different roles at different stages of myogenesis. Using molecular dynamics simulations, the authors go on to show that the two FAM134B isoforms are likely to have different structures but that both have the ability to remodel the ER membrane. The authors then apply proteomic analyses to define the interactome of FAM134B1 and B2 homo- and heterodimers, the protein abundance changes that occur in myoblasts and myotubes upon FAM134B knockout and the protein abundance changes upon re-expression of wild-type and autophagy-defective variants of the B1 and B2 isoforms. Based on these data, the authors conclude that ER remodeling by FAM134B-dependent ER-phagy plays an important role in myogenesis.

This study is interesting because it addresses a potential role of ER-phagy in cell differentiation. It thus analyzes a more physiological setting than the extreme starvation conditions that are typically applied to trigger autophagy experimentally. The initial analysis of ER-phagy induction and the characterization of the two FAM134B isoforms (Figures 1-3) are, for the most part, carefully done, informative and technically impressive (as far as I can tell because I cannot judge the MD simulations). The second part consists mainly of transcriptomics and proteomics data (Figures 4-7), which unfortunately remain largely descriptive. The proteomics data show broad trends but there is little biochemical and morphological follow-up work (essentially only Figure 6D,E and Figure 7C-F). As a result, the data do not elucidate the specific roles of FAM134B in myogenesis. It therefore remains unclear which ER proteins are, and perhaps need to be, degraded via ER-phagy to allow efficient myogenesis. In fact, it is not clear if and to what extent the formation of myotubes is disturbed in the absence of ER-phagy. To bring this study to a satisfying level of understanding regarding both the molecular role of FAM134B and its functional significance

in myogenesis it would be necessary to determine the turnover of specific ER proteins during myogenesis and assess the efficiency of myotube formation itself.

Overall, this is an interesting study but lacks in-depth analysis of the proteomics data and functional follow-up experiments, and thus remains too descriptive (for more detail, see major comments below). Other than that the data are overall presented well but there are inaccuracies and places in which they could be made more accessible to the reader (see minor comments below). Therefore, I do not recommend publication in the EMBO Journal, for which I think the authors would need to revise and extend their study substantially.

Major comments:

1. Figure 4E-H: The IP-MS shows hundreds of proteins that are precipitated with dimeric FAM134, including many mitochondrial proteins. The authors point out that LC3B family proteins are more strongly enriched in the precipitate of the B2 isoform but otherwise little effort is made to understand which of the many co-precipitating proteins are the real and main interactors of FAM134. As a result, the value of the data is unclear. I understand that this is what mass spec data look like but I cannot envision that FAM134 dimers directly bind to several hundred proteins.

We agree with the Referee that the number of proteins in FAM134B1 dimers looks unusual. This large number is since we used a DIA setup in the sample analyses.

DIA entails the fragmentation of each and every single analyzed ion present in a sample, after which the corresponding mass-to-charge (m/z) ratios are calculated. This gives the spectrum of fragment ions, a "mass spectrum", that can be exploited to recognize the various components within the sample. Respect to data-dependent acquisition (DDA), where the instrument selects specific ions to fragment based on their intensity or abundance, DIA captures all of the fragment ions within a predetermined m/z range in a methodical and impartial fashion. This enables the detection and quantification of every detectable analyte in the sample, irrespective of their level of abundance or m/z value. DIA has several advantages over the DDA. It is able to detect low-abundance peptides, has an increased specificity (it can differentiate isobaric peptides that has the same m/z but different sequences) and highly reproducibility (all ions within a pre-defined m/z range are analyzed in every run).

Considering our interest in the autophagy machinery we validated the interaction between FAM134B1 or FAM134B2 with LC3B, GABARAP in myotubes (New Fig EV4G).

2. Figures 5E-F, 6A-B and 7A-B: From the presentation of the data, I am unable to picture the impact of FAM134 manipulation on the proteome. Proteins are grouped in large clusters, no individual protein is subsequently analyzed (although this was done for REEP5 and RTN4 in the first part of the paper, see Figure EV1) and extent of abundance changes is unclear. To make the data intelligible for the reader, a more in-depth analysis is needed. A point of reference could be the conceptually similar study Hoyer et al, NCB 2024, PMID 38429475, in which the authors do extensive follow-up work on their proteomic data to highlight quantitatively changes in the levels as well as the autophagic turnover of individual proteins.

We agree with the Referee that the amount of MS data could be difficult to digest. We tried to better explain the analysis in the text, and we re-elaborated our mass spectrometry data.

We now better explain the meaning of the heatmap in Fig 5E. We added the plot (New Fig 5F) where we highlighted the GOCC that are commonly upregulated (mostly mitochondria terms) or downregulated (mostly nuclear terms) in both wild-type and Fam134b^{KO} myotubes. This plot also highlights that the ER terms are upregulated only in Fam134b^{KO} myotubes. In the New Fig 5G we reported a volcano plot of the Wild-Type and Fam134b^{KO} full proteome where we highlighted the ER proteins.

We moved the old Fig 5F that represents the profile plots of the protein clusters obtained from the Wild-Type and Fam134b^{KO} full proteome analysis. Now the plots are in the New Fig EV5F and next to them we reported the GOBP and GOCC terms associated to each cluster (New Fig EV5G,H).

Fig 6A represent the profile plots of the ER associated proteins obtained from the full proteome analysis. To clarify the meaning of the clusters, we added the New Fig 6B where we highlighted the ER proteins whose intensity is higher in Fam134b^{KO} myotubes compared to wild type. We validated our MS data performing WB analysis of some ER proteins from the myotubes of the two genotypes (New Fig EV 6A).

The old Fig 6B represents the content of the ER proteins obtained from the IP-MS/MS of the lysosomal structure immunoprecipitated from the Wild-Type and the Fam134b^{KO} myoblasts. In the New Fig 6D, we showed the co-localization, in Wild-Type myotubes but not in Fam134b^{KO} ones, of the ER marker Reep5 with the lysosomal marker TMEM192.

Fig 7A represent the heatmap of the ANOVA significant ER proteins obtained from the full proteome analysis presented in Fig EV7B. We wanted to highlight the dominant effect of hFAM134B2 in the reconstitution of the ER protein landscape. Fig 7B represents the profile plot of two clusters of ER proteins where the effects of hFAM134B2, hFAM134B2 Δ LIR and hFAM134B1 were more evident.

Our work is less devoted to the proteomic analysis respect to Harper's manuscript. We think that performing extensive follow-up work on the proteomic data, like Harper's group did, is out of the scope of this work. We employed several experimental setups (mass spectrometry, mRNA seq, confocal microscopy, gene editing and generation of several cell lines) to address our biological question. Adding further validation will not add additional novelty.

3. p14, first line and Figure 6D, E: The authors state that 'the rate of differentiation diverged compared to that of WT myoblasts' in FAM134B knockout cells. However, they look at MF20 as the only marker of differentiation (without justifying that choice) and see that its levels rise later in the knockout but then increase more strongly than in the wildtype. This may indicate a problem in myogenesis, but it remains unclear what the altered expression pattern of MF20 means. I think more experiments are needed here to assess myotube formation. The authors used a count of myotubes/field of view in Figure 3I, K, which is a start. Additionally, I wonder if perhaps it could be informative to look at ER stress during differentiation. Seeing the massive but transient ER dilation during myotube formation, there may be an activation of the UPR in these cells and it would be interesting to know if this potential UPR activation is altered in the absence of FAM134B.

MF20 is the most used marker to investigate muscle differentiation *in vitro* and *in vivo*. Altered expression of MF20 are closely associated with abnormalities in myogenesis. We added an additional differentiation marker MEF2C to support our findings (New Fig EV6B,C,F). Moreover, we also add new measurement of myotubes formation upon FAM134B isoforms expression (New Figs 3L,M and EV7H) to better explain the role of FAM134B and ER-phagy in myotubes maturation.

Regarding the UPR activation, this is for sure an interesting point and it will likely deserve a separate manuscript.

We collected our data from the transcriptomic and proteomic, performed on Wild-Type and Fam134b^{KO} myoblasts and myotubes, and investigated the variations of the hits associated to the ER Stress/UPR. We reported here the results. We did not highlight major differences between the two genotypes. The heat maps represent the fold change of the mRNA and protein of the indicated genes in myoblasts (D2) and myotubes (D10).

In addition, we investigated the activation state of PERK that is one of the UPR markers. We did not see differences in the total amount of PERK but we could detect a small increase in its phosphorylation state in Fam134b^{KO} myotubes. We tried to investigate also ATF4 and CHOP but these Abs did not give good signal.

Minor comments:

1. p3, first paragraph: "Few studies indicate ...". I suspect the authors want to point out that there indeed are studies on ER proteins in myogenesis, not that there is only a small number of them. In this case, the sentence should read "A few studies indicate ...".

OK

2. p4, second paragraph: "helixes" should be helices.

OK

3. p4, second paragraph: when citing papers on the oligomerization of FAM134 (references 24 - 26), it would be good to include also Wang et al, JCB 2023, PMID: 37043189.

OK

4. p4, results, first paragraph: "... spontaneous differentiation in standard growth media ...". Was there only one growth medium or several media? If the former, then it should be "in standard growth medium ...".

Initially, we performed the experiments employing regular DMEM and a differentiation medium with 2% horse serum to be sure that the differentiation set up was not affecting the results. We did not notice major differences between the different media so we performed the experiment with the DMEM to have an easier setting (Fig EV1G-N and EV2D,E,H).

5. Figure 1A: Spell out 'GM'. Indicate that the schematic represents progression of time. Please explain in the text or the figure legend what MF20 is.

OK

6. p4, results, first paragraph. Please make explicit in the text that cells spontaneously differentiate once they reach confluency and state how long this takes. This becomes apparent when looking at the figures but explaining these things in the beginning would be helpful for readers unfamiliar with C2C12 cells.

We better explained this point in Methods section.

7. Figure EV1A: Please explain what MyoG is.

OK

8. Figure 1C: Please explain in the text or the figure legend what is plotted, especially in the

lower part of the panel. I assume that the lower part in some way shows a comparison of the abundance of various proteins in myoblasts versus myotubes. Not being a mass spectrometry expert, it was not clear to me why a PCA plot is shown rather than a simple volcano plot.

OK. We better explained in Figure Legend

9. p6, first paragraph and Figure 2A: In the text it says that "Western blot analysis of Lc3b ... confirmed a progressive activation of autophagy flux ...". The western and the accompanying quantification appears to show the levels of LC3B-I and -II. This does not allow to say anything about autophagic flux. For this, it is not sufficient to show the combined abundance of non-lipidated LC3B-I and lipidated LC3B-II but it would be necessary to show that their levels increase upon inhibition of lysosomal degradation (see, for example, PMID 17611390 or PMID 22966490).

We performed new experiments generating GFP-LC3B-RFP stable C2C12. This cell line allows to monitor the autophagy flux during C2C12 differentiation.

New data are reported in the New Fig 2C and New Fig 6F.

10. Figure 2C: Please label in the figure that ER-phagy is assayed with RFP-GFP-KDEL so that this is immediately obvious to the reader.

OK

11. p8, first paragraph: the N-terminal end of a protein is called N-terminus, not "N-terminal".

OK

12. Figure EV3D: Please explain why TGN38 was included and also what the comparison of Triton and Digitonin signifies.

We better explained this in Methods. TGN38 is a control for the experimental set up.

13. p8, second paragraph: "The embedded molecules perturbed of the bicelle". I think the "of" should be deleted.

OK

14. p8, last line: "...both molecules exhibited an almost similar ability". 'Almost similar' sounds odd. Two things can be similar or almost identical but not almost similar.

OK we fixed it

15. p9, first paragraph and Figure 3G, H: "Overexpression of FAM134B2 ... was sufficient to fragment ER membranes, albeit at a slower rate compared to FAM134B1." The micrographs show puncta of FAM134B, which in itself does not mean that the ER membrane fragments but could simply mean that FAM134B clusters within the ER. To show ER fragmentation, it would be necessary to show that a luminal ER marker forms physically separate structures or that FAM134B is transported to lysosomes (as in Figure 4D). Second, the snap shots in Figure 3G do not allow any statement about "rates".

We performed IF using Ab against HA together with specific Abs for ER proteins. The colocalization of HA signal with the ER proteins staining indicates that the ER membranes are subjected to remodeling that is driven by FAM134B (New Fig 3A). The fact that FAM134B dots are associated to a remodel of the ER membranes is also supported by previous literature on FAM134B: Khaminets et al. 2015 Nature; Bhaskara et al. 2019 Nature Comm; Reggio et al. 2021 EMBO Rep; Gonzalez et al. 2023 Nature.

We also performed additional IF using C2C12 expressing the lysosomal marker TMEM192-3xHA and staining the ER with an Ab specific for REEP5 to confirm the colocalization of the ER proteins and the lysosomal structures (New Figs EV2B). We performed similar experiments for FAM134B LIR mutant (Fig. 6D).

Regarding the rates, the snapshot is a representative picture that must be considered together the analysis reported in New Figs EV3D-G.

16. p9, first paragraph and Figure 3I, K: To show that FAM134B2 "contributed to myotube development" it would be necessary to show that FAM134B2 overexpression changes the

efficiency of myogenesis, i.e. it would be necessary to compare myotube number in FAM134B2-overexpressing and control cells. As is, it is impossible to say whether FAM134B1 or FAM134B2 overexpression has any impact.

To clarify this point, we added the control condition. We used ssRFP-GFP-KDEL that is cloned in the same viral backbone vector we employed for Fam134b1 and Fam134b2 overexpression. Moreover, we acquired more images to increase the number of counts and improve the statistical analysis (New Figs 3L,M). The effect of FAM134B2 is also now reported in the New Fig 7H.

17. p9, first paragraph: "regenerated less ER fragments". It should be 'fewer', not 'less'.

OK

18. p9, second paragraph: "FAM134B2 confers more stability of the ER network promoting myotubes development." I do not understand what is meant by "stability of the ER network" here. As for myotube development, see comment above.

FAM134B2 has a more static structure. It can vesiculate ER membranes but with a lower rate respect to FAM134B1. This fact is more evident in myotubes where the ER network has a particularly organized structure (New Fig 3J,K). The stability of the ER is linked to less propensity to form vesicles out of the ER network.

19. p10, first paragraph: Please explain the assay and how only FAM134B dimers are precipitated by reconstituted GFP. This is a very elegant idea but many readers won't be familiar with it.

OK. We better clarify this point in Methods

20. p11, second paragraph: "FAM134B1 exhibits ... but a higher affinity for ER-related proteins, autophagy adaptors and development elements." I do not think that the mass spec results allow any statement about affinities.

We edited this sentence

21. Figure 7F: The authors show fluorescence micrographs to illustrate the morphological impact of loss and re-expression of FAM134B. These images look promising - can they be quantified?

We provided a quantification in the New Fig. EV7G

Dear Paolo,

We have now received re-review reports from three referees, which I have included below. As you will see, reviewers #1 and #2 are both satisfied with the changes you have made. However, reviewer #3 has an extensive list of outstanding concerns. Though I would like you to respond to these points, I do not judge they need to be answered with another round of experimentation. Instead, please provide more detailed explanations, data presentation and method descriptions. A consideration of any outstanding points after these improvements should be added to a section of the discussion that deals with the study's limitations. I also suggest you temper the conclusion (from your over-expressor experiments) that Fam134b is required for the formation of myotubes, as opposed to their healthy maintenance.

In addition to these changes, there are some remaining editorial points which need to be addressed. In this regard would you please:

- in our online submission system, acknowledge funding from the Deutsche Forschungsgemeinschaft (DFG, German Research Foundation) CRC project on Selective Autophagy - grant numbers 259130777 (SFB1177); the EU/EFPIA/OICR/McGill/KTH/Diamond Innovative Medicines Initiative 2 Joint Undertaking (EUbOPEN grant n{degree sign} 875510); AFM-T  l  thon, grant number 23551, Fondazione Umberto Veronesi; and the ERC (ER-REMODEL),
- use an alphabetical reference format; for articles with more than ten authors, use '10 authors + et al.' in the reference section,
- rename the conflict of interest statement as the "Disclosure and competing interests statement",
- remove the author credit section from the manuscript text,
- complete and return the author checklist,
- limit your manuscript to five EV figures (the remaining two should be compiled in an appendix PDF with a table of contents with page numbers); the legends of these two figures should be removed and included in the appendix, from the manuscript file and included as a separate tab/sheet in each Excel file,
- remove dataset legends from the manuscript file and include as a separate tabs/sheets in each Excel file,
- include a 'Reagents and Tools' table,
- source data for EV6 should be removed from zip folder for Fig. 6 and uploaded as a separate zip folder (For EV and/or appendix figures, ZIP together all source data),
- replace author email for Malan Silva (malan.silva@fht.org) with a functioning address, and
- correct the section order so it reads: title page with complete author information, abstract, keywords, introduction, results, discussion, methods, data availability section, acknowledgements, disclosure and competing interests statement, references, main figure legends, tables, expanded figure legends.

We include a synopsis of the paper (see <http://emboj.embopress.org/>). Please provide me with a general summary image, a two-sentence summary statement and 3-5 bullet points that capture the key findings of the paper.

I am looking forward to receiving your revised manuscript.

EMBO Press is an editorially independent publishing platform for the development of EMBO scientific publications.

Best wishes,

William

William Teale, PhD
Editor
The EMBO Journal
w.teale@embojournal.org

- a point-by-point response to the referees' comments, with a detailed description of the changes made (as a word file).
 - a word file of the manuscript text.
 - individual production quality figure files (one file per figure)
 - a complete author checklist, which you can download from our author guidelines (<https://www.embopress.org/page/journal/14602075/authorguide>).
 - Expanded View files (replacing Supplementary Information)
- Please see out instructions to authors
<https://www.embopress.org/page/journal/14602075/authorguide#expandedview>
- a Reagents and Tools Table as part of the Methods section, which can be downloaded from our author guidelines (<https://www.embopress.org/page/journal/14602075/authorguide#structuredmethods>)

We realize that it is difficult to revise to a specific deadline. In the interest of protecting the conceptual advance provided by the work, we recommend a revision within 3 months (24th Feb 2025). Please discuss the revision progress ahead of this time with the editor if you require more time to complete the revisions. Use the link below to submit your revision:

Referee #1:

The authors have sufficiently answered all questions and concerns. No further revisions are requested.

Referee #2:

The authors have addressed my initial points and I fully support publication of the manuscript.

Referee #3:

In their revised manuscript, Buonomo and colleagues attempt to improve their study but unfortunately fall way short of resolving the concerns I raised in my previous review. Below is a detailed explanation of which of my points I think were insufficiently addressed, and why.

In my opinion, the authors essentially do not get to the chore of any of my major points, so that the manuscript remains rich in data and, for the most part, technically impressive but fails to define the functional significance of FAM134B-mediated ER-phagy in myogenesis. I therefore cannot change my previous assessment of this study and do not recommend publication in the EMBO Journal.

Point 1 (Figure 4E-H)

My previous concern was that the analysis of the IP data is superficial and does not allow a real assessment of the interactomes of dimeric FAM134B variants. The mass spectrometry of immunoprecipitated FAM134 variants identified hundreds of proteins, which is not unusual. However, in the analysis, all of these putative interactors were taken at face value which to me did not make sense given their number and the fact that the precipitates included many mitochondrial proteins. In their rebuttal, the authors responded with a technical explanation of data acquisition strategies for mass spectrometry, which is besides the point.

Point 2 (Figures 5, 6 and 7)

My point here was that the data were not intelligible and required experimental follow-up. The authors rework the presentation of the mass spectrometry data a bit but do not solve the basic problem that the data interpretation is vague. As experimental follow-up, they add a panel a new panel 6D, in which they provide a single image of wild-type and FAM134B knockout cells, purportedly showing that the ER protein REEP5 colocalizes with lysosomes in wild-type but not in FAM134B knockout cells. In my view, this is clearly insufficient.

Point 3 (Figure 6D, E, now Figures EV6 and EV7H)

The previous version of the manuscript contained little evidence that FAM134B-mediated autophagy was functionally important for myogenesis, which in my view was a critical point to address. Essentially the only piece of evidence was that protein levels of MF20 during the time-course of differentiation were abnormal in FAM134B knockout cells. In the revised version, the authors add MEF2C as a second marker and find that its protein abundance pattern during differentiation is essentially the same in control and knockout cells (panel EV6C). Furthermore, myogenesis, as assessed by the number of myotubes per field of view, does not appear to differ markedly between these cells (panel EV7H). Hence, the importance of FAM134B for myogenesis remains unclear.

Point 5 (Figure 1A)

The figure still shows MF20 staining without explanation why this marker was used.

Point 8 (Figure 1C)

From the explanation provided, I still cannot judge from the PCA plot in the lower panel how the authors arrive at the conclusion that the separation of samples from undifferentiated cells and myotubes in the proteomic analysis was 'primarily driven by alterations in sarcomere proteins' (p 4). Similarly, in Figure 1D I do not understand the plot from the explanation provided in the figure legend.

Point 6 (p 6 and Figure 2A)

I previously pointed out that the assay shown in Figure 2A is inappropriate to infer an upregulation of general autophagy during myogenesis and another referee made the same point (referee 2, major point 2). The authors responded to my criticism by leaving panel 2A and the relevant passage in the results section unchanged and adding new data obtained with a different assay (new panel 2C). Thus, they essentially ignored my comment about Figure 2A being problematic.

Furthermore, the new data are poorly explained. In the new panel 2C the authors show the general autophagy reporter GFP-LC3-RFP, plot the GFP/RFP ratio and state in the figure legend that the panel shows autophagic flux "represented by the GFP/RFP" ratio. In panel 2D, they use a similar reporter for ER-phagy (ssRFP-GFP-KDEL) but plot the RFP/GFP ratio and say in the figure legend that the panel shows ER-phagy flux "represented by the RFP/GFP ratio". The principle of how the two reporters in panel C and D work (quenching of GFP but not RFP fluorescence in lysosomes) is not explained anywhere. Even more seriously, the new data in panel 2C do not fit well with the results in panels A, B and D. There is no clear explanation of how the data shown with the general autophagy reporter in panel 2C (drop in GFP/RFP ratio starting at day 4, no recovery, suggesting sustained autophagy activation) fits with the data in panel 2A (rise in LC3B levels at day 4 and 6, return to baseline at day 8), panel 2B (which according to the results section illustrates that the number of autophagosome-like structures increased from day 0 to day 7 and then gradually decreased again by day 10) and panel 2D (transient activation of ER-phagy). All of this is certain to be confusing for the reader.

Point 15 (p 9 and previous Figure 3G, H, now Figure 3H-J)

I previously explained that single images showing FAM134B puncta is, in my view, insufficient to conclude that there is ER fragmentation because the puncta could simply reflect an accumulation of FAM134B in organelle subdomains that are still part of the ER. Furthermore, I voiced my opinion that single images do not allow any statements about rates of fragmentation. In their rebuttal letter, the authors respond by pointing to other figures in the manuscript that support their conclusion about ER fragmentation but they change nothing about their interpretation of Figure 3H-J.

Point 16 (p 9 and previous Figure 3I, K, now Figure 3L, M)

In the previous version of the manuscript, the authors overexpressed FAM134B1 or FAM134B2, assayed myogenesis (as judged by the number of myotubes per field of view) and concluded that FAM134B2 "contributed to myotube development". However, no control condition was present in these experiments. This control has been added now but there is no difference between control cells and FAM134B2-overexpressing cells. Nevertheless, the authors conclude the section 'FAM134B2 is an ER morphogen' by stating that "Moreover, FAM134B2 confers more stability to the ER network promoting myotube development". The new data contradict that statement. In the rebuttal letter, the authors make no reference to the discrepancy between their previous conclusion and the new data and simply say that they added more data. That is baffling.

Rebuttal

Referee #1:

The authors have sufficiently answered all questions and concerns. No further revisions are requested.

We thank the Referee for her/his positive feedback

Referee #2:

The authors have addressed my initial points and I fully support publication of the manuscript.

We thank the Referee for her/his positive feedback

Referee #3:

In their revised manuscript, Buonomo and colleagues attempt to improve their study but unfortunately fall way short of resolving the concerns I raised in my previous review. Below is a detailed explanation of which of my points I think were insufficiently addressed, and why.

In my opinion, the authors essentially do not get to the chore of any of my major points, so that the manuscript remains rich in data and, for the most part, technically impressive but fails to define the functional significance of FAM134B-mediated ER-phagy in myogenesis. I therefore cannot change my previous assessment of this study and do not recommend publication in the EMBO Journal.

We are sorry to read that the Referee did not consider the revision of the manuscript sufficient for publication. We are going to explain our point of view regarding the Referee's concerns in a point-by-point response.

Before that, we would like to shortly comment on the criticism of the reviewer, who in summary states that the manuscript fails to show the importance / functional significance of FAM134B short and long isoform in myogenesis.

The fact that FAM134B1 must gradually disappears during the differentiation process, and it is completely absent in differentiated myotubes, been replaced by the short isoform FAM134B2, is the strongest evidence of functional importance of the fine tuning of the proteins. The persistence of FAM134B1 is detrimental for ER dynamics as we showed keeping FAM134B1 active during myoblasts differentiation. The open questions are rather why and how this swap is facilitated.

While we believe that our study significantly contributes to answering both questions by the "rich in data and, for the most part, technically impressive" part, we never thought to be able to answer these "why and how" questions exhaustedly within a single study. We provided the first evidence and there will be additional publications to add knowledge.

Point 1 (Figure 4E-H)

My previous concern was that the analysis of the IP data is superficial and does not allow a real assessment of the interactomes of dimeric FAM134B variants. The mass spectrometry of immunoprecipitated FAM134 variants identified hundreds of proteins, which is not unusual. However, in the analysis, all of these putative interactors were taken at face value which to me did not make sense given their number and the fact that the precipitates included many mitochondrial proteins. In their rebuttal, the authors responded with a technical explanation of data acquisition strategies for mass spectrometry, which is besides the point.

In his/her previous concern the Reviewer was not convinced regarding the number of interactors “*I cannot envision that FAM134 dimers directly bind to several hundred proteins* “ For this reason, we clarify the point with more technical explanation regarding the mass spectrometry DIA setting. We used a setup that is commonly used by the scientific community; therefore, we consider our final data solid and properly analyzed.

The idea behind this dataset was to highlight the diversity in the interactome between the two isoforms of FAM134B and provide an overview of the differences that, on one hand, serves as a resource data set for future studies and, at the same time, supports other data (not mass spectrometry related) present in the manuscript.

The finding that the two interactomes indeed differ, indicates that (1) swap of the two isoforms is needed to support the turnover of a different subset of ER-phagy substrates and that (2) the associated machinery before and during activation may differ between the two isoforms. Both are important findings that need to be refined in future studies. For now, we grouped the interactors following their GO terms to give an easier interpretation of data.

Along these lines – mitochondria are known to form many contact points with the ER and presumably they need to be removed before degradation of certain ER subdomains. We are aware of several (unpublished) studies that are ongoing in the field. As such, we believe the co-immunoprecipitated mitochondrial proteins are indeed connected to the function of FAM134B, however we did not follow that line because it was not of our interest at this point.

Point 2 (Figures 5, 6 and 7)

My point here was that the data were not intelligible and required experimental follow-up. The authors rework the presentation of the mass spectrometry data a bit but do not solve the basic problem that the data interpretation is vague. As experimental follow-up, they add a panel a new panel 6D, in which they provide a single image of wild-type and FAM134B knockout cells, purportedly showing that the ER protein REEP5 colocalizes with lysosomes in wild-type but not in FAM134B knockout cells. In my view, this is clearly insufficient.

Figure 6D indicates that FAM134B is the major ER-phagy receptor not only in myoblasts but also in myotubes, because if another ER-phagy receptor would take over the function of FAM134B, REEP5 would start to colocalize again in similar amount with lysosomes upon

differentiation. The same result we obtained using the ER-phagy reported (Fig 6E) and with lyso-IP (Fig 6C). This is of important because – as the reviewer rightfully states – differentiation as such is not prevented by the lack of FAM134B. Given the impact of a differentiation process, we believe that this is also not to be expected. Instead, we expected and found a rather specific impact on the ER proteome, ER-phagy, and associated organelles/processes like ER membrane reshaping. We provided evidence of these aspects employing different techniques and obtaining the same results.

Along our initial statement, we agree with the reviewer that (several) experimental follow-ups are needed to comprehensively understand the process. However, in follow up studies, as this clearly succeeds the scope of the current manuscript. We described the first link between FAM134B2 and the differentiation processes in muscle. As such, we do not consider our reported findings vague, but findings that open to new (therefore to be further studied) investigations in the field of ER biology / ER-phagy.

Point 3 (Figure 6D, E, now Figures EV6 and EV7H)

The previous version of the manuscript contained little evidence that FAM134B-mediated autophagy was functionally important for myogenesis, which in my view was a critical point to address. Essentially the only piece of evidence was that protein levels of MF20 during the time-course of differentiation were abnormal in FAM134B knockout cells. In the revised version, the authors add MEF2C as a second marker and find that its protein abundance pattern during differentiation is essentially the same in control and knockout cells (panel EV6C). Furthermore, myogenesis, as assessed by the number of myotubes per field of view, does not appear to differ markedly between these cells (panel EV7H). Hence, the importance of FAM134B for myogenesis remains unclear.

We can follow the reviewer's argument in the aspect that MEF2C is less deregulated than MF20. However, we believe that the - small yet clearly visible (EV6C and EV6F) - differences in protein levels are indeed relevant and indicate a delay / difference in the efficiency of myogenesis.

Regarding Fig EV7H - there is indeed a significant difference between the genotypes. We focused on the rescue of the phenotype and missed to add the statistic between wild-type and Fam134b^{KO} myotubes, which were now added. We also analyzed additional new imagines to substantiate our findings.

Point 5 (Figure 1A)

The figure still shows MF20 staining without explanation why this marker was used.

MF20 indicates that we had a proper differentiation of myotubes. MF20 is a marker for muscle differentiation and it is not expressed in myoblasts. We now added this in the Figure Legend.

Point 8 (Figure 1C)

From the explanation provided, I still cannot judge from the PCA plot in the lower panel how the authors arrive at the conclusion that the separation of samples from undifferentiated cells and myotubes in the proteomic analysis was 'primarily driven by alterations in sarcomere proteins' (p 4). Similarly, in Figure 1D I do not understand the plot from the explanation

provided in the figure legend.

We better clarify these points in Figure Legends. Plots are elaborated, as reported in the Method part, from the Datasets uploaded as supplementary material. We understand that mass spectrometry data could be complicated for readers unfamiliar with the technique. However, we used well established methods and elaborated data with commonly used programs in the mass spectrometry field. Therefore, an exhaustive explanation exceeds the scope of this paper.

Figure 1C lower panel reports the loadings of PCA. The mentioned scatter plot represents the bidimensional distribution of all the profiled proteins contributing to drive separation in PCA (Fig 1C upper panel). As the referee can observe, proteins annotated as “sarcomere” (according to GOCC) distribute in the space where myotubes are segregated meaning that such proteins contribute to distinguish these cells from their undifferentiated counterpart.

Point 6 (p 6 and Figure 2A)

I previously pointed out that the assay shown in Figure 2A is inappropriate to infer an upregulation of general autophagy during myogenesis and another referee made the same point (referee 2, major point 2). The authors responded to my criticism by leaving panel 2A and the relevant passage in the results section unchanged and adding new data obtained with a different assay (new panel 2C). Thus, they essentially ignored my comment about Figure 2A being problematic.

We did not ignore this Referee’s comment, on the contrary we addressed the point by providing a complementary assay and thereby substantiated our claim that autophagy is involved.

The same point was indeed made by the other Referees, and they were both convinced after the new data set was provided. Moreover, they did not ask to remove the panel or change the results.

Furthermore, the new data are poorly explained. In the new panel 2C the authors show the general autophagy reporter GFP-LC3-RFP, plot the GFP/RFP ratio and state in the figure legend that the panel shows autophagic flux "represented by the GFP/RFP" ratio. In panel 2D, the use a similar reporter for ER-phagy (ssRFP-GFP-KDEL) but plot the RFP/GFP ratio and say in the figure legend that the panel shows ER-phagy flux "represented by the RFP/GFP ratio". The principle of how the two reporters in panel C and D work (quenching of GFP but not RFP fluorescence in lysosomes) is not explained anywhere.

Even more seriously, the new data in panel 2C do not fit well with the results in panels A, B and D. There is no clear explanation of how the data shown with the general autophagy reporter in panel 2C (drop in GFP/RFP ratio starting at day 4, no recovery, suggesting sustained autophagy activation) fits with the data in panel 2A (rise in LC3B levels at day 4 and 6, return to baseline at day 8), panel 2B (which according to the results section illustrates that the number of autophagosome-like structures increased from day 0 to day 7 and then gradually decreased again by day 10) and panel 2D (transient activation of ER-phagy). All of this is certain to be confusing for the reader.

We believe to have sufficiently explained the principle of how the two different reporters work in Methods session and providing References. We further elaborate the technique

explanation for clarity. Both, western blot of LC3B I/II and the dual reporter systems are well established in the field and known to be complementary.

The decision which fluorophore is in the denominator depends on the abundance of each species and therefore on experience. For LC3, a GFP/RFP ratio is commonly used, for KDEL the opposite. We employed these tools as we and other groups did in several other publications.

An increase in LC3II may indicate autophagy induction, however, could also indicate a block in autophagosome-lysosomal fusion or lysosome de-acidification. Autophagy induction indicated by the reporter system depends on functional lysosomal fusion and acidification. At the same time, WB analysis is based on the endogenous protein but does not distinguish between cytosolic and lysosomal LC3B II localization, the reporter assay does but is based on overexpression.

In summary, our data indicates an activation of general autophagy during myogenesis and a higher basal level of autophagy in myotubes compared to myoblasts. However, based on our understanding of the used tools (we employed these reporters as suggested in the original publications) the same precise x-fold cannot be stated.

ER-phagy flux seems to be considerably increased during the differentiation process, but seems to slow down again, once differentiation is completed. Also here, a slightly higher basal level may be reported. However, the KDEL-based reporter has a broad spectrum and the basal flux of specific ER-phagy paths may differ a lot / be more or less active in myotubes compared to myoblasts.

We used the available and most employed tools in the field.

Point 15 (p 9 and previous Figure 3G, H, now Figure 3H-J)

I previously explained that single images showing FAM134B puncta is, in my view, insufficient to conclude that there is ER fragmentation because the puncta could simply reflect an accumulation of FAM134B in organelle subdomains that are still part of the ER. Furthermore, I voiced my opinion that single images do not allow any statements about rates of fragmentation. In their rebuttal letter, the authors respond by pointing to other figures in the manuscript that support their conclusion about ER fragmentation but they change nothing about their interpretation of Figure 3H-J.

We provided quantification for every data we presented. For IF we provided a single representative image together with quantification of many more images.

FAM134B has the intrinsic property of fragmenting the ER membrane in discrete portions. This property has been demonstrated since the very first paper on the role of FAM134B as ER-phagy receptor and ER morphogen. The same phenotype (fragmentation of discrete portions of ER) has been observed in several other manuscripts (PMID: 26040720; 28617241; 30559329; 31147549; 37225996; 37225994; 38102139; 39094564; 38705724; 34338405; 33591770; 31930741).

Point 16 (p 9 and previous Figure 3I, K, now Figure 3L, M)

In the previous version of the manuscript, the authors overexpressed FAM134B1 or FAM134B1, assayed myogenesis (as judged by the number of myotubes per field of view) and concluded that FAM134B2 "contributed to myotube development". However, no control

condition was present in these experiment. This control has been added now but there is no difference between control cells and FAM134B2-overexpressing cells. Nevertheless, the authors conclude the section 'FAM134B2 is an ER morphogen' by stating that "Moreover, FAM134B2 confers more stability to the ER network promoting myotubes development". The new data contradict that statement. In the rebuttal letter, the authors make no reference to the discrepancy between their previous conclusion and the new data and simply say that they added more data. That is baffling.

We don't think there are discrepancies in our data. Over-expressing FAM134B2 in myotubes, where FAM134B2 is physiologically present, does not necessary mean that the protein needs to have an additive effect. Conferring stability and promoting myogenesis does not mean necessary making more myotubes rather ensuring that the myotubes are properly made (in our specific case, the ER is properly reshaped for example). We better explained this in the text, probably we were not sufficiently clear.

On the contrary, over-expression of FAM134B1, that under physiological conditions progressively disappear in myotubes, destabilizes the physiological process and the number of myotubes are less. The persistence of FAM134B1 is detrimental and we provided several evidence for this.

Moreover, when we reconstituted Fam134b^{KO} myotubes with Fam134b2 we had a perfect rescue of the phenotype with numbers identical to wild-type myotubes so, again, the importance is to ensure that the myotubes are properly done, not that there are more myotubes respect to the physiological condition. The value of this is even more evident when we observed that Fam134b1 had an "over-correcting" effect, and this data confirms even more that the previous observations were valuable.

Dear Paolo,

I am pleased to inform you that your manuscript has been accepted for publication in the EMBO Journal.

Congratulations to you and your team on a really interesting study!

Best wishes,

William

William Teale, PhD
Editor
The EMBO Journal
w.teale@embojournal.org
